# Precancerous niche remodelling dictates nascent tumour persistence

G. Skrupskelyte[1,2,18 ✉], J. E. Rojo Arias[1,2,16,18], H. Ajith[1,2], Y. Dang[3,4,5], D. Rossetti[1,2], S. Han[6], M. K. S. Tang[1,2], M. T. Bejar[1,2], B. Colom[7,17], J. C. Fowler[7], K. Murai[7], W. Knight[7], D. Aust[8,9], M. H. H. Schmidt[10], J. Jászai[10], S. Zeki[11], A. Noorani[7,12], P. H. Jones[7,13], S. Rulands[4,14], B. D. Simons[1,6,15] & M. P. Alcolea[1,2 ✉]

Interactions between mutant cells and their environment have a key role in determining cancer susceptibility[1-3]. However, understanding of how the precancerous microenvironment contributes to early tumorigenesis remains limited. Here we show that newly emerging tumours at their most incipient stages shape their microenvironment in a critical process that determines their survival. Analysis of nascent squamous tumours in the upper gastrointestinal tract of the mouse reveals that the stress response of early tumour cells instructs the underlying mesenchyme to form a supportive 'precancerous niche', which dictates the long-term outcome of epithelial lesions. Stimulated fibroblasts beneath emerging tumours activate a wound-healing response that triggers a marked remodelling of the underlying extracellular matrix, resulting in the formation of a fibronectin-rich stromal scaffold that promotes tumour growth. Functional heterotypic 3D culture assays and in vivo grafting experiments, combining carcinogen-free healthy epithelium and tumour-derived stroma, demonstrate that the precancerous niche alone is sufficient to confer tumour properties to normal epithelial cells. We propose a model in which both mutations and the stromal response to genetic stress together define the likelihood of early tumours to persist and progress towards more advanced disease stages.

Groundbreaking studies in human genomics over the past decade have revealed that our healthy tissues accumulate cancer-associated mutations with age[4-8]. These observations highlight new levels of complexity in the early pathophysiology of cancer, raising the question of what other factors, beyond cancer mutations, may have a role during early carcinogenesis.

Models of early tumours spanning a range of epithelial tissues, including oesophagus, skin and intestine, have started to offer a clearer understanding of what drives tumour initiation[1,3,9-11]. Work in this area has shown that tumour formation represents more than the mere accumulation of genetic alterations, highlighting the important role of environmental cues and non-genetic mechanisms in this process[3,12-15]. Indeed, mounting evidence indicates that the predisposition of a mutated epithelium to develop tumoral lesions depends on complex interactions between mutant cells and their dynamic surroundings. Coexisting mutant clones could either synergize or compete, contributing to early tumour initiation[16,17]. Indeed, even after tumours have formed, the presence of neighbouring mutant clones can continue to influence tumorigenesis[1]. Alternative environmental cues, such as the stiffness of the extracellular matrix (ECM)[3,13], as well as direct cell–cell communication between mutant cells and non-mutant cells[2,9-11,14], have also been shown to affect the expansion of mutant clones, susceptibility to tumour initiation and invasion[18-20]. Despite this, understanding of the mechanisms by which environmental factors determine the formation and long-term persistence of emerging tumours remains limited.

Previous studies using an oesophageal early-tumour model demonstrated that not all nascent tumours have the same chance of survival. Most tumours are cleared from the tissue soon after formation by competition with neighbouring mutant clones. Surviving tumours instead persist long term, becoming susceptible to cancer progression[1]. But a key question remains: how are precancerous tumours able to withstand the competitive mutant environment that surrounds them? Understanding the processes underlying early tumour persistence and the relevance of the microenvironment at pre-neoplastic stages provides a critical opportunity to dissect the mechanisms driving precancer progression, opening new avenues to halt cancer in its tracks.

[1]Cambridge Stem Cell Institute, Jeffrey Cheah Biomedical Centre, University of Cambridge, Cambridge, UK. [2]Department of Physiology, Development and Neuroscience, University of Cambridge, Cambridge, UK. [3]Max Planck Institute for Molecular Cell Biology and Genetics, Dresden, Germany. [4]Max Planck Institute for the Physics of Complex Systems, Dresden, Germany. [5]Center for Systems Biology, Dresden, Germany. [6]Gurdon Institute, University of Cambridge, Cambridge, UK. [7]Wellcome Sanger Institute, Hinxton, UK. [8]University Hospital Carl Gustav Carus Dresden, Faculty of Medicine of TUD Dresden University of Technology, Dresden, Germany. [9]Institute of Pathology, University Hospital CGC Dresden, TU Dresden, Dresden, Germany. [10]Institute of Anatomy, Faculty of Medicine of TUD, University of Technology, Dresden, Germany. [11]Department of Gastroenterology, Guy's and St. Thomas' Hospital, London, UK. [12]Addenbrooke's Hospital, Cambridge University Hospital NHS Trust, Cambridge, UK. [13]Department of Oncology, University of Cambridge, Hutchison Research Centre, Cambridge Biomedical Campus, Cambridge, UK. [14]Arnold Sommerfeld Center for Theoretical Physics, Ludwigs-Maximilians-Universität Munchen, Munich, Germany. [15]Department of Applied Mathematics and Theoretical Physics, Centre for Mathematical Science, University of Cambridge, Cambridge, UK. [16]Present address: RhyGaze, Basel, Switzerland. [17]Present address: Cambridge Institute of Science, Altos Labs, Cambridge, UK. [18]These authors contributed equally: G. Skrupskelyte, J. E. Rojo Arias. ✉e-mail: gs463@cam.ac.uk; mpa28@cam.ac.uk

Here we combine single-cell RNA sequencing with lineage tracing and 3D heterotypic cultures to study the unique features of the few nascent tumours that escape the existing protective barriers preventing tumorigenesis. We demonstrate that, during the earliest stages of tumour development, fibroblasts react to the pre-neoplastic epithelium by promoting the formation of a fibrotic precancer niche that, in turn, feeds back on the epithelium favouring early tumour growth and survival.

## Precancerous tumour persistence

To study the processes that underlie the persistence of pre-neoplastic nascent tumours, we used a well-established, clinically relevant mouse model of upper gastrointestinal tract (including oesophagus and forestomach) tumorigenesis driven by a mutagen found in tobacco smoke (diethyl-nitrosamine (DEN))[1,21] (Extended Data Fig. 1a–j). After DEN treatment, the tissue becomes an evolving patchwork of mutant clones competing for space and survival, recapitulating the complex mutational landscape of the normal human ageing oesophagus[22]. This results in the emergence of pre-neoplastic squamous tumours with the potential to persist long-term (Extended Data Fig. 1a–c).

Nascent epithelial tumours, marked by KRT17 (keratin 6A (KRT6A) and keratin 17)[1], can be detected in tissue whole-mounts from their most incipient stages, from as early as 10 days after DEN treatment (Extended Data Fig. 1d–e). The emerging tumours are microscopic, containing as few as 10 cells, and are characterized by their distinctive rosette-like structure[1,21] (Extended Data Fig. 1d,f). This brief window of formation is followed by a tumour-clearing process, in which more than one-third of the initial tumours are progressively eliminated[1]. The surviving tumours can persist in the tissue for more than a year, largely as low-grade dysplasia (pre-neoplastic or precancer stages), with sporadic progression to invasive squamous cell carcinomas[1] (Extended Data Fig. 1c,g,h), mimicking human carcinogenesis[23]. As a result, only a subset of the original tumours survive long term, enabling us to study the mechanisms that modulate precancerous tumour persistence.

To understand what drives early tumour survival, we first set out to compare the phenotypic traits of nascent tumours and those persisting long term (10 days and more than 8 months, respectively, after DEN treatment; Fig. 1a). Histological analysis showed that persistent dysplastic tumours (Extended Data Fig. 1h) were characterized by a prominent stromal remodelling (Fig. 1b). These nest-like structures were formed by stromal fibroblasts (PDGFRα⁺) that protruded towards the epithelial compartment, seemingly enclosing early tumours to create a supportive scaffold or a 'precancerous niche' (Fig. 1b,c). Unlike in persisting tumours, at nascent stages, most epithelial lesions (around 70%; 199 of 296) showed no apparent stromal reorganization (Fig. 1c, d), denoting the existence of two phenotypically different nascent tumour subtypes, referred to here as Niche+ and Niche− (Fig. 1c).

Next, we assessed the dynamic nature of these two nascent tumour subtypes. We found that the number of Niche+ tumours, despite constituting the minority of all initial tumours, remained constant over time, whereas the number of Niche− tumours decreased markedly (Extended Data Fig. 2a). As a result, the tissue became progressively enriched in Niche+ tumours, with most (around 82%; 65 of 79) showing a supportive stromal scaffold by 8 months following DEN treatment (Fig. 1d). This enrichment in Niche+ tumours prompted us to explore whether stromal remodelling was associated with nascent tumour persistence. We found that Niche+ lesions were hyperproliferative and were more likely to persist and enlarge than Niche− tumours were (Fig. 1e–g and Extended Data Fig. 2b–f). Close analysis revealed that keratinocytes in contact with the niche showed a particularly high proliferative activity (Extended Data Fig. 2e), indicating active epithelial–stromal communication at precancerous stages. These results were further reinforced by observations in the squamous forestomach, where long-term pre-neoplastic tumours also exhibited a remodelled stromal niche (Extended Data Fig. 2g,h).

Collectively, our data support a model in which the remodelled stromal scaffold acts as a 'precancerous niche' promoting tumour growth and survival. These observations link stromal remodelling in nascent tumours with pre-neoplastic tumour progression.

## Niche signals drive tumour traits

The importance of the precancerous niche during early tumorigenesis became evident in 3D heterotypic cultures. These cocultures revealed that signals from the early tumour stroma are sufficient to confer tumour features to epithelial cells that had never been exposed to carcinogens[24] (Extended Data Fig. 2i–l). Using reporter mouse lines to track the tissue origin, we found that untreated phenotypically normal epithelium directly exposed to the denuded tumour niche (lacking the epithelial compartment) acquired a tumour-like morphology and became highly proliferative, reaching levels similar to those of early tumours in vivo (Extended Data Fig. 2k,l).

Moreover, heterotypic tissue constructs grafted into immune-deficient NOD-SCID-γ mice (NSG; NOD.Cg-*Prkdc*[scid] *Il2rg*[tm1Wjl]/SzJ) showed that the pro-survival phenotype conferred by the tumour niche was also observed in vivo. Normal epithelial cells were more likely to engraft long term when exposed to early tumour stromal signals (Extended Data Fig. 2m–o).

Overall, these results demonstrate that the early tumour microenvironment promotes epithelial cell growth, favouring precancerous tumour survival and, ultimately, disease progression.

## Local fibroblasts form nascent tumour niche

Given the key role of the niche in nascent tumour survival, we next explored its cellular composition.

Under normal conditions, the squamous upper gastrointestinal tract is characterized by three distinct layers of stromal tissue: the lamina propria, a thin loose connective tissue directly beneath the epithelium; the muscularis mucosae, a layer of smooth muscle cells; and the submucosae, a dense irregular lower stromal compartment[25] (Extended Data Fig. 3a–d). In line with observations in other epithelial tissues[26,27], immunofluorescence analysis revealed two fibroblast populations that showed distinctive tissue compartmentalization and morphology and different expression levels of the pan-fibroblast marker PDGFRα (Extended Data Fig. 3b–d). PDGFRα[low] fibroblasts resided in the upper stroma (the lamina propria), whereas PDGFRα[high] fibroblasts populated the deepest stromal layer (the submucosae; Extended Data Fig. 3c,d). Histological analysis of emerging tumours revealed that the main cellular component of the precancerous niche was PDGFRα[low] fibroblasts, phenotypically indistinguishable from neighbouring lamina propria fibroblasts (Fig. 2a). Further characterization showed that endothelial and immune cells were largely absent from the niche in nascent tumours (10 days after DEN treatment; Fig. 2b,c).

Since the role of cancer-associated fibroblasts (CAFs) in tumorigenesis, drug resistance and disease progression is well recognized[28,29], we next assessed whether niche-forming fibroblasts exhibited CAF features. Except for the nuclear localization of YAP (active YAP, aYAP; Extended Data Fig. 3e), the expression of CAF markers, such as fibroblast activation protein (FAP) and α-smooth muscle actin (α-SMA), was not detectable in 10-day tumour fibroblasts (Extended Data Fig. 3f). We conclude that, although Niche+ fibroblasts in incipient tumours lack a full CAF phenotype, their aYAP status is consistent with a pre-CAF transitional state in the nascent tumour niche[30].

Another important stromal contributor to tumorigenesis and cancer progression is the immune compartment[31]. Immune-cell characterization of Niche+ and Niche− tumours, however, did not show significant differences (Fig. 2b,c and Extended Data Fig. 3g). Accordingly, the

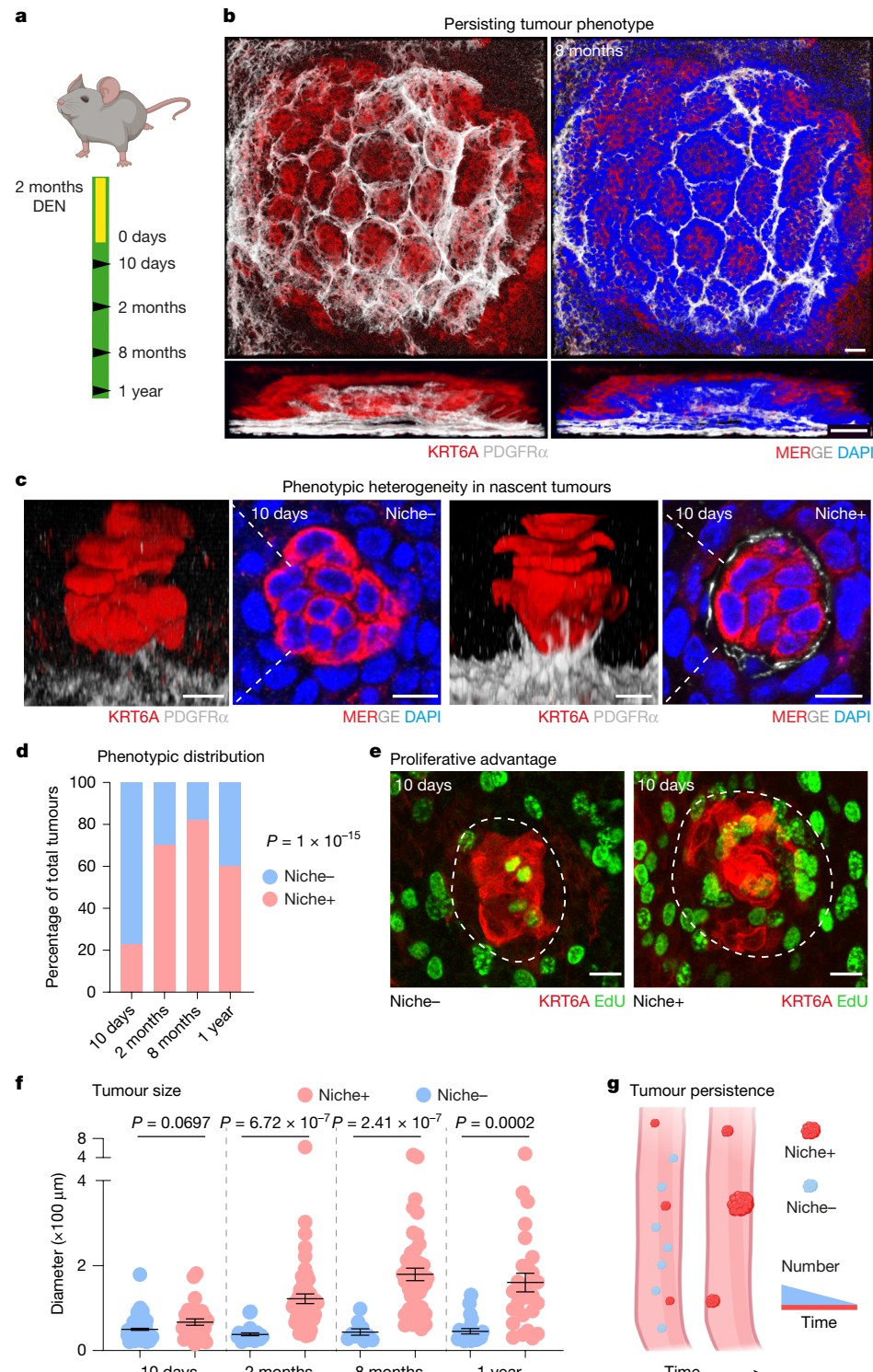

**Fig. 1 | Precancerous niche remodelling is linked to long-term tumour persistence. a**, The experimental DEN carcinogen protocol. Wild-type mice were exposed to DEN in the drinking water for 2 months. Tissues were collected at 10 days, 2 months, 8 months and 1 year. **b,c**, Representative confocal images of long-term-persisting tumours 8 months after DEN treatment (**b**) and nascent tumours 10 days after DEN treatment (**c**), stained for DAPI (blue), KRT6A (tumour marker; red) and PDGFRα (fibroblast marker; grey). Scale bars: 50 μm (**b**) and 10 μm (**c**). Image settings were adjusted to the upper stromal layer. **d**, Percentage of Niche+ and Niche− tumours at the indicated time points after DEN administration from three mice per time point; statistical significance was determined by a one-sided chi-squared test. **e**, Confocal images showing the incorporation of 5-ethynyl-2′-deoxyuridine (Edu; green)

in KRT6A⁺ nascent tumours (red, dashed line) 10 days after DEN treatment. Scale bars: 10 μm. Images were generated omitting the uppermost suprabasal layer. **f**, Diameter (×100 μm) of Niche+ (red) and Niche− (blue) tumours at the indicated time points after DEN treatment. Tumours were quantified in three mice per time point: at 10 days, $n$ = 128 tumours; at 2 months, $n$ = 84 tumours; at 8 months, $n$ = 53 tumours; and at 1 year, $n$ = 49 tumours. Data are expressed as mean ± s.e.m. Two-tailed Welch's $t$-test comparing Niche− versus Niche+ tumours. **g**, Cartoon illustrating the association between tumour niche remodelling and long-term tumour survival. Red, surviving tumours; blue, disappearing tumours. Illustrations in **a** and **g** were created in BioRender; Alcolea, M. https://BioRender.com/0g5wodl (2026).

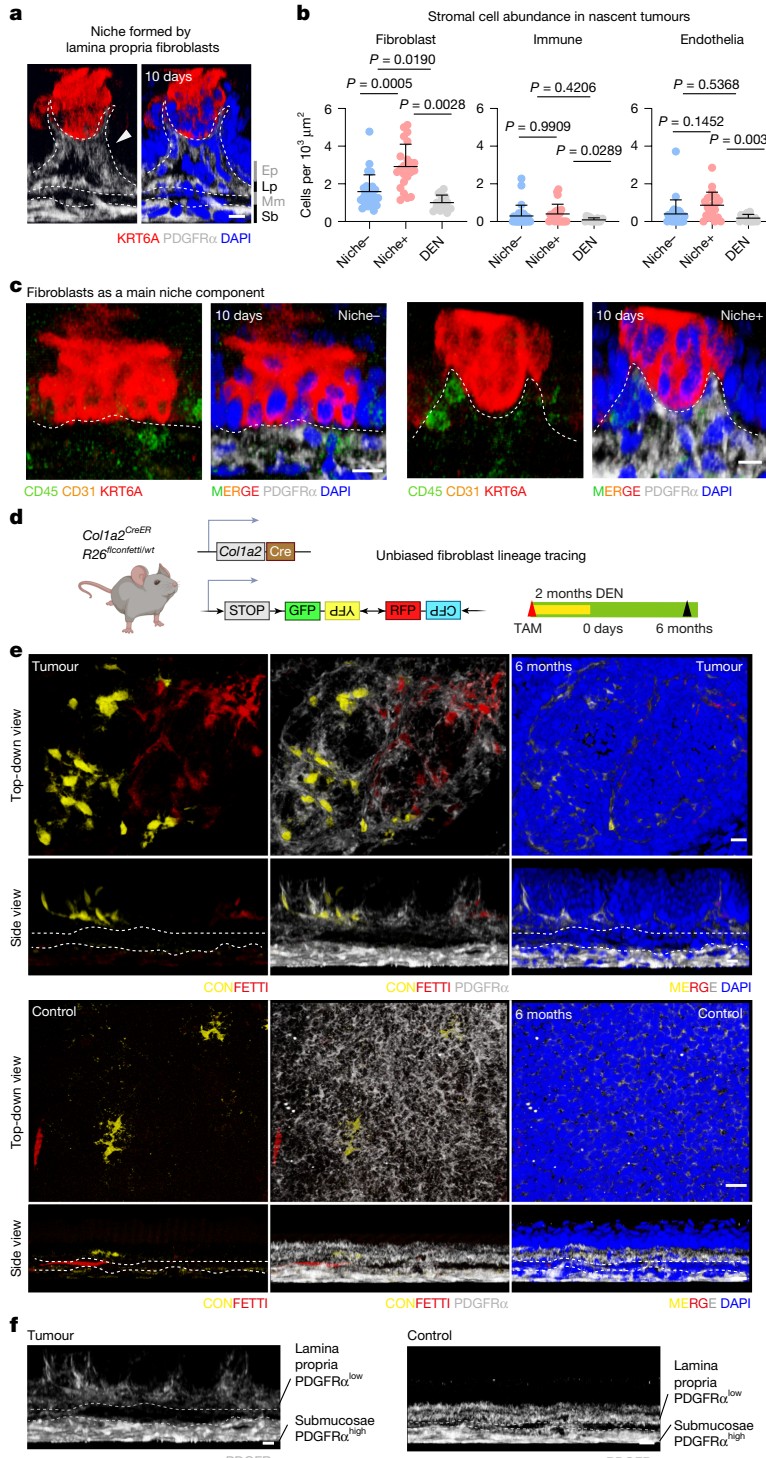

**Fig. 2 | The early tumour niche is formed by a PDGFRα^low fibroblast population. a**, Representative confocal image of a nascent Niche+ tumour 10 days after DEN withdrawal. The side view shows that the niche is composed of lamina propria (Lp), not submucosae (Sb), fibroblasts. Mm, muscularis mucosae; Ep, epithelium. Dashed lines show the layers. The white arrowhead points to the nascent tumour niche arising from lamina propria. Blue, DAPI; red, KRT6A; grey, PDGFRα. Scale bar, 25 μm. **b**, Number of stromal cells in tumour-free DEN tissue, Niche− and Niche+ tumours per unit of surface area, 10 days after DEN withdrawal; n = 27 Niche−, n = 22 Niche+, n = 15 (DEN) areas, from 3 mice; dots represent each area; PDGFRα+ fibroblasts, CD45+ immune cells and CD31+ endothelial cells are shown. Data are expressed as mean ± s.e.m. Statistical significance was assessed by one way Welch's analysis of variance (ANOVA) with multiple comparisons. **c**, 3D-rendered confocal side views of Niche− and Niche+ tumours 10 days after DEN withdrawal; green, CD45; orange,

CD31 (absent); red, KRT6A; grey, PDGFRα. Dashed lines show the basal membrane. Scale bar, 10 μm. Image settings were adjusted to the upper stromal layer. **d**, Experimental protocol for fibroblast lineage tracing: *Col1a2^CreER* and *R26^FlConfetti/wt* mice received a dose of tamoxifen (TAM) followed by DEN treatment. Samples were collected 6 months after DEN treatment. **e**, Representative top-down (top) and side views (bottom) of control and tumour tissue from **d**, Grey, PDGFRα; yellow and red, lineage-traced Confetti+ cells. Scale bars, 50 μm. Dashed lines separate stromal compartments. Control, n = 3 mice; DEN, n = 4 mice. **f**, A single channel from **e** shows a marked difference in PDGFRα expression across the two stromal compartments, both in control and tumour samples; lamina propria, PDGFRα^low (underlying epithelium); submucosae, PDGFRα^high (deeper stromal layer). The illustration in **d** was created in BioRender; Alcolea, M. https://BioRender.com/hwbs32m (2026).

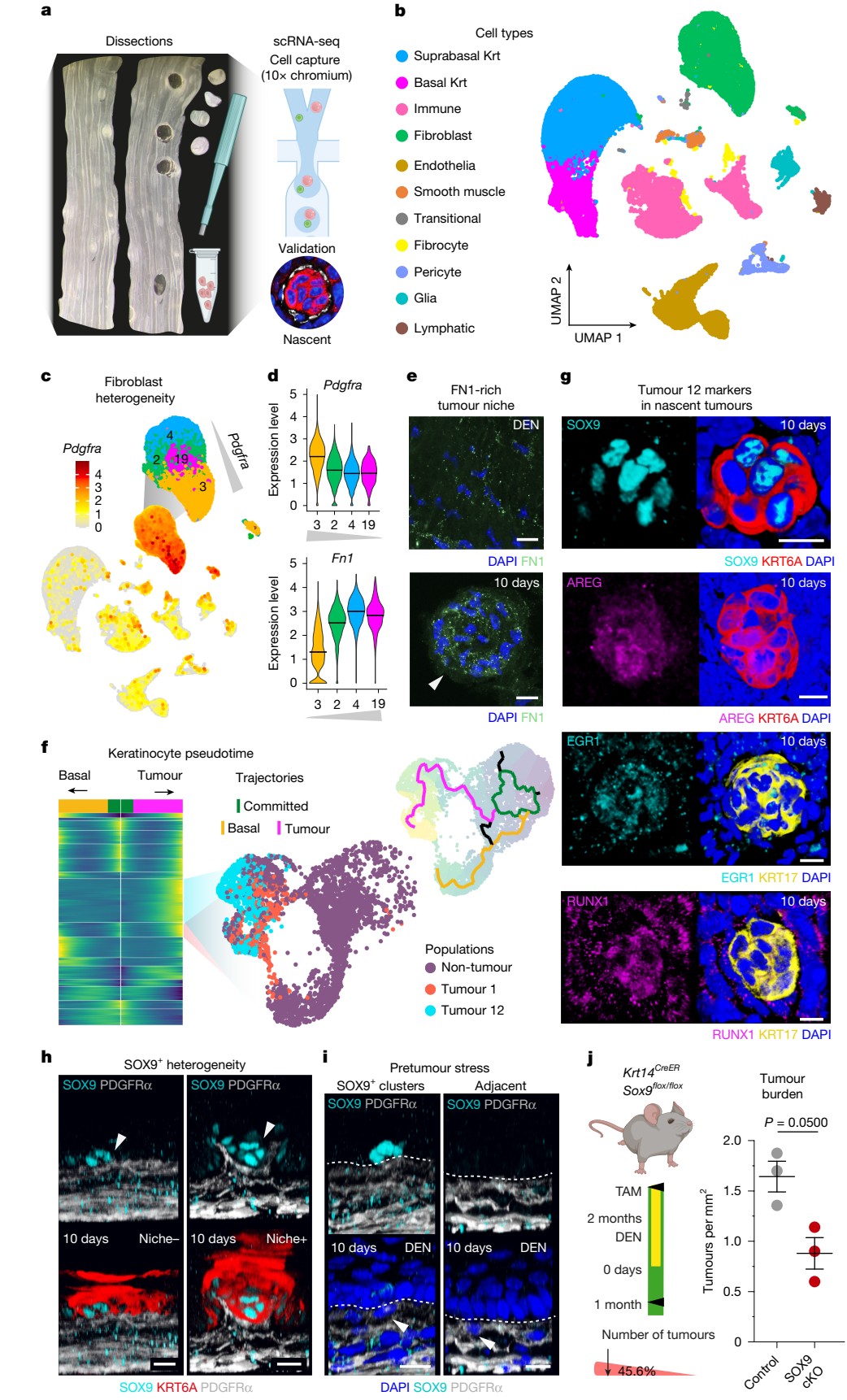

**Fig. 3** | See next page for caption.

**Fig. 3 | Nascent tumour heterogeneity in the epithelial compartment is linked to stromal remodelling. a**, Microdissection of squamous upper gastrointestinal tract 8 months after DEN treatment for single-cell RNA sequencing. **b**, Uniform manifold approximation and projection (UMAP) showing cell-type annotation. Krt, keratinocytes. **c**, Heterogeneous expression of *Pdgfra* in fibroblasts in the UMAP space. The inset shows fibroblast clusters. **d**, Violin plots showing levels of *Pdgfra* and *Fn1* expression in fibroblast clusters. Black line, mean. **e**, Representative images from 6 mice of the DEN area and nascent tumour stroma 10 days after DEN treatment, showing the accumulation (white arrowhead) of fibronectin (FN1, green) in the niche. Blue, DAPI. Scale bars, 10 μm. **f**, Heatmap (left) of the top 1,500 differentially expressed genes along the basal keratinocyte pseudotime trajectory. Pseudotime trajectories, top right (blue, committed; yellow, basal; magenta, tumour). Populations representing two tumour transcriptional modules (tumour 1, red; and tumour 12, cyan; bottom right UMAP). **g**, Representative images from 6 mice, showing tumour 12 markers in nascent tumours 10 days after DEN treatment, showing homogeneous KRT6A (red) and KRT17 (yellow); heterogenous SOX9 and EGR1 (cyan); and AREG and RUNX1 (magenta); DAPI (blue). Scale bars, 10 μm. **h**, Representative images showing SOX9 distribution in Niche− and Niche+ tumours 10 days after DEN treatment. White arrowheads highlight SOX9⁺ keratinocytes (cyan), KRT6A (red) and PDGFRα (grey). Scale bars, 10 μm. **i**, Images of SOX9⁺ clusters in DEN-treated tumour-free areas 10 days after DEN treatment. The control was the adjacent area negative for SOX9. Blue, DAPI; cyan, SOX9; grey, PDGFRα. Scale bars, 10 μm. White arrowheads highlight keratinocyte to fibroblast proximity; the white dashed line shows the epithelia to stroma border. **j**, Left, experimental protocol: *Krt14^CreER^;Sox9^flox/flox^* mice received tamoxifen (TAM) followed by DEN. Tissues were collected 1 month after DEN treatment and compared with DEN-treated uninduced controls. Right, quantification of tumour burden; *n* = 3 mice per condition; data shown as mean ± s.e.m.; one-tailed Mann–Whitney test. Images captured by confocal microscopy. Illustrations in **a** and **j** were created in BioRender; Alcolea, M. https://BioRender.com/xjxwb1m (2026).

emergence and persistence of Niche+ and Niche− tumours remained unaltered in immune-deficient mice (NSG; Extended Data Fig. 3h–l). These results indicate that immune cells do not discriminate between nascent Niche+ and Niche− tumours, acting as bystanders in early tumour persistence.

## Precancerous stromal reorganization

To better understand the contribution of stromal fibroblast to tumour niche formation, we used an unbiased genetic lineage-tracing approach to target fibroblasts. Sporadic confetti labelling of fibroblasts, across the lamina propria and submucosae compartments, was induced in *Col1a2-Cre^ER^;R26^FlConfetti/WT^* mice followed by DEN treatment (Fig. 2d and Extended Data Fig. 4a–d). To trace the origin of the tumour 'niche', recombination of the confetti cassette was induced before DEN treatment (Fig. 2d). Analysis of confetti clones 6 months after DEN treatment showed that fibroblasts in the niche underwent clonal expansion (Fig. 2e, Extended Data Fig. 4e–i and Supplementary Table 1). Immunostaining further revealed that clones in the niche were formed by lamina propria PDGFRα^low^ fibroblasts (Fig. 3e,f and Extended Data Fig. 4e).

To confirm this, we traced the PDGFRα^low^ and PDGFRα^high^ fibroblast populations separately in *Pdgfra-Cre^ER^;R26^FlConfetti/WT^* mice (Extended Data Fig. 4j–o). We reasoned that differential PDGFRα expression levels would enable us to control the level of recombination in the lamina propria and submucosae. Paradoxically, the PDGFRα^low^ fibroblast population showed a markedly higher recombination efficiency, with negligible recombination detected in the lower PDGFRα^high^ compartment (Extended Data Fig. 4j–m), potentially owing to different tamoxifen accessibility between stromal layers[32].

The distinctive recombination efficiency enabled us to trace PDGFRα^low^ fibroblasts at early tumour stages (6 weeks after DEN treatment) to explore their contribution to the niche-formation process (Extended Data Fig. 4n,o). We observed that the early tumour niche was composed of PDGFRα^low^-derived fibroblast clones that expanded locally in the upper stromal compartment. No clonal expansion events were found in the lower stroma (PDGFRα^high^ compartment) or spanning across stromal compartments (Extended Data Fig. 4o).

Overall, these observations demonstrate that local PDGFRα^low^ fibroblasts in the lamina propria not only maintain, but also contribute to, the formation of the pre-neoplastic tumour niche.

## Pro-fibrotic PDGFRα^low^ niche fibroblasts

To determine the mechanisms underlying the formation of the precancerous niche, we set out to study epithelial–mesenchymal communication. Single-cell RNA sequencing (scRNA-seq) was done in individually micro-dissected dysplastic tumours of the squamous upper gastrointestinal tract 8 months after DEN treatment (Fig. 3a,b, Extended Data Fig. 5a–f and Supplementary Tables 2 and 3).

The transcriptional profile of pre-neoplastic tumour cells was compared with that of cells from adjacent tumour-free areas in DEN-treated mice (internal control), and with that of cells from healthy untreated control animals (Extended Data Fig. 5g).

In line with our histological and lineage-tracing characterization (Fig. 2 and Extended Data Figs. 3c,d and 4), the scRNA-seq analysis revealed a marked degree of heterogeneity in the expression of the fibroblast marker *Pdgfra* (Fig. 3c). Distinctive *Pdgfra^low^* and *Pdgfra^high^* fibroblast populations were present across conditions (Supplementary Table 5). *Pdgfra^low^* fibroblasts expressed higher levels of genes encoding structural (scaffolding matrix) ECM components (such as *Fn1, Fbn1, Has1, Has2, Loxl2, Mfap5, Cd248* and *Col1a1*)[33] common in loose connective tissue. *Pdgfra^high^* fibroblasts instead showed an enrichment of genes encoding ECM fibrillar (underlying matrix) collagens (*Col6a3* and *Col5a3*), vascular support collagens (such as *Col4a1/2, Col8a1, Col15a1* and *Col13a1*) and other basement-membrane components[33] (such as *Lama1* and *Thbs1/2*; Fig. 3d, Extended Data Fig. 6a–c and Supplementary Table 6). Accordingly, immunolabelling of *Pdgfra*-derived clones (Extended Data Fig. 4j) in the lamina propria and submucosae, respectively, revealed increased fibronectin production in the *Pdgfra^low^* fibroblast compartment (Extended Data Fig. 6d).

We then investigated the tumour-specific transcriptional signature of the *Pdgfra^low^* niche-forming fibroblasts. Fibroblasts in cluster 19 (C19), enriched in pre-neoplastic tumour stroma (Extended Data Fig. 6e), showed a significant upregulation of matrisome genes associated with wound healing/fibrosis[9,34] (matrix deposition: *Fn1, Fbln5, Tnxb, Cd248, Vim, Plaur, Mfap5*; thrombospondins: *Thbs1, Thbs3*; collagens: *Col3a1, Col1a1/2, Col5a3*; remodelling: *Timp1, Adam9, Loxl1*; other factors: *Fgfr2, Bmp1, Bmp6, Cx3cr1*; Extended Data Fig. 6f and Supplementary Tables 7 and 8). The pro-fibrotic nature of these niche fibroblasts was supported by immunolabelling of both nascent and surviving tumours (Fig. 3e and Extended Data Fig. 6g), as well as by second-harmonic generation (SHG) imaging, which showed a marked ECM remodelling in the tumour niche (Extended Data Fig. 6h–j). Accordingly, alterations in integrin α6 (CD49f) expression in nascent epithelial tumour cells hinted at 'wound healing-like' changes in epithelial–ECM interactions[35] (Extended Data Fig. 6k). Together, these findings pointed at the activation of a tissue repair response in the stroma of early tumours, in line with the long-standing notion that tumours are "wounds that do not heal"[36].

We next explored whether niche-forming fibroblasts (C19) in long-term surviving tumours presented a CAF signature. Despite a subtle upregulation of a reduced subset of CAF-associated genes[12,28,37–39] (*Vim, S100a4, Mfap5* and *Col1a2*; Extended Data Fig. 6l–n), their expression at the protein level (VIM, FSP and FAP) remained largely unaltered or undetectable (Extended Data Fig. 6o). Accordingly,

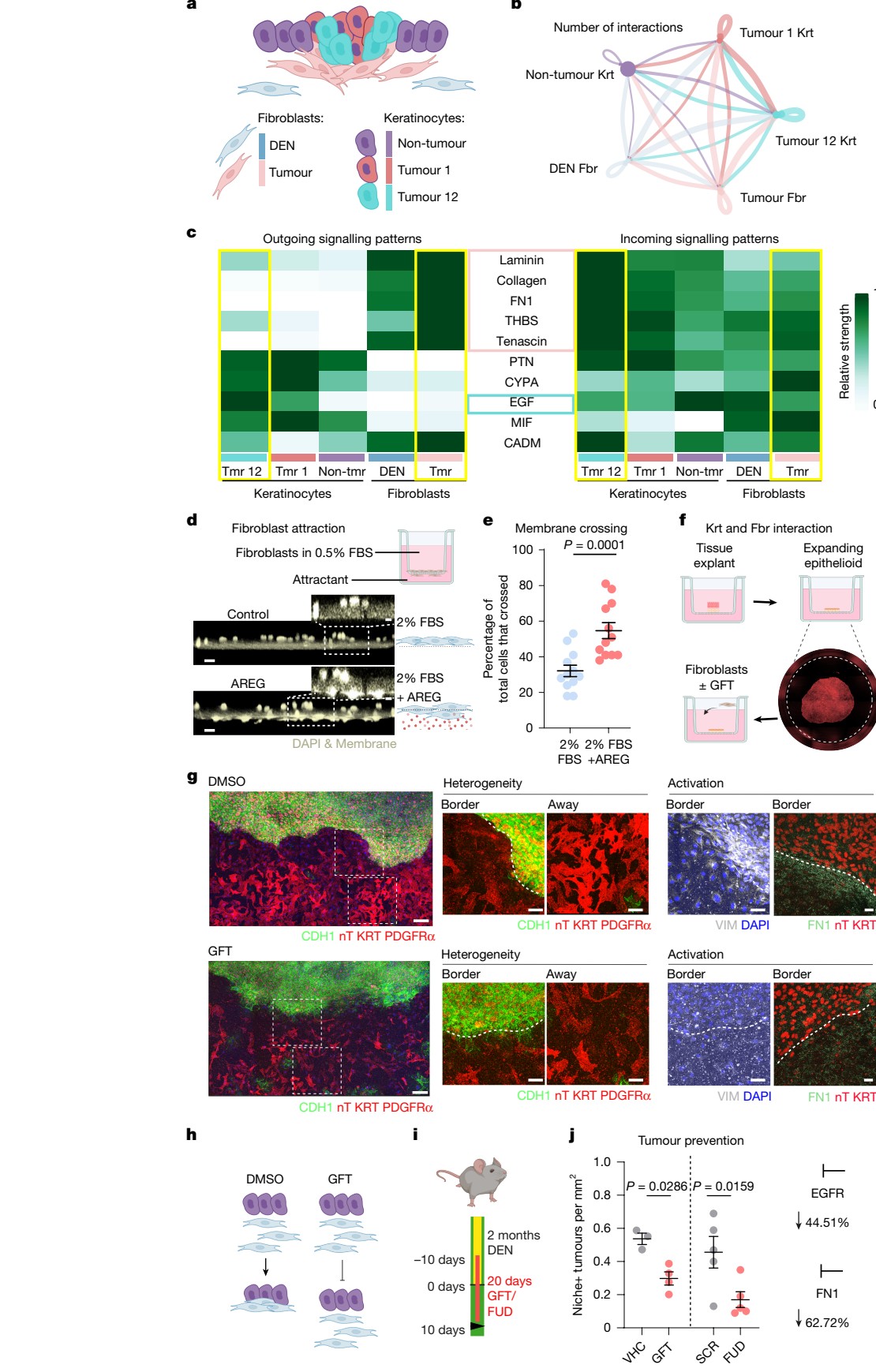

**Fig. 4** | See next page for caption.

**Fig. 4 | EGF–SOX9–FN1 axis promotes tumour survival. a**, Cartoon showing the composition of a tumour cell. **b**, The cell–cell communication network. Circles represent cells; circle sizes show cell numbers; and the thickness of the connections represents the number of significant interactions. **c**, Heatmap showing the top 10 signalling predictions. Interactions in Tumour (Tmr) 12 keratinocytes and Tmr fibroblasts are highlighted in yellow. Signals with the strongest interactions in Tmr fibroblasts are highlighted in pink; in Tmr 12, keratinocytes are highlighted in cyan. **d**, Top: schematics of the chemoattractant assay. Bottom: representative images. Scale bars, 25 μm; inset, 10 μm. FBS, fetal bovine serum. **e**, Percentage of fibroblasts crossing the membrane. Data are expressed as mean ± s.e.m. Dots represent replicate cultures from n = 4 mice. Significance assessed by one-way Welch's ANOVA with multiple comparisons (other groups in EDF 10c). **f**, Schematic representation of epithelioid and fibroblast co-culture treated with gefitinib (GFT) or vehicle (DMSO). **g**, Representative images from **f** showing fibroblast interaction with keratinocytes in DMSO and EGFR inhibition (GFT) conditions. Top insets show PDGFRα⁺ fibroblast (red) heterogeneity: PDGFRα^low fibroblasts assemble adjacent to keratinocytes, whereas PDGFRα^high fibroblasts position further away in controls. Fibroblast activation at the border is labelled by VIM (white) and FN1 (green). This was inhibited in GFT (bottom). Green, CDH1; nT, nuclear Tomato; KRT, keratinocytes, red; blue, DAPI. Scale bars: main, 500 μm; insets, 50 μm. **h**, Schematic representation of keratinocyte–fibroblast interactions under DMSO and GFT conditions. **i**, Experimental protocol of drug intervention with a fibronectin assembly-inhibiting peptide (FUD) or GFT 20-day regimen with the DEN treatment. **j**, Tumour burden decreased in the GFT and FUD groups compared with vehicle (VHC) or scrambled (SCR) control, respectively. Tissues were collected 10 days after DEN treatment. Data are from n = 3 (VHC), n = 4 (GFT), n = 5 (SCR) or n = 5 (FUD) mice. Data represent mean ± s.e.m. Significance was assessed by one-tailed Mann–Whitney test. Images were captured by confocal microscopy. Illustrations in **a**, **d**, **f**, **h** and **i** were created in BioRender; Alcolea, M. https://BioRender.com/eghet5p (2026).

fibroblast proliferation did not show significant changes[28] (Extended Data Fig. 6p). These observations indicate that Niche+ fibroblasts in surviving dysplastic tumours lack a fully established CAF phenotype. However, the presence of nuclear YAP and the profibrotic nature of Niche+ fibroblasts point to a pre-CAF state[30] (Extended Data Fig. 3e), transitioning to myCAFs at more advanced stages (invasive squamous cell carcinomas 14 months after treatment; Extended Data Fig. 6q). Label-transfer analysis from the fibroblast atlas in ref. 40 indicated that tumour niche fibroblasts probably derive from universal, rather than tissue-specific, fibroblast populations. *Pdgfra^low* fibroblasts, including tumour-enriched cluster 19, partly recapitulated the transcriptional signature of the *Pi16*⁺ universal population, whereas *Pdgfra^high* fibroblasts aligned with the *Col15a1*⁺ universal fibroblast subset (Extended Data Fig. 6r,s).

To assess whether stromal genetic alterations drive the tumour niche phenotype, we performed deep-targeted sequencing of 192 cancer-related genes[22] (Supplementary Table 9). The results argued against somatic mutations in fibroblasts being responsible for precancerous niche formation. The data revealed that DEN treatment induces gene perturbations, mainly in the epithelial compartment, showing a minimal mutational burden in the tumour stroma that matched the level of untreated or internal DEN control samples (Extended Data Fig. 7a,b).

In contrast to nascent stages (Fig. 2b,c), we found that, as tumours progressed, they showed a marked remodelling of the vascular network and an increased immune infiltrate (Extended Data Fig. 7c,d). In line with previous findings, transcriptional analysis revealed notable changes in the immune-cell composition in long-term surviving tumours, indicative of active immune-cell recruitment with progression towards an immunosuppressive microenvironment at later stages[41] (Extended Data Fig. 7e–h and Supplementary Table 10).

Taken together, our data reveal a significant stromal reorganization in surviving tumours. Analysis of lamina propria-derived *Pdgfra^low* fibroblasts was consistent with a notable fibrotic ECM remodelling in the precancerous niche, before the emergence of a fully established CAF phenotype.

## Nascent tumour heterogeneity

Next, we explored the epithelial transition from healthy and normal to pre-neoplastic states. Pseudotime analysis revealed two distinctive basal cell trajectories denoting tumour and non-tumour states. These trajectories largely converged in committed and differentiating cells (Fig. 3f and Extended Data Fig. 8a–f). Gene score enrichment analysis of tumour-specific gene modules identified by pseudotime analysis revealed further heterogeneity in epithelial tumour states, referred to as Tumour 1 and Tumour 12 (Fig. 3f, Extended Data Fig. 8d–h and Supplementary Tables 11–13), that expressed increased levels of the early tumour markers *Krt6a* and *Krt17* (refs. 1,21) (Extended Data Fig. 8i).

Gene set enrichment analysis revealed a unique signature in Tumour 12 cells (Extended Data Fig. 8j,k), enriched for genes related to cancer-associated processes such as tissue development and morphogenesis (*Cdh1*, *Cdh4*, *Sox4*, *Klf4*, *Sox9* and *Foxa1*), hypoxia (*Hmox1*, *Vegfa*, *Edn1*, *Cited2* and *Sirt1*) and stress-induced pathways, including EGFR (*Areg*, *Hbegf*, *Nrg1*, *Nrg2*, *Egfr*, *Epha2* (ref. 42) and *Lamc2* (ref. 43)), Hippo, TNF, p53 and TGF-β (*Tead1*, *Tnfaip3*, *Nfkbia*, *Ccng2*, *Rela*, *Nfkb1*, *Ptgs2*, *Gadd45a*, *Cdkn1a* and *Id1-4*) (Extended Data Fig. 8k and Supplementary Table 14). This was reinforced by increased levels of genes encoding transcription factors associated with a tumour stress response (*Jun*, *Fos*, *Fosb*, *Runx1*, A*tf3*, *Egr1*, *Egr3* and *Myc*)[41,44–48] (Extended Data Fig. 8l). Crucially, validation at the protein level further supported the heterogenous nature of the epithelial cells populating nascent early tumours, with Tumour 12 markers staining only a subset of tumour cells (Fig. 3g and Extended Data Fig. 8m,n).

Further expression changes in Tumour 12 comprised the upregulation of genes associated with stromal communication, including cell adhesion (*Col12a1*, *Itgav*, *Itga2*, *Itgb6*, *Lama3*, *Vcl*, *Cadm1*, *Icam1* and *Runx1* (ref. 49)), as well as ECM breakdown (*Adamst1* (ref. 50)). Increased expression of the cell-adhesion genes *Ccn1*, *Ccn2* (ref. 51) and *Thbs1* (ref. 52 was of particular interest, owing to their recognized role in communication with fibroblasts (Extended Data Fig. 8l).

Since the data so far had indicated that there was close communication between Tumour 12 epithelial and stromal cells, we reasoned that Tumour 12 cells could be linked to the emergence of the tumour niche, and thereby to tumour persistence and survival. Indeed, SOX9 expression, used as a proxy to mark the Tumour 12 state (alongside KRT6A and/or KRT17 (refs. 1,21)), was expressed mainly in early tumour cells in direct contact with the niche (Extended Data Fig. 8o), indicative of their close interaction. Accordingly, nascent Niche+ tumours showed significantly higher SOX9 expression than Niche− tumours did (Fig. 3h and Extended Data Fig. 9a–c).

To explore whether SOX9⁺ cells were associated with the formation of the pre-neoplastic tumour niche, we took advantage of sporadic clusters of KRT6A⁺ and SOX9⁺ cells in phenotypically normal (non-tumour) regions of DEN-treated tissue, potentially marking prospective tumour cells before lesion formation. Isolated SOX9⁺ cell clusters showed signs of fibroblast attraction, presenting fibroblasts in closer proximity and at a higher density than in the surrounding tissue (Fig. 3i and Extended Data Fig. 9d–f). Overall, these data establish the Tumour 12 state as a relevant player in early tumour stromal remodelling and niche formation. Accordingly, SOX9 depletion in *Krt14-Cre^ER*; *Sox9^flox/flox* DEN-treated mice (Fig. 3j and Extended Data Fig. 9g) led not only to a significant reduction in early tumour survival (1 month after DEN treatment; Fig. 3j) but also to a reduction in the size of nascent Niche+ tumours compared with that of Niche− tumours (Extended Data Fig. 9h).

## Cell–cell communication in early tumours

To study epithelial–mesenchymal communication in surviving tumours, we assessed ligand–receptor interactions[53] enriched across coexisting cell populations in DEN-treated tissue (Fig. 4a,b and Extended Data Fig. 10a; fibroblasts: Fig. 3c and Extended Data Fig. 6e; epithelia: Fig. 3f and Extended Data Fig. 8g–k). We analysed interactions between non-tumour cells, including control keratinocytes (non-tumour Krt) and control fibroblasts (*Pdgfra^low*; C19 DEN Fbr used as an internal control), as well as among tumour associated cells; that is, tumour keratinocytes (Tumour 1), tumour niche keratinocytes (Tumour 12) and tumour niche fibroblasts (*Pdgfra^low*; C19 Tumour Fbr). Pro-fibrotic ECM-related pathways were among the top outgoing interactions predicted to preferentially signal from tumour niche fibroblasts to tumour niche keratinocytes. These pathways included laminin, collagen, fibronectin, thrombospondin and tenascin (Fig. 4c). Given its well-established role in ECM assembly and association with fibrosis and advanced cancer progression[54], we reasoned that fibronectin (FN1) might be a central player modulating ECM interactions across tumour cell compartments. The relevance of FN1 interactions was supported both by the specific upregulation of FN1 receptor genes in tumour niche keratinocytes (receiver cells) and by the increased expression of FN1, at both mRNA and protein level, in tumour niche fibroblasts (sender cells; Fig. 3d,e and Extended Data Figs. 6f,g and 10b). Overall, our analyses predicted robust pro-fibrotic and wound healing epithelial–mesenchymal interactions in surviving tumours.

Next, we explored outgoing signals from tumour niche keratinocytes. Here, EGF was identified as one of the strongest incoming signals for tumour niche fibroblasts (Fig. 4c). EGF ligands (including AREG and HBEGF) were enriched in tumour niche keratinocytes (sender cells) at both the mRNA and protein levels (Fig. 3g and Extended Data Figs. 8l, m and 10b), with its receptor (EGFR) being markedly expressed in the underlying tumour niche fibroblast population (receiver cells; Extended Data Fig. 10b).

The upregulation of both AREG (amphiregulin) and FN1 in nascent tumours (10 days after DEN treatment; Fig. 3e,g and Extended Data Figs. 6g and 8m) highlighted the importance of epithelial–mesenchymal communication from nascent tumour stages.

## EGF–SOX9–FN1 supports tumour survival

Given the well-known role of EGF and FN1 signalling in epithelial–stromal communication during wound healing and tissue damage[55,56], we reasoned that they might have a similar role in response to DEN-induced genetic stress.

To determine whether tumour niche keratinocytes (Tumour 12) exert their mesenchymal remodelling effect (Fig. 3h,i and Extended Data Figs. 8k,o and 9a–f) through the EGFR pathway (Fig. 4c), we used a chemoattractant assay (Fig. 4d). We found that the EGFR ligand AREG positively stimulated fibroblast migration, confirming that keratinocytes in nascent tumours can promote fibroblast chemotaxis and mesenchymal remodelling through paracrine EGF stimulation (Fig. 4d,e and Extended Data Fig. 10c).

We gained further insights into the dynamic nature of epithelial–mesenchymal communication by coculturing regenerative 3D oesophageal cultures (epithelioids[57]) with primary fibroblasts (Fig. 4f). We found that expanding keratinocytes, which exhibit increased levels of stress markers (including SOX9; Extended Data Fig. 10d), prompted fibroblasts to segregate spatially into two distinct populations, mirroring the in vivo scenario. PDGFRα^low fibroblasts localized immediately adjacent to the growing epithelium, whereas PDGFRα^high fibroblasts were found in distant areas (Fig. 4g and Extended Data Fig. 10e). The interaction between expanding epithelial cells and fibroblasts also led to FN1 deposition and vimentin upregulation

in the PDGFRα^low population (Fig. 4g and Extended Data Fig. 10e). Gefitinib-mediated inhibition of EGFR signalling in epithelioids further confirmed the role of EGFR signalling in epithelial–mesenchymal communication in a regenerative or stress context, showing reduced epithelial SOX9 expression, diminished fibroblast segregation or compartmentalization, and hindered fibroblast ECM remodelling (Fig. 4f–h and Extended Data Fig. 10d,e). These data directly link EGFR signalling and SOX9 expression in the epithelium with mesenchymal remodelling.

FN1, which is an important component of the fibrotic niche in early tumours (Fig. 3d, e and Extended Data Fig. 6b–g), also represents a critical and well-established regulator of the stromal wound-healing response[58]. To determine whether the newly formed fibronectin-rich tumour niche has a critical role in promoting early tumour growth and survival, we treated established 3D epithelioids (exhibiting steady-state levels of proliferation)[57] with soluble FN1 for 24 hours. Indeed, FN1 promoted epithelial proliferation (Extended Data Fig. 10f). By contrast, in vivo bleomycin treatment, which promotes tissue fibrosis, resulted in a negligible increase in SOX9 expression in DEN-untreated mice (Extended Data Fig. 10g). These data indicate that, although fibrosis promotes tumour growth, a fibrotic environment alone is not sufficient to drive a pre-neoplastic response.

These ex vivo experiments revealed that the EGF–SOX9–FN1 axis governs epithelial–mesenchymal communication and subsequent tissue reorganization in response to epithelial perturbations. Accordingly, in vivo experiments showed that inhibition of either fibronectin fibrillogenesis (using the functional upstream domain (FUD) peptide)[59] or EGFR signalling with Gefitinib (GFT) led to a significant reduction in the number of Niche+ tumours (Fig. 4i,j).

Taken together, our results demonstrate the central role of the EGF–SOX9–FN1 axis in early tumour niche formation. In response to genetic stress, SOX9⁺ epithelial cells stimulate fibroblast migration and ECM remodelling through EGF signalling. This, in turn, promotes the formation of a pro-fibrotic, fibronectin-rich tumour scaffold, which favours tumour persistence and progression by perpetuating the pro-tumorigenic phenotype.

## Human tumour niche remodelling

To validate the relevance of our observations in a human context, we analysed chemo-naive early-stage human oesophageal squamous cell carcinomas (T1a, T1b and Tis; Fig. 5a–d) and residual dysplastic tissues after chemotherapy (ypT0; Fig. 5e,f).

Immunolabelling showed that patient tumours, unlike adjacent tissue, recapitulate observations in the DEN mouse model. T1a and T1b tumours displayed homogeneous expression of KRT6A and KRT17 (pan-tumour markers in mice) and heterogeneous expression of SOX9 (marking tumour cells associated with stromal remodelling in mice; Tumour 12; Fig. 5b,c). Accordingly, SOX9-expressing cells exhibited high AREG expression levels (Fig. 5b) and were found next to areas with increased fibroblast density (PDGFRα; Fig. 5c). Analysis of tissue whole-mounts reinforced the link between SOX9 expression and stromal remodelling, with marked FN1 deposition in the vicinity of SOX9⁺ tumour cells (Fig. 5d).

These features were also observed in residual dysplastic tissue after chemotherapy (Fig. 5e,f), which showed marked ECM remodelling (fibronectin deposition) in the proximity of AREG⁺ or SOX9⁺ tumour cells.

Our observations support the presence of a heterogeneous AREG⁺ and/or SOX9⁺ population in early-stage squamous tumours of the human oesophagus. The data further reinforce the association between this population, mesenchymal changes and ECM remodelling in early human oesophageal tumorigenesis, revealing the potential clinical relevance of our study.

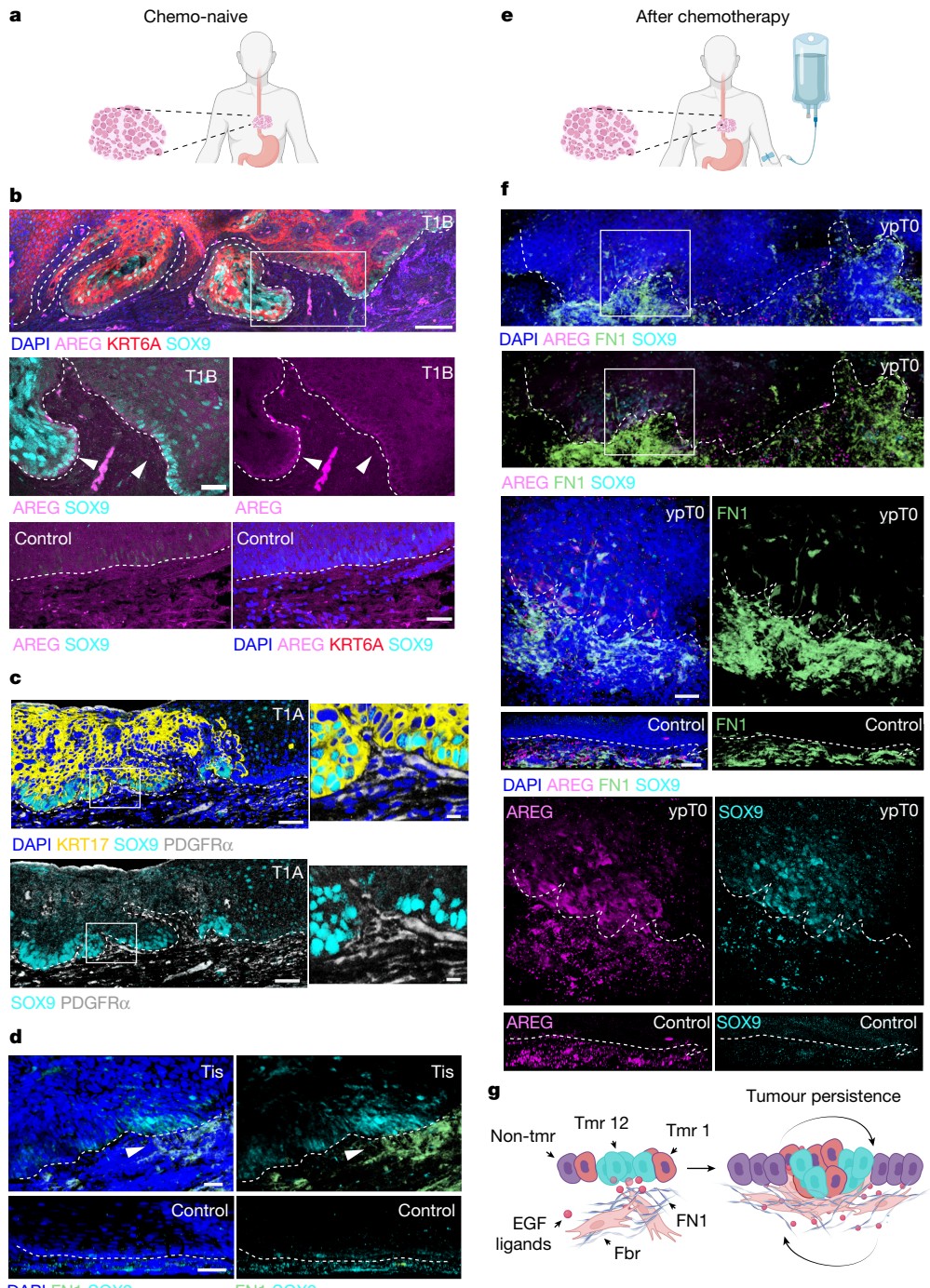

**Fig. 5 | Early tumour niche remodelling in human oesophageal squamous cell carcinoma. a**, Schematic representation of the tumour sample origin shown in **b**–**d**; tumour resections from human patients performed before chemotherapy (chemo-naive). **b**, Representative confocal image (from $n = 4$ patients) of a T1B tumour section showing widespread KRT6A (red), and heterogenous SOX9 (cyan) and AREG (magenta) expression; blue, DAPI. The control sample is normal area identified by a pathologist in the tumour section. Scale bar, 200 µm; in insets, 50 µm. **c**, Representative confocal image (from $n = 5$ patients) of a T1A tumour section showing widespread KRT17 (yellow), heterogenous SOX9 (cyan) expression and fibroblast attraction to areas marked by PDGFRα (grey) expression. Blue, DAPI. Scale bars, 200 µm; in insets, 10 µm. **d**, Representative confocal image (from $n = 1$ patient) of a carcinoma in situ (Tis) whole mount showing heterogenous SOX9 (cyan) expression and fibronectin (FN1, green) accumulation underneath (white arrowhead). Blue, DAPI. Scale bars, 50 µm. **e**, Schematic representation of the tumour

sample origin shown in **f**; resections were performed after chemotherapy. **f**, Representative confocal image (from $n = 1$ patient) of a post-treatment, pathological staging T0 (dysplasia) whole mount, showing heterogenous expression of SOX9 (cyan) and AREG (magenta) and fibronectin (FN1, green) accumulation underneath. Blue, DAPI. The control sample is a biopsy distal to the tumour. Scale bar, 500 µm; in insets, 50 µm. **g**, Schematic of proposed model whereby epithelial cells respond to genetic perturbation by activating a stress gene signature (tumour 12), denoted by SOX9 and EFG ligand overexpression. This drives the migration of underlying fibroblasts towards the nascent tumour, followed by the formation of a fibrotic scaffold. The establishment of this EGF–SOX9–FN1 signalling axis between epithelial and mesenchymal nascent tumour cells results in the formation of an early tumour niche that favours tumour growth and promotes long-term persistence. Illustrations in **a**, **e** and **g** were created in BioRender; Alcolea, M. https://BioRender.com/e94fdb9 (2026).

## Discussion

Studies in the past decade have shown that mutations, conventionally thought to be the sole cause of cancer, can also be found in healthy ageing tissues[4–8], where they form part of normal tissue physiology. This has redirected the interest of the cancer community to fill the knowledge gap around the earliest disease stages, and particularly to understand how mutant cells interact with adjacent tissue compartments[1–3,13,21,22]. This study provides mechanistic insights into the processes that determine whether tumours emerging in complex mutant landscapes persist long term or are outcompeted and eliminated from the tissue[1]. We show that early tumour survival and subsequent progression rely on intricate interactions between nascent tumour cells and their dynamic niche.

This work reveals that exposure to mutagens activates a heterogeneous 'tissue stress' response, whereby incipient epithelial and mesenchymal tumour cells signal and feedback onto each other through the EGF–SOX9–FN1 axis. Tumours failing to activate this communication axis are less likely to persist and grow. In particular, a tumour-specific stress state, defined by high SOX9 expression, promotes the recruitment of fibroblasts to the nascent tumour through EGF signalling. This in turn facilitates the formation of a precancerous niche, rich in fibronectin, that perpetuates a pro-tumorigenic phenotype, favouring tumour growth and persistence. Interfering with fibronectin fibrillogenesis in vivo impaired niche formation, prevented tumour survival and reduced the overall tumour burden. These findings support a self-sustaining process in which the reciprocal communication between niche mesenchymal cells and SOX9[high] epithelial cells supports tumour survival, favouring disease progression over time (Fig. 5g).

These data are relevant to early human carcinogenesis. The heterogeneous expression of SOX9, EGFR ligands and associated deposition of FN1 are recapitulated in early stage human oesophageal squamous cell carcinomas, consistent with active epithelial–stromal communication in nascent human tumours. Whether interfering with ECM assembly represents a valid approach to prevent cancer progression in patients, and whether analogous mechanisms operate in other tumour types, requires further investigation.

Overall, our data demonstrate the unprecedented capacity of the early tumour niche to perpetuate tumour survival signals beyond intrinsic changes driven by genetic alterations, enabling nascent tumours to persist in highly competitive mutant landscapes. This offers the new perspective that not only mutations, but also the environmental response to genetic stress, defines the likelihood of tumours to progress towards more advanced disease stages. Our findings indicate that strategies targeting tumour cells, as well as supporting neighbouring cells, could open new avenues for cancer prevention and improve long-term outcomes.

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

## Methods

### Clinical samples

High-grade dysplasia, squamous cell carcinoma and macroscopically normal, healthy clinical samples, as well as the corresponding clinical information, were collected following research ethics approval and individual informed consent from patients who underwent oesophagectomy for oesophageal cancer.

The T1A and T1B stage chemo-naive surgical tumour samples were donated by patients who had undergone surgery at the Clinic for Visceral, Thoracic and Vascular Surgery at TU Dresden or at the Medical Department I of the Carl Gustav Carus University Hospital. Macroscopically normal samples adjacent to the proximal resection margin were sampled from cancer resection specimens. The corresponding formalin-fixed, paraffin-embedded material (tumour and healthy tissue) from a total of ten characterized oesophageal squamous cell carcinomas was selected from the archive of the Institute of Pathology of the University Hospital Carl Gustav Carus (EK 59032007) by the Tumour and Normal Tissue Bank (TNTB) Dresden. Studies presented in the manuscript involving early chemo-naive human oesophageal tumour samples from Dresden were approved by the Ethics Committee of TU Dresden, Germany (ref. SR+BO-ff (Mono)-EK-161042025). Studies presenting chemo-naive or post-chemo human oesophageal tumour samples from Guy's and St Thomas' (London) and Addenbrooke's Hospital (Cambridge, UK), respectively, were approved by the East of Scotland Research Ethics approval committee (REC 18/ES/133). Histological sectioning of the tissue samples and haematoxylin and eosin staining of reference slide series for determining the tumour cell content of the individual patient samples were done at the Institute of Pathology, University Hospital CGC Dresden, TU Dresden.

### Mice strains

All animal experiments were approved by the local ethical review committees of the University of Cambridge and conducted according to the Home Office project licences PPL70/8866 and PP7037913 of the Cambridge Stem Cell Institute, University of Cambridge.

Unless otherwise specified, C57BL/6J mice (Charles River, strain code 632) were used. Other mouse strains used include: cell-cycle reporter line $R26^{Fucci2aR}$ ($Fucci2a$)[60], provided by I. J. Jackson; $Pdgfra^{EGFP}$ (007669, Jackson Laboratory); $Sox9^{flox/flox}$ (ref. 61; MRC-Harwell, on behalf of the European Mouse Mutant Archive; https://www.infrafrontier.eu); $K14^{CreER}$ (005107, Jackson laboratory); $R26^{mT-mG}$ ($mTmG$, Jackson Laboratory); $Col1a2^{CreER}$ (029567, Jackson Laboratory); $R26^{FlConfetti}$ (ref. 62; 017492, Jackson laboratory, provided by H. Clevers)[62]; $R26^{nT-nG}$ ($nTnG$; 023537, Jackson Laboratory); $H2B$-$EGFP$ ($CAG::H2B$-$EGFP$; 006069, Jackson Laboratory); NOD-SCID-γ (NOD.Cg-$Prkdc^{scid}$ $Il2rg^{tm1Wjl}$/SzJ; 005557, Jackson Laboratory); and $Pdgfra^{CreER}$ (018280, Jackson Laboratory)[63]. Further information about the experimental mouse lines can be found in the Supplementary Methods section Experimental mouse lines.

Recombination of $Col1a2^{CreER}R26^{FlConfetti/WT}$ mice was induced by a single intraperitoneal tamoxifen injection (3 mg per 20 g body weight). The $Col1a2^{CreER}R26^{FlConfetti/WT}$ mice were induced by a single intraperitoneal tamoxifen injection (0.5 mg or 5.0 mg per 20 g body weight). The $K14^{CreER}Sox9^{flox/flox}$ received two subcutaneous tamoxifen injections (5 mg per 20 g body weight) 48 h apart. Tamoxifen was prepared by dissolving in ethanol (less than 10% total volume) and diluting in sunflower-seed oil.

All strains were maintained in a C57BL/6 background. All experiments used a mixture of male and female mice with no gender-specific differences observed (unless specified otherwise). For RNA-sequencing experiments, only male animals were used to avoid confounding effects from the oestrous cycle. All animals exposed to the carcinogen and their respective controls were adults between 8 and 14 weeks of age (see the section on chemically induced mutagenesis below). Mice were bred and maintained under specific-pathogen-free conditions at the Gurdon Institute and the Anne McLaren Building, University of Cambridge. All animals were housed at 20–24 °C, 45–65% humidity and a 12 h:12 h light:dark cycle.

### Chemical tumorigenesis model

Mice were treated with DEN (Sigma-Aldrich; N0756) at 40 mg l$^{-1}$ in Ribena-flavoured water for 24 h, three times a week (Monday, Wednesday and Friday) for 8 weeks[1,21]. Mice received sweetened water between DEN dosages and normal water after the completion of DEN treatment. Control mice received sweetened water as a vehicle control for the length of the treatment. Animals exposed to DEN were monitored for adverse effects as stated in our Home Office project licences (PPL70/8866 and PP7037913) for regulated procedures on protected animals. In summary, animals were weighed daily on weekdays for the first week, weekly for the next month and then monthly thereafter. Animals were also checked every day for any clinical signs or abnormal behaviour. Any concerning animals were weighed every other day or daily, if necessary, until the weight was stable again. If the weight loss approached 10%, animals were weighed daily until stable and received wet mash or palatable diet. Animals showing 15% weight loss measured for 2 consecutive days were killed immediately.

### EdU tracing

For EdU labelling experiments, mice received 100 μg EdU in PBS (Life Technologies, A10044) intraperitoneally 2 h before tissue collection. In vitro 3D cultures (see above) received media supplemented with 10 μM EdU and were incubated for 2 h at 37 °C and 5% $CO_2$ before fixation. EdU incorporation in tissue whole-mounts (see above) was detected using a Click-iT EdU kit according to the manufacturer's instructions (Invitrogen, C10337). EdU$^+$ cells were quantified using confocal microscopy.

### Inhibitor treatment in vivo

Mice were treated with Gefitinib for 20 days at 80 mg per kg body weight (or vehicle control) three times a week to inhibit the EGFR pathway. Treatment started 10 days before the end of DEN treatment and ended 10 days after it. Gefitinib was prepared in concentrated form by dissolving it in DMSO and was diluted in corn oil.

Pharmacological inhibition of FN1 fibrillogenesis was achieved by treating mice with functional upstream domain (FUD) peptide[64] intraperitoneally for 20 days at a concentration of 12.5 mg per kg body weight. Control mice were treated with scrambled (SCR) control peptide. Treatment started 10 days before the end of DEN treatment and ended 10 days after it. Peptides were synthesized at more than 95% purity (WatsonBio; peptide sequence below). Lyophilized peptides were reconstituted in PBS.

The peptide sequences were:

FUD, - Cys-GSKDQSPLAGESGETEYITEVYGNQQNPVDIDKKLPNETG FSGNMVETEDTKLN;

SCR, - Cys-QGQTGPVNSKVKIDNYELESNPEKIEANDLQVEGTTTYESKF MGDLTGSGNPED.

### Whole-mount preparation

The upper gastrointestinal tract (oesophagus and forestomach) from control and DEN-treated mice was dissected at the time points indicated in the main text and/or figure legends. Oesophagi were excised and cut open longitudinally. The muscle layer was then removed and the tissue was flattened under a dissecting microscope using fine forceps. Stomachs were cut open longitudinally and rinsed twice with PBS to remove any food remains. The glandular stomach was excised away and the forestomach kept and flattened for downstream analysis. For epithelial-only and stromal-only whole-mounts, tissues were incubated in 5 mM ethylene-diamine-tetraacetic acid (EDTA) (Life Technologies, 15575020) in PBS for 3 h at 37 °C. After incubation, the epithelium was

gently peeled from the stroma using fine forceps. Subsequently, each of these layers was flattened individually.

Oesophageal and forestomach whole-mounts (either peeled or unpeeled) were fixed in 4% paraformaldehyde (Alfa Aesar, 043368) in PBS for 30 min at room temperature.

## Histology

For histology, tissues were fixed in 10% formalin in PBS overnight at room temperature before storage at 4 °C in absolute ethanol. Haematoxylin and eosin staining was done in 7-μm paraffin-embedded sections by the Histology Core Service at the Cambridge Stem Cell Institute and imaged using a Zeiss AxioScan Z1 microscope. Histological analysis of murine tumour samples was done by B. Mahler-Araujo at the MRC Metabolic Diseases Unit (MC_UU_00014/5).

## Ex vivo tissue recombination assay

Under a dissecting microscope in a laminar flow hood, oesophagi were dissected and epithelial–stromal layers isolated as described above in the section 'Whole-mount preparation'. Thereafter, tissues were rinsed in 1% P/S in PBS three times to remove residual EDTA and flattened. Combinations of epithelium and stroma from different experimental conditions (DEN treated and/or control) were prepared by carefully placing the epithelial layer over the relevant stroma (referred to as 'tissue recombination' composites). The remaining epithelium was trimmed to match the size of the stroma, and the resulting construct was cut in half. Flattened epithelium–stromal constructs were cultured in six-well plate inserts (ThinCert Greiner Bio-One, 657641). Size-matched polydimethylsiloxane (PDMS) stencil frames were placed around the tissue construct to prevent cell expansion (see the 'Stencil production' section below). The tissue was allowed to settle for 10 min before adding 2 ml of minimal medium (mFAD) containing one-part DMEM (Fisher Scientific, 41966029) and one-part DMEM/F12 (Fisher Scientific, 11320033) supplemented with 5 μg ml$^{-1}$ insulin (Sigma-Aldrich, 15500), 5% fetal calf serum (Fisher Scientific, 26140079), 1% P/S and 5 μg ml$^{-1}$ Apo-Transferrin (Sigma-Aldrich, T2036), as previously described[24,65]. The 3D heterotypic cultures were maintained in standard humidified cell-culture incubators at 37 °C with 5% CO$_2$ for up to 7 days. At the end point, samples were fixed in 4% PFA in PBS for 30 min at room temperature and stored for downstream confocal analysis.

## Stencil production

Silicone elastomer (PDMS) was mixed with a curing agent (Avantor VWR; Sylgard 184 Elastomer Kit, 634165S) at a 10:1 ratio and centrifuged at 300g for 10 min to remove the bubbles. The resulting mix was poured on a dish at around 70 mg cm$^{-2}$ and left on an even surface to polymerize overnight at 37 °C. The next day, the resulting polymer was cut into 2 × 5 mm rectangle-shaped frames, sterilized in 70% ethanol overnight, and treated with 1% pluronic acid (Sigma-Aldrich, P2443-250g) in PBS for 1 h at 37 °C. The frames were then left to air dry before use.

## In vivo tissue recombination grafting

Tissue recombination composites (as described above in the section 'Ex vivo tissue recombination assay') of DEN-treated oesophageal stroma and untreated (control) oesophageal epithelium were prepared for in vivo grafting adapting the strategy described above. Before separating the epithelium from the stroma, all visible tumours were marked with a partial incision using a punch biopsy tool (1 mm diameter; Merck, WHAWB100040). After separating the tissue layers, all the stromal compartments were assessed for peeling efficiency under a fluorescence dissecting microscope (Leica M165 FC), and any remaining epithelium, identified by the dense epithelial nuclei clusters, were excised from the tissue using a 1 mm biopsy punch. For heterotypic tissue constructs, 2 mm biopsies (Selles Medical, instrument BP20F) of tumour or control stroma were excised and a 2 mm healthy untreated epithelium biopsy placed above. Composites were cultured overnight as described above

and grafted in the back skin of anaesthetized shaved NSG female mice (two constructs per incision, and two incisions per animal). Longitudinal incisions for grafting were approximately 5 mm in length. The wounds were closed with GLUture glue (Fisher Scientific, NC0632797) and the mice were left to recover. Then, 3–6 months later, the mice were killed and the back skin fixed with 4% PFA in PBS for 30 min at room temperature and stored for downstream confocal analysis.

## Primary mouse fibroblast isolation and migration assay

Oesophagi were dissected as described above and cut in half. Tissue was incubated in 0.5 mg ml$^{-1}$ Dispase (Sigma-Aldrich, D4818) for 10 min at 37 °C while rotating. After incubation, the epithelium was peeled away and the stroma was minced finely and incubated in Trypsin-EDTA (0.25%) (ThermoFisher, 25200056) for 15 min at 37 °C while rotating. The resulting suspension was mixed by pipetting and DMEM supplemented with 10% FBS and 1% P/S was added (1:1 v/v). The suspension was passed through a 70 μm filter (PluriSelect, 43-10070-40) and cells were pelleted by centrifugation at 300g for 5 min at 4 °C. Pellets were resuspended in 0.5% FBS, 1% P/S in DMEM, and seeded on 8.0 μm pore transwell insert (24-well plates; ThinCert Greiner Bio-One, 662638). The lower compartment of the transwell contained 1% P/S DMEM supplemented either with 0.5% FBS, 2% FBS, 10% FBS or 2% FBS with 1 μg ml$^{-1}$ Amphiregulin (AREG) (R&D, 989-AR-100/CF). Primary fibroblasts were cultured for 48 h before fixation in 4% PFA in PBS for 10 min. Membranes were incubated with 1 μg ml$^{-1}$ DAPI (Sigma-Aldrich, D9542) in PBS for 30 min at room temperature, cut and mounted in 1.52 Rapiclear mounting medium (SUNJin Lab, RC152001) keeping their original orientation, followed by confocal analysis. Further information on quantification can be found in the Supplementary Methods section 'Analysis of fibroblast migration assay'.

## Keratinocyte cultures and fibronectin treatment in vitro

Oesophagi were cultured using the 3D epithelioid organ culture approach[65]. In brief, tissues were dissected, cut into 3 × 5 mm rectangles and placed on a transwell insert with the epithelium side up. The tissue was left to settle for about 5 min. Explants were expanded in complete medium (cFAD) containing mFAD supplemented with 1 × 10$^{-10}$ M cholera toxin (Sigma-Aldrich, C8052), 10 ng ml$^{-1}$ EGF (Fisher Scientific Pepro-Tech, AF-100-15), 0.5 μg ml$^{-1}$ hydrocortisone (Calbiochem, 386698). Tissue explants were removed by aspiration 5 days after culture set-up and maintained in mFAD for 2 weeks to confluence. Soluble fibronectin (Fisher Scientific Corning, 356008) was added to the medium for 24 h at 100 μg ml$^{-1}$ after diluting it in mFAD with 25 mM HEPES (Fisher Scientific, 15630056). Samples were fixed with 4% PFA in PBS for 30 min at room temperature, after a 2 h EdU chase, and kept for downstream confocal analysis.

## Keratinocyte and fibroblast interaction assay

Epithelioids were set up as described above and maintained in mFAD until keratinocyte migration started. The original tissue was then removed and the explant left overnight before adding freshly isolated oesophageal fibroblasts (as described above). DMSO as vehicle control or 2 μM of Gefitinib prepared in DMSO were added together with the fibroblast suspension. Then, 3 days after the fibroblasts were introduced to culture, samples were fixed with 4% PFA in PBS for 30 min at room temperature and kept for downstream confocal analysis.

## Immunostaining

After fixation, epithelial–stromal composites or human tissue wholemounts were incubated for 30 min in permeabilization buffer (PB1; 0.5% bovine serum albumin (VWR International, 126575-10), 0.25% fish-skin gelatin (Sigma-Aldrich, G7765), 1% Triton X-100 (Fisher Scientific, 10102913) in PBS), then blocked for 2 h in PB1 containing 10% donkey serum (DS) (Scientific Laboratory Supplies, D9663). Next, tissues were incubated with primary antibodies diluted in 10% DS in PB1

for 3 days at 4 °C followed by four washes of 30 min each with 0.2% Tween-20 (Promega UK, H5151) in PBS. Thereafter, tissues were incubated overnight with secondary antibodies diluted 1:500 in 10% DS in PB1 at room temperature. Unbound antibody was removed by four washes with 0.2% Tween-20 in PBS throughout the next day. Antibody details are provided in Supplementary Table 15. To stain cell nuclei, tissues were incubated with 1 µg ml⁻¹ DAPI in PBS at 4 °C overnight. Afterwards, samples were rinsed three times in PBS and mounted in 1.52 RapiClear mounting media for imaging.

Immunolabelling of individual tissue layers (epithelium or stroma) or sections consisted of an incubation for 30 min in permeabilization buffer (PB2; 0.5% bovine serum albumin, 0.25% fish-skin gelatin, 0.5% Triton X-100 in PBS). Tissues were then blocked for 2 h in PB2 containing 10% DS. Next, samples were incubated overnight at room temperature with primary antibodies diluted in 10% DS in PB2 followed by three washes with 0.2% Tween-20 in PBS for 30 min each. Secondary antibodies were diluted 1:500 in 10% DS in PB2 and incubated with tissues overnight at 4 °C, after which unbound antibody was removed by three washes with 0.2% Tween-20 in PBS, and staining continued as above. Thick cryosections of fixed tissues embedded in optimal cutting temperature compound (OCT; Thermo Scientific, 12678646), cut with a thickness of 50 µm onto glass slides, were immunolabelled using the same protocol. Likewise, 7-µm paraffin-embedded sections were immunolabelled according to the protocol described above, after antigen retrieval performed by heating of tissue sections in either 1 mM EDTA buffer (pH 8.0) or 10 mM sodium citrate buffer (pH 6.0) for 10 min at 95 °C.

When staining with primary antibodies raised in the same host, one of the antibodies was acquired as preconjugated with a fluorophore or conjugated in house following the manufacturer's instructions (Invitrogen, A20186/A20187). The staining proceeded as described above with the unconjugated primary antibodies. After incubation with the corresponding secondary antibodies, the samples were blocked for 3 h at room temperature with 10% DS in PB with the IgG from the relevant host species (1:500). Afterwards, samples were incubated with conjugated antibodies diluted in PB containing 10% DS and the relevant host IgG (1:500) overnight at room temperature. At this point, staining proceeded as described above. Immunostained samples were analysed by confocal imaging.

## Confocal imaging

Confocal images were acquired using either an inverted Leica SP5 microscope with standard laser configuration or a Stellaris 8 FALCON FLIM microscope with a white-light laser using LAS X 4.7.0.28176 or 3.5.5.19976 software. Typical confocal settings used included: bidirectional scanning, a 40× immersion objective lens, an optimal pinhole size (as defined by the software), a scan speed of 400–600 Hz with 2–3× line averaging, optimal Z-step size (as defined by the software) and a resolution of 512 × 512 or 1,024 × 1,024 pixels, unless stated otherwise. Then, sD reconstructions from optical sections and their corresponding image renders were generated using Volocity 5.5.5 (PerkinElmer) and Volocity 7 (Quorum), Zen 3.2 and Arivis 3.5.1. Further information about specific types of image analysis, such as second-harmonic generation imaging, can be found in the Supplementary Methods.

## Transcriptomics

**Library preparation and scRNA-seq.** Sample preparation methods for libraries can be found in the Supplementary Methods in the section 'Single-cell and RNA isolation for single-cell RNA-sequencing (scRNA-seq)'.

The scRNA-seq libraries were generated using the 10× Genomics Chromium Next GEM Single Cell 3′ Reagent Kit (v.3) and sequenced at the Genomics Core Facility of Cancer Research UK (CRUK), Cambridge Institute. Libraries were generated in two different batches. Information about library batches can be found in Supplementary Table 2. Control samples were included in both batches to provide a reference to assess

potential batch effects. The cells for each biological replicate were loaded into a 10× Chromium microfluidics chip channel to generate one library from each. In total, 17 libraries were sequenced on either an Illumina HiSeqx4000 or a NovaSeq6000 system using one SP, two S1 and two S2 flow cells. Note that, given the punch biopsy approach used, DEN samples could contain sporadic tumour cells from tumours not visible under the dissection microscope.

**scRNA-seq preprocessing, dimensionality reduction and visualization.** The raw scRNA-seq data were processed with CellRanger (v.7.0.1). Reads were aligned to the mouse reference genome (mm10 2020-A), empty droplets were filtered and unique molecular identifiers were counted to generate gene-expression matrices. Doublets were identified using Scrublet[66] (v.0.2.3) and removed, along with low-quality cells, on the basis of per-sample quality-control metrics (Supplementary Table 2); cells with more than 15% mitochondrial reads or genes expressed in fewer than three cells were excluded, resulting in 91,347 high-quality cells. Count matrices were processed using a standard Seurat workflow[67] (v.5.0.3) up to dimensionality reduction. Data were integrated by tissue of origin (oesophagus or forestomach) using Harmony[68] (v.1.2.3), and the integrated embeddings were projected in two dimensions using Seurat's RunUMAP function. Further details on data clustering, annotation, and trajectory and communication analysis can be found in the Supplementary Methods and on the Alcolea lab's GitHub page.

## Low-input targeted DNA sequencing

Oesophagi from control and DEN-treated animals were dissected as described above. Tissues were flattened with the epithelial side up and visible tumour lesions were marked with a partial incision using a 1 mm diameter punch tool under a dissecting microscope. Tissues were then incubated in 5 mM EDTA for 3 h at 37 °C while rotating. After incubation, the epithelium was removed, the stroma was flattened and tissues were fixed as described above.

Immunofluorescent labelling against KRT14 was done as described above. Only tumour stroma footprints negative for KRT14 (lacking epithelial cells) were considered for DNA sequencing to avoid the identification of genetic mutations present in epithelial cells. Tumour stroma matching the criteria was dissected under a fluorescence dissecting microscope (Leica M165 FC) using a 1 mm punch biopsy tool. Punch biopsies of equivalent size were collected from untreated healthy tissues as a control. The DNA from individual biopsies was extracted using the Arcturus PicoPure DNA Extraction Kit (Fisher Scientific, KIT0103) following the manufacturer's instructions. In brief, proteinase K was reconstituted in 155 µl and the sample was lysed in 20 µl at 65 °C overnight. Thereafter, proteinase K was inactivated by incubation at 95 °C for 10 min.

Samples were then sheared, libraries prepped using the NEBNext Ultra II Fragmentase System, and index tags applied (Sanger 168 tag set). Material was subjected to 12 PCR cycles (initial denaturation; 95 °C for 5 min; 12 cycles of 98 °C for 30 s; 65 °C for 30 s; 72 °C for 2 min; final elongation, 72 °C for 10 min) and quantified (Accuclear dsDNA Quantitation Solution, Biotium). Then, 500 ng of pooled material was taken forward for hybridization capture and enrichment (SureSelect Target enrichment system, Agilent Technologies) using a previously designed bait panel of 192 genes (Supplementary Table 9), including those commonly mutated in squamous cancers[22]. After clean-up, libraries were normalized to around 6 nM and submitted to cluster formation for sequencing on a Novaseq6000 (Illumina) to generate 100-base pair paired-end reads.

Aligned reads were mapped to the mouse GRCm38 reference genome using BWA-mem (v.0.7.17)[69]. Duplicate reads were marked using SAMtools[70] (v.1.11). Depth of coverage was also calculated using SAMtools to exclude reads that were unmapped, not in the primary alignment, failed platform or vendor quality checks, or PCR or optical duplicates.

Bedtools (v.2.23.0)[71] was then used to calculate the depth of coverage per base across samples.

Variant calling was done using the deepSNV R package (also commonly referred to as ShearwaterML; v.1.21.3; https://github.com/gerstung-lab/deepSNV). Variants were annotated using VAGrENT. R v.3.3.0 was used[7,72].

Mutations called by deepSNV ShearwaterML were filtered using the following criteria: first, positions of called single nucleotide variants (SNVs) have a coverage of at least 100 reads; second, germline variants called from the same individual are omitted from the list of called variants; third, adjustment for false discovery rate and mutations use support from at least one read from both strands for the mutations identified; and finally, pairs of SNVs on adjacent nucleotides in the same sample are merged into a dinucleotide variant if at least 90% of the mapped DNA reads containing at least one of the SNV pairs also contained the other one. DeepSNV ShearwaterML was run with a normal panel of approximately 12,000 reads.

### Statistics and reproducibility

The numbers of biological replicates and animals are indicated in the figure legends (*n* refers to the number of independent replicates per time point and/or condition). A minimum of three independent mice or ex vivo cultures were used in all cases. All experiments were done independently at least three times with similar results, unless otherwise stated. The reproducibility of all key findings was confirmed in independent experiments conducted on different days and using independent biological samples. For image analysis, a minimum of three independent samples were inspected in all cases. All figures show representative images. The data are expressed as mean values ± s.e.m. unless otherwise indicated.

All statistical tests were done comparing biological replicates. Differences in tumour burden were assessed by one-tailed unpaired non-parametric Mann–Whitney *U*-tests. Differences between Niche– and Niche+ tumour distribution in ageing animals was assessed by one-sided chi-squared test. For large datasets, normality was assessed using a Kolmogorov–Smirnov test; for normally distributed data, differences between two groups were assessed by two-tailed Welch's *t*-tests; for non-normally distributed data, a two-tailed Mann–Whitney *U*-test was used. Differences between more than two groups were calculated using either one-way Welch's ANOVA, followed by a Dunnett's T3 multiple-comparison test, or Kruskal–Wallis one-way ANOVA, followed by Dunn's multiple-comparison test, for normally distributed or non-normally distributed data, respectively, unless specified otherwise. Exact *P*-values are indicated in the relevant figures with a precision of up to four decimal places. Statistical tests were conducted in GraphPad Prism (v.10.5.0) with 95% confidence intervals. No statistical method was used to predetermine sample size. The experiments were done without randomization. Blinding was done for tumour count per condition and in vitro sample analyses by confocal microscopy. In cases for which quantification was done in tumours and morphologically normal areas, blinding was not possible owing to differences in physical sample appearance.

### Reporting summary

Further information on research design is available in the Nature Portfolio Reporting Summary linked to this article.

### Data availability

Mouse reference genomes GRCm38 and mm10 2020-A were used. The single-cell RNA sequencing data generated in this study have been deposited in the Gene Expression Omnibus (GEO) repository under accession code GSE271962. The DNA sequencing dataset was deposited at the European Nucleotide Archive (ENA) under dataset accession number ERP134942. Source data are provided with this paper.

## Code availability

No new algorithms were developed for this paper. The analysis code is available on the Alcolea lab's GitHub page.

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

**Acknowledgements** We thank members of the Alcolea lab for comments and suggestions; I. Michalk for coordinating human sample collection and documentation at the Tumour and Normal Tissue Bank Dresden; J. Cordle for coordinating human sample collection and delivery from Guy's and St Thomas' NHS Foundation Trust; the Cancer Aging and Somatic Mutation (CASM) teams for accommodating work related to human samples, especially E. Anderson, L.-A. Gilbey and the rest of the CASM research management team; P. Humphreys and D. Clements for guiding imaging analysis at the Jeffrey Cheah Biomedical Centre (JCBC); N. Lawrence for guiding imaging analysis at the Gurdon Institute Imaging Facility; I. Pshenichaya for technical histology support at the JCBC; B. Mahler-Araujo and J. Warner for histopathology analysis; D. Streichert for technical support with human-sample staining; M. Paramor and V. Murray; K. Kania; I. Mohorianu's team for their contribution to scRNA-seq sample processing, library preparation and data pre-processing; the staff of the University Biomedical Services Gurdon Institute and the Anne McLaren Building technical biomedical assistance; and I. J. Jackson (Fucci2a) and H. Clevers (Rosa26Confetti) for donating mouse lines. This work was supported by grants to M.P.A. from the Wellcome and The Royal Society (105942/Z/14/Z and 105942/Z/14/A), the Medical Research Council (MR/P019013/1), the Worldwide Cancer Research and Guts UK (19-0192 and 23-0063). This research was funded in whole, or in part, by Wellcome (203151/Z/16/Z, 203151/A/16/Z) and the UKRI Medical Research Council (MC_PC_17230), and core support grant for Cambridge Stem Cell Institute Discovery Research Wellcome Platform Discovery Research Platform for Tissue Scale Biology (226795/Z/22/Z). This work also received funding from the European Research Council (ERC) Executive Agency under HORIZON ERC Synergy Grant Programme (grant agreement 101167202 — ClonEScape — ERC-2024-SyG, to M.P.A.). G.S. was funded by the Isaac Newton Trust (21.07(a)), the Medical Research Council (MR/P019013/1) and Worldwide Cancer Research (19-0192 and 23-0063). J.E.R.A. was supported by a University of Cambridge/Wellcome Junior Interdisciplinary Fellowship (ISSF 11/2/2020) and the Medical Research Council (MR/P019013/1). Y.D. was funded by an ELBE Postdoctoral Fellowship from the Center for Systems Biology Dresden (CSBD). M.T.B. received funding from the European Union's Horizon 2020 research and innovation programme under Marie Sklodowska-Curie grant agreement 794664 (OESOPHAGEAL FATE). H.A. and M.T.B were also supported by the Isaac Newton Trust (research grants 16.24(e)) and the Leverhulme Trust (RPG-2023-136). D.R. was funded by the Cancer Research UK Cambridge Centre (CANCTA-2023/100003). S.H. acknowledges funding from the Human Frontier Science Program (LT000092/2016-L). P.H.J., K.M. and J.C.F. were supported by Wellcome (108413/A/15/D). P.H.J. was also supported by Cancer Research UK programme grants (C609/A27326 andDRCRPG-Nov24/100002). A.N. and W.K. were funded by CRUK (RCCCSF-Nov22/100005). D.A., M.H.H.S. and J.J were funded by intramural sources of TU Dresden allocated to the Institute of Pathology and Institute of Anatomy. S.R. received funding from the ERC under the European Union's Horizon 2020 research and innovation program (grant agreement 950349) and acknowledges the Center for Nanoscience (CeNS), Munich. B.D.S. acknowledges funding from the Royal Society (E.P. Abraham Research Professorship, RSRP\R\231004) and Wellcome (219478/Z/19/Z).

**Author contributions** G.S., J.E.R.A. and M.P.A. designed, validated and conducted experiments; S.H. guided the experimental design for single-cell RNA sequencing and, together with H.A. and J.E.R.A., did the data processing. H.A. and J.E.R.A. did the trajectory inference analysis. H.A. and Y.D. did cell-to-cell communication analysis and advised and supported the single-cell RNA sequencing analysis. S.R. analysed lineage-tracing data and provided advice and guidance on single-cell RNA sequencing data analysis. J.C.F. processed DNA sequencing samples and did data analysis. B.C. provided insights and technical expertise in targeted DNA sequencing. M.T.B. did in vivo transplantation assays. A.N. and S. Z. helped with human

tissue recruitment and D.R., K.M. and W.K. assisted with human tissue experiments. D.A. did histopathological staging of human tumour samples. M.H.H.S. and J.J. advised on comparative anatomical aspects of human and rodent oesophagus and sample processing. M.K.S.T. assisted with in vitro validation experiments. M.T.B., B.C., P.H.J. and B.D.S. advised on parts of the study, provided expertise regarding epithelial stem cells and tumour biology, and assisted with writing the manuscript. G.S. and M.P.A. conceived the project, supervised and performed experiments, and wrote the manuscript with input from all authors. Review and editing of the manuscript was done by all authors. For the purpose of open access, the author has applied a CC BY public copyright licence to any Author Accepted Manuscript version arising from this submission.

**Competing interests** The authors declare no competing interests.

**Additional information**
**Correspondence and requests for materials** should be addressed to G. Skrupskelyte or M. P. Alcolea.

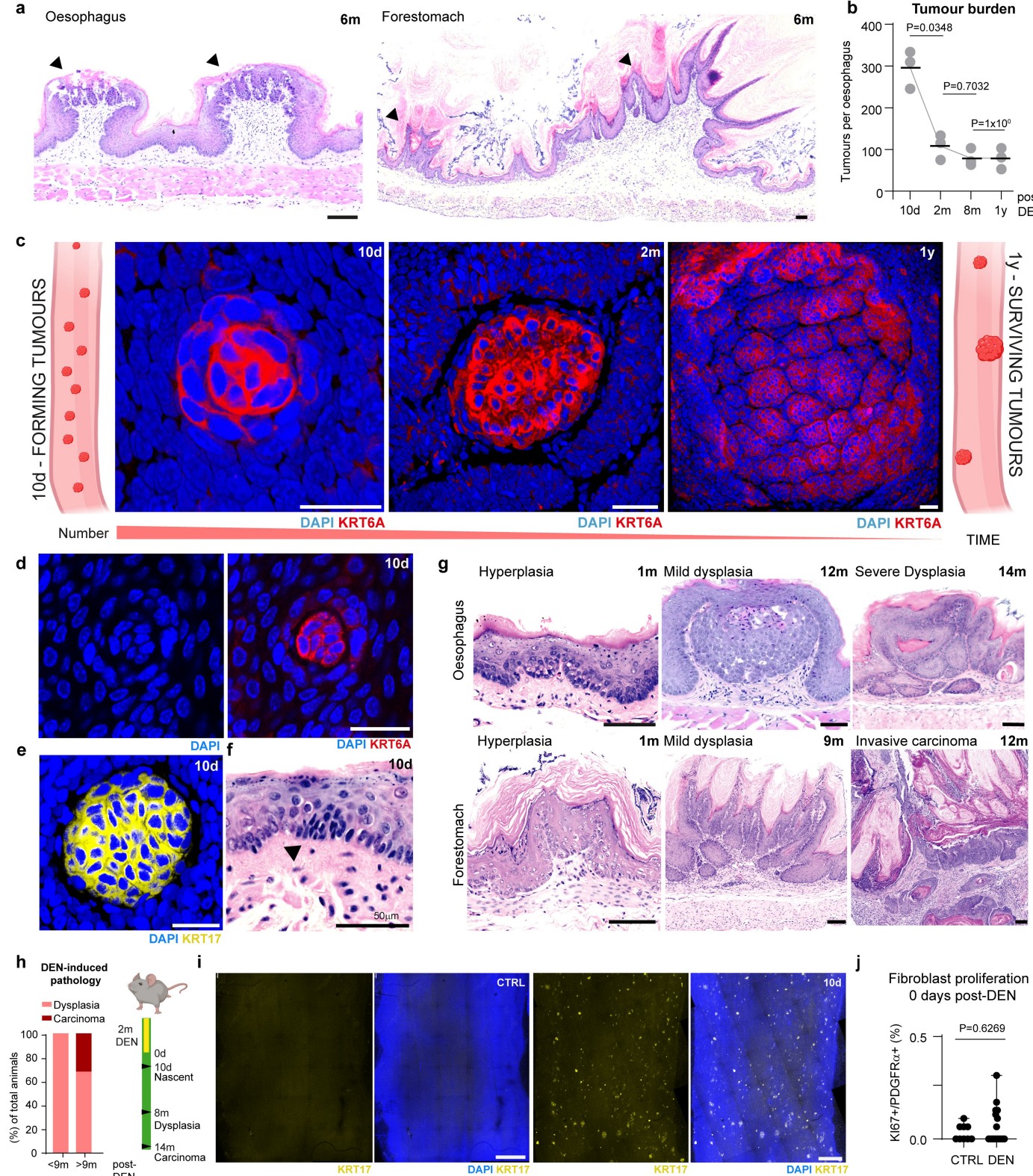

**Extended Data Fig. 1** | See next page for caption.

**Extended Data Fig. 1 | Early tumorigenesis model in squamous upper-gastrointestinal tract of a mouse; (Corresponds to Fig. 1). a**, H&E image of transversal cross-sections from the murine upper gastro-intestinal tract (including oesophagus and forestomach) displaying multiple tumours (black arrowheads) 6 m after DEN treatment. Scale bars, 100 μm. **b**, Number of tumours per oesophagus (OE) at the indicated time points after DEN treatment, n = 3 mice per time point. A line is drawn across the means of each time point. One way Welch's ANOVA with multiple comparison was used to assess significance, indicated p values show two consecutive timepoint comparison indicated by the black line. **c**, Cartoon (edges) illustrating tumour clearance over time, with examples of confocal images (middle) of growing persistent tumours stained for DAPI (blue) and KRT6A (red). Tissues were collected at the indicated time points after DEN treatment: 10 d, 2 m and 1 y from 5 mice per timepoint. Scale bars, 25 μm. **d-e**, Confocal images showing early tumour marker expression in nascent tumours 10 d after DEN treatment from 18 mice. DAPI (blue) (**d** & **e**); KRT6A (red) (**d**); KRT17 (yellow) (**e**). Scale bars: 25 μm. **f**, H&E image of a transversal cross-section depicting a nascent tumour 10 d after DEN treatment (black triangle), representative image from 4 mice. **g**, H&E images of tumours transversally cross-sectioned at different time points after DEN treatment at different stages of progression in the squamous upper gastro-intestinal tract. Scale bars, 50 μm. n = 4 (1 m), 4 (9 m), 10 (12 m), 2 (14 m) mice per timepoint. **h**, Bar plot (left) showing distribution of DEN induced pathology before (<) and after (>) 9 m after DEN treatment. Note, 10 d tumours are not identifiable to pathologists due to their size. **i**, Representative whole mount confocal images from control (ctrl) or 10 d post-DEN oesophagi showing localized KRT17 expression (yellow) in tumours, otherwise absent in surrounding or age-matched control tissues. Scale bar, 500 μm. **j**, Percentage of proliferating (KI67+) fibroblasts in DEN-treated oesophagi (0 d post-DEN) and in normal age-matched controls from *Pdgfra*$^{EGFP}$ mice. n = 17 and 9 areas per condition respectively, each from 3 animals. Data is expressed as mean±s.e.m. Statistical significance was assessed by a two-tailed Mann-Whitney U test. Days (d), months (m), year (y). Illustrations in **c**,**h** were created in BioRender; Alcolea, M. https://BioRender.com/z50odrm (2026). Source data (**b**,**h**,**j**).

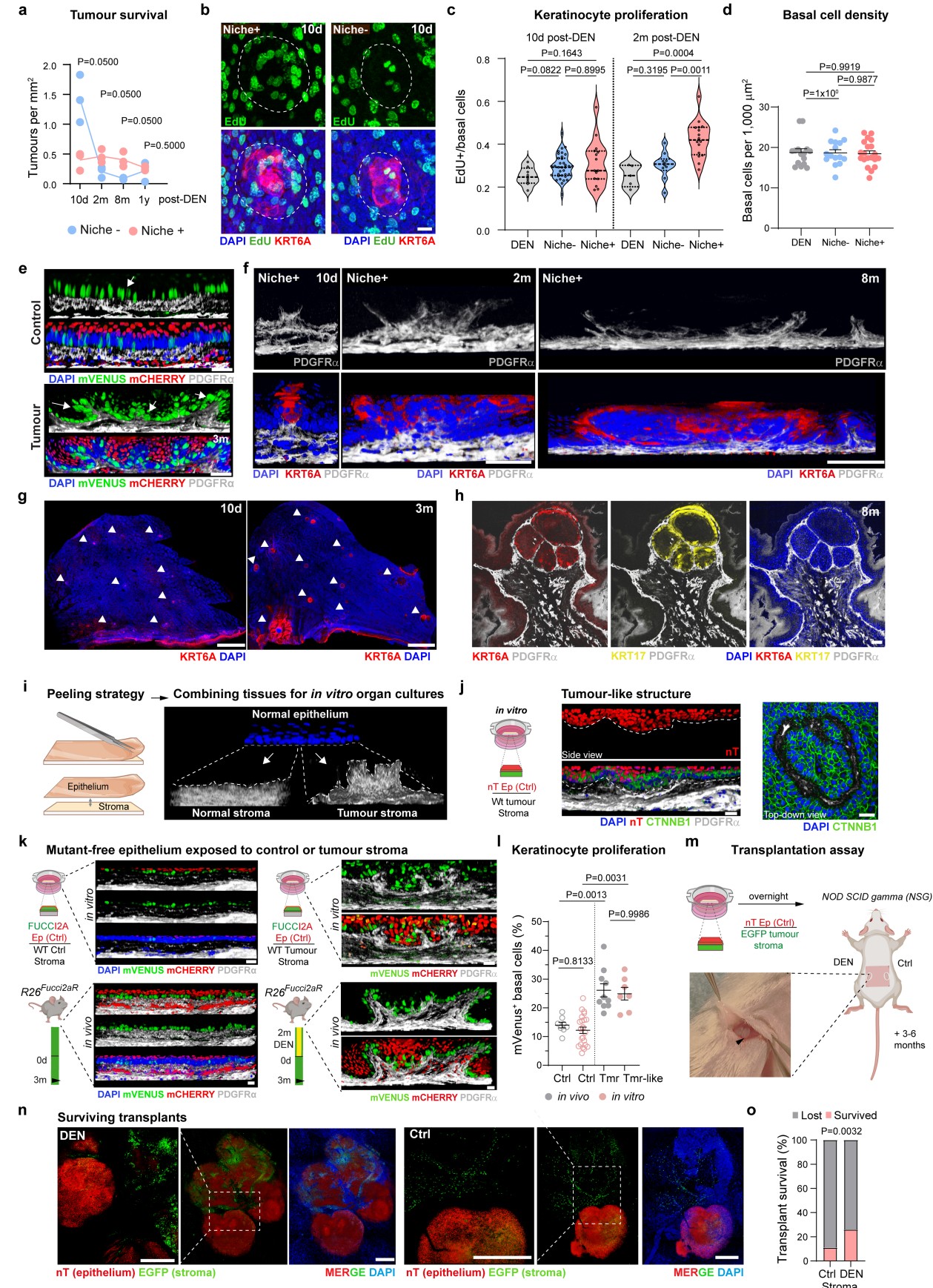

**Extended Data Fig. 2 |** See next page for caption.

**Extended Data Fig. 2 | Pre-cancerous niche promotes epithelia proliferation and tumour growth; (Corresponds to Fig. 1). a**, Density of Niche+ and Niche− tumours at the indicated time points after DEN treatment. n = 3 mice per time point; solid lines represent means. One-tailed Mann-Whitney test Niche− versus Niche+ tumours per time-point. **b**, Confocal images showing EdU incorporation (green) in KRT6A+ nascent tumours (red, dashed line) 10 d after DEN treatment. DAPI (blue). Scale bars, 10 μm. **c**, Number of EdU+ basal keratinocytes normalised per basal cell density at the indicated time points after DEN treatment. N, number of normal areas or Niche− or Niche+ tumours; n = 84 (10 d), 20 (2 m) Niche− tumours, n = 35 (10 d), 38 (2 m) Niche+ tumours, and n = 20 (10 d), 14 (2 m) DEN areas from 3 mice per condition. Data expressed as violin plots with mean (solid lines) and quartiles (dashed lines). One-way Welch's ANOVA with multiple comparison was used to assess significance. **d**, Quantification of basal keratinocyte cells per area in DEN, Niche− or Niche+ early tumours. N is number of DEN or tumour areas from 3 mice. Data expressed as mean+s.e.m. One-way Welch's ANOVA with multiple comparisons was used to assess significance. **e**, 3D-rendered confocal side-views of a tumour 3 m after DEN treatment and its respective age-matched control showing $R26^{Fucci2aR}$ tissue (mCherry, G1 cells; red mVenus, S/G2/M cells; green). PDGFRα (fibroblasts, grey). Scale bar, 50 μm. White arrows mark proliferating cells in the basal layer. **f**, 3D-rendered confocal side-views showing growing Niche+ tumours at the indicated time points after DEN treatment; DAPI (blue); KRT6A (red); PDGFRα (greyscale). **g**, Representative confocal images of forestomach epithelia 10 d and 3 m after DEN treatment. DAPI (blue); KRT6A (red). White arrowheads highlight tumours at indicated time points. Scale bars, 1.5 mm. Data form n = 3 animals per timepoint. **h**, Representative confocal images of a transversal cross-section of a forestomach tumour showing the presence of the stromal niche 8 m after DEN treatment. DAPI (blue); KRT6A (red); KRT17 (yellow); PDGFRα (grey). Scale bar, 25 μm. Data from n = 4 animals. **i**, Schematic illustration of ex vivo 3D heterotypic organ cultures. After separating epithelial and stromal layers, control epithelia were combined with stroma of surviving tumours 3 m after DEN withdrawal (dashed lines). This was compared to tissue constructs combining control epithelia and control stroma. **j**, Schematic representation of the heterotypic construct approach used (left). Ep, epithelium; WT, wild-type. Images (middle and right) show the emerging tumour like structure in control epithelium, side and top-down views, respectively. Grafted epithelium expressed nuclear tdTomato (nT, red, from *nTnG* mice). DAPI (blue); β-catenin (CTNNB1, green), PDGFRα (greys). Scale bars, 25 μm. Data from 10 biological replicates. **k**, Experimental protocol (left) and representative side-views of 3D-rendered confocal images (right) of 3D heterotypic tissue constructs 7 d post-culture and in vivo control or tumour tissue for comparison. PDGFRα labels fibroblasts (greyscale); in $R26^{Fucci2aR}$ tissue mCherry, G1 cells (red); mVenus, S/G2/M cells (green). Scale bars, 25 μm. Ep, epithelium; WT, wild-type; Ctrl, control. Tumour bearing tissue was collected 3 m after DEN treatment. **l**, Quantification of cycling (mVenus+) basal keratinocytes (Krt) from **k**, expressed as percentage of total number of basal cells. Dots are data from an individual tumour (Tmr, in vivo), tumour-like structure (Tmr-like, heterotypic culture), and the respective in vivo and in vitro control areas. Dots represent areas assessed across different biological replicates n = 9 control areas, from 3 mice in vivo, 23 control areas from 3 mice in vitro and n = 10 tumours from 6 mice in vivo and 7 tumour like structures from 5 mice in vitro. Data expressed as mean±s.e.m. One-way Welch's ANOVA with multiple comparisons were used to assess significance. **m**, Schematic representation of the subcutaneous heterotypic grafting approach to NOD SCID gamma (NSG) immunodeficient mice. Grafted epithelium expressed nuclear tdTomato (nT, red, *nTnG* mice), while accompanying stroma was EGFP (green, from *H2B-EGFP* mice) to distinguish the origin of cells in the graft. DEN treated tissue was collected 3 m after DEN treatment. **n**, Representative confocal images (from **m**) showing the long-term survival of a graft combining healthy untreated nT epithelium with EGFP tumour or control stroma. Scale bars 500 μm. **o**, Quantification of surviving and lost graft constructs from **n**, 3-6 m post-transplantation expressed as percentage. 19 control and 23 DEN constructs were transplanted across 10 animals, respectively. Statistical significance was determined by a one-sided Chi-squared test. Days (d), months (m), control (ctrl). Illustrations in **i,j,k,m** were created in BioRender; Alcolea, M. https://BioRender.com/kvs2ti9 (2026). Source data (**a,c,d,l,o**).

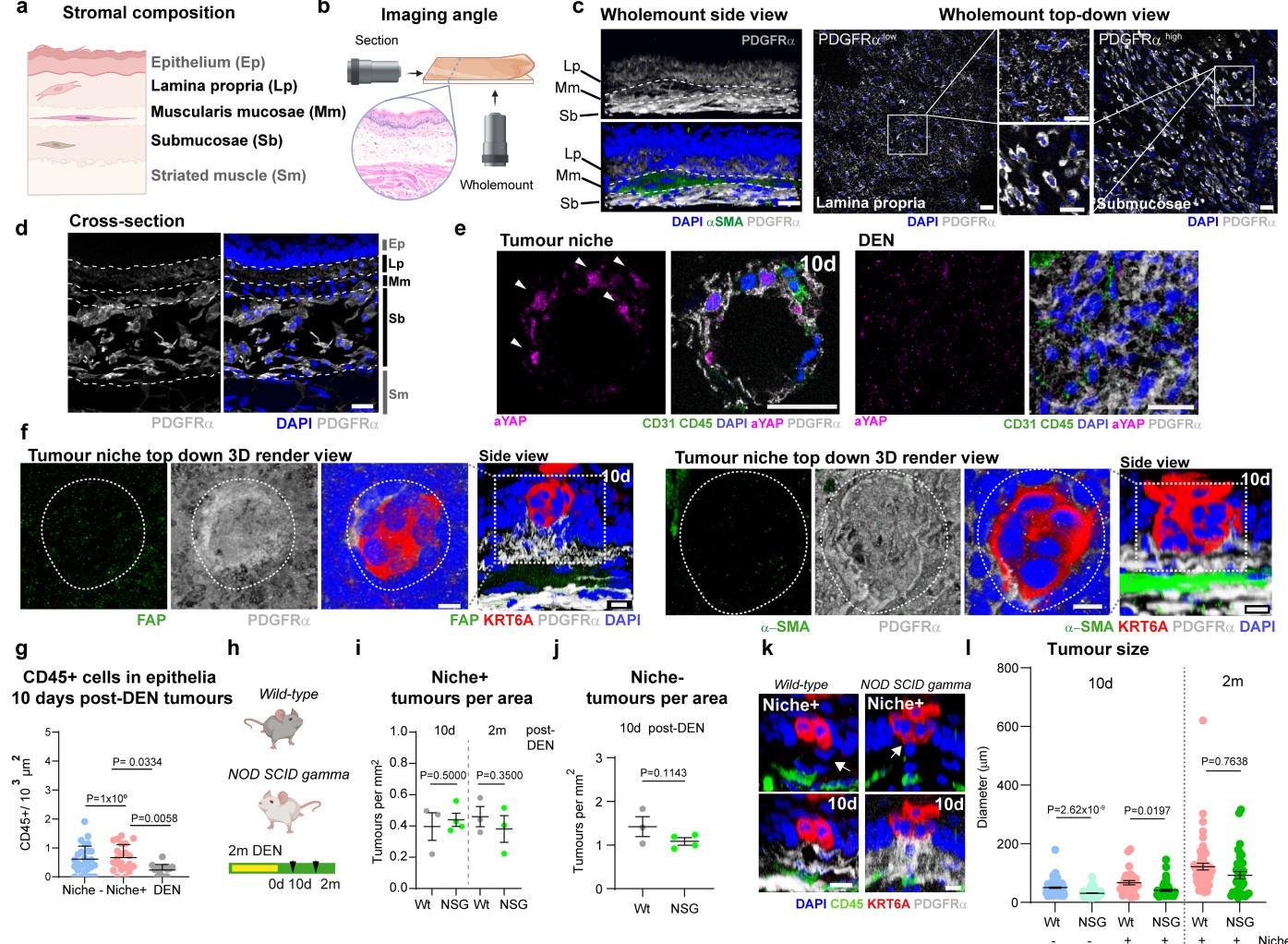

**Extended Data Fig. 3 | Nascent tumour niche is mainly comprised of fibroblast; (Corresponds to Fig. 2). a**, Schematic representation of tissue layers of the murine upper-gastrointestinal tract. **b**, Schematic representation of the imaging angle used according to the sample preparation method. **c**, Image of tissue whole-mounts showing differential PDGFRα (grey) expression between the fibroblast layers. α-SMA smooth muscle layer (green), DAPI (blue). Scale bar, 25 μm. Lamina propria-Lp, Muscularis mucosae- Mm, Submucosae-Sb. **d**, Image of a tissue section showing reduced PDGFRα expression in upper stromal layers (i.e. lamina propria). Dashed lines (**c,d**) separate the tissue layers as listed in schematic. **e**, Confocal images of nascent tumour stroma 10 d after DEN treatment. Images show active YAP (aYAP) expression in PDGFRα labelled fibroblasts (white arrowheads). aYAP (magenta); CD31 and CD45 (green); PDGFRα (grey); DAPI (blue). Scale bars, 25 μm. **f**, Representative confocal images of Niche+ tumours 10 d after DEN treatment, showing the absence of typical cancer associate fibroblast (CAF) marker in the niche forming fibroblasts. **g**, Number of CD45+ (immune) cells in epithelia of Niche−/Niche+ tumours and adjacent DEN area normalised to surface area, 10 d post-DEN. n = 27 (Niche−), 22 (Niche+) tumours, from 3 mice each, dots represent a

tumour. Data expressed as mean±s.e.m. Statistical significance assessed by one-way Welch's ANOVA with multiple comparisons. **h**, Experimental DEN carcinogen protocol in wild-type (wt) and immunocompromised NOD SCID gamma (NSG) mice. **i,j**, Niche+ (**i**) and Niche− (**j**) tumours per area in wt and NSG mice at 10 d and 2 m post-DEN withdrawal. N = 3 wt and 4 NSG mice at 10 d and n = 3 wt and 3 NSG at 2 m after DEN treatment time point; expressed as mean±s.e.m. One-tailed Mann-Whitney test was used to assess statistical significance. **k**, Representative confocal images of Niche+ nascent tumours from 10 d post-DEN showing fibroblast (PDGRα, grey) recruitment (white arrow) to a forming epithelial tumour (KRT6A, red) in wt and NSG mice. CD45 (green), DAPI (blue). Scale bar, 10 μm, **l**, Diameter (μm) of Niche+ and Niche− tumours, n = 126 (tumours in wt from 3 mice) and n = 189 (tumours in NSG from 4 mice) at 10 d post DEN and n = 57 (tumours in wt from 3 mice) and n = 36 tumours in NSG from 3 mice) at 2 m after DEN treatment. Data expressed as mean±s.e.m. Two-tailed Welch's t-test was used to assess statistical significance. Days (d), months (m). Illustrations in **a, b, h** were created in BioRender; Alcolea, M. https://BioRender.com/8klr068 (2026). Source data (**g, i, j, l**).

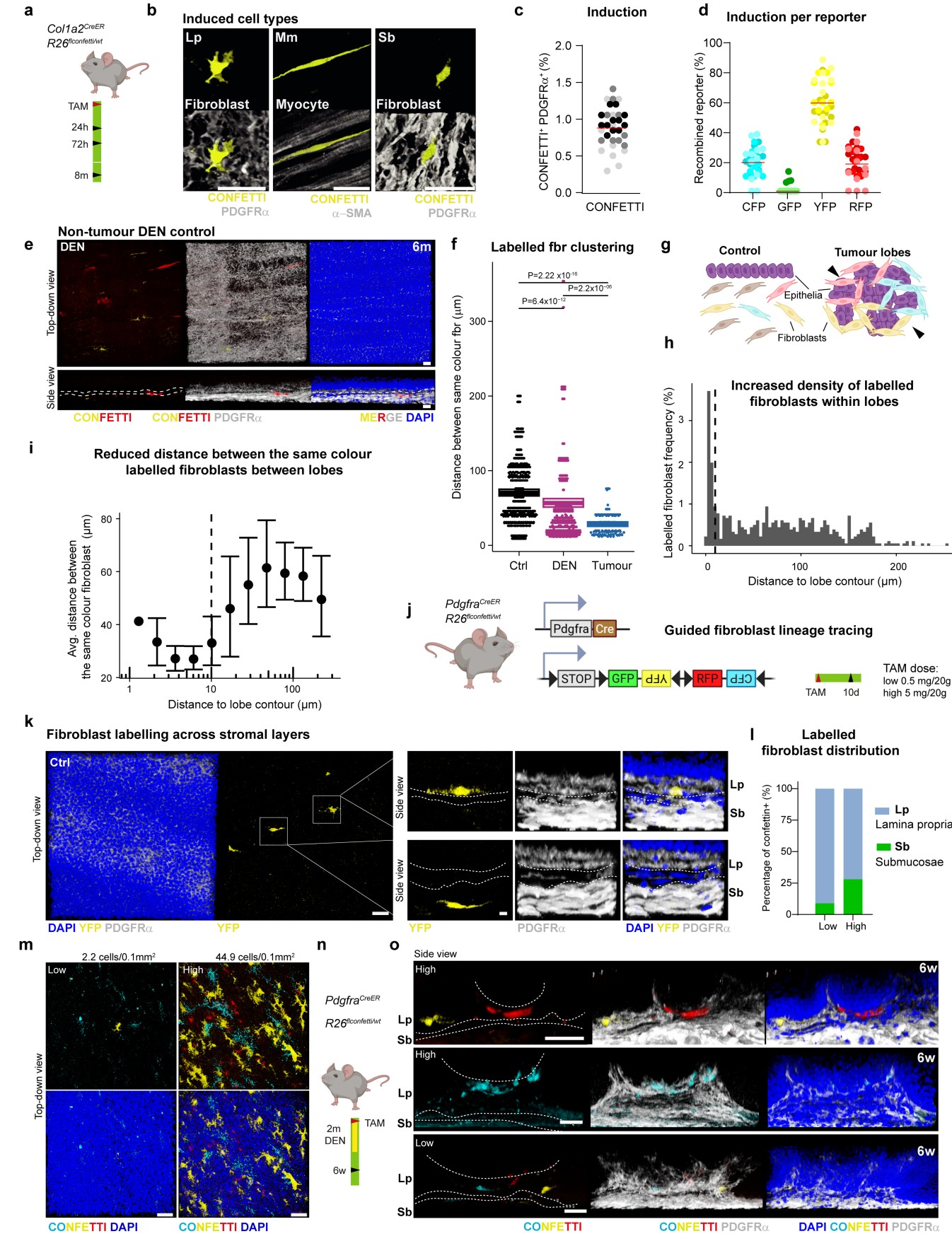

**Extended Data Fig. 4 |** See next page for caption.

**Extended Data Fig. 4 | Expansion of PDGRα^low fibroblast population in the early tumour niche; (Corresponds to Fig. 2). a**, Experimental protocol of fibroblast lineage tracing in *Col1a2^CreER Rs26^FConfetti* mice. **b**, Confocal images of confetti-labelled cells 24 h after induction in different stromal layers. Scale bars, 25 μm. **c, d**, Efficiency of confetti construct recombination after a 72 h chase in the PDGFRα+ fibroblast population. Total percentage of recombined cells in the PDGFRα+ fibroblast population (**c**), and split colour percentage within the recombined population (**d**). N represents number of fields from 3 animals, n = 36; each replicate shown with a different shade of colour. **e**, Confocal 3D top-down or side-view images of a DEN area. Scale bars, 50 μm. Corresponds to Fig. 2d-f. **f**, Distance to the nearest neighbour fibroblast (fbr) labelled in the same colour in external control (Ctrl), internal control (DEN) and tumours. The horizontal line denotes the average, and the box signifies 95% confidence intervals obtained via bootstrapping. Pairwise Wilcoxon signed-rank tests were performed, and P-values were adjusted using the Holm-Bonferroni correction. **g**, Cartoon illustrating fibroblast expansion in the early tumour niche: lineage tracing results revealed that stroma between tumour lobes show increased fibroblast density and decreased distance between the same colour fibroblasts (black arrowheads) if compared to the tissue outside of lobes or control. **h**, Histogram showing the percentage of cells in a given bin of the distance to the nearest lobe contour. **i**, The average distance to the nearest neighbour of the same colour as a function of the distance to the lobe boundary. Results show that labelled cells are closer together on the lobe contour than

elsewhere. Data expressed as mean and error bars denote 95% confidence intervals obtained from bootstrapping. The vertical dashed line in **h** and **i** denote 10 μm, the estimated width of the tumour lobe contour. **f,h** and **i** data from n = 18 areas in control from 3 animals, n = 18 areas in DEN from 3 animals and n = 16 tumours from 4 animals. **j**, Schematic representation of the confetti cassette and experimental protocol of fibroblast lineage tracing in *Pdgfra^CreERT Rs26^FConfetti* mice. **k**, Images of labelled fibroblasts (from **j**) in lamina propria and submucosa; data from 5 animals. **l**, Percentage of fibroblasts labelled in lamina propria (Lp) and submucosae (Sb) (from **j**). **m**, Representative confocal images of tissue from **j** showing top-down projections and confetti labelled fibroblast; 5 and 3 animals for low and high TAM dose, respectively. Quantified induction efficiency per field of view shown as an average above the images. **n**, Experimental DEN protocol in TAM induced *Pdgfra^CreERT Rs26^FConfetti* mice. **o**, Representative confocal images of confetti labelled fibroblasts in early tumours 6w after DEN treatment. Dashed lines mark tumour and stromal layer margins. 0.5 mg/20 g body weight is low; 5 mg/20 g body weight is high tamoxifen dose. DEN treated animals represent n = 4 (high TAM) and 3 (low TAM). PDGFRα (grey), α-SMA (in **b** grey); CONFETTI (cyan, yellow and red) in (**b**). Dashed lines in (**e,k,o**) label lamina propria and submucosae layers in side views. Hours (h), weeks (w), days (d), months (m), tamoxifen (TAM). Scale bars (**k, m, o**) 50 μm. Illustrations in **a,g,j,n** were created in BioRender; Alcolea, M. https://BioRender.com/7zgcsff (2026). Source data (**c, d, l**).

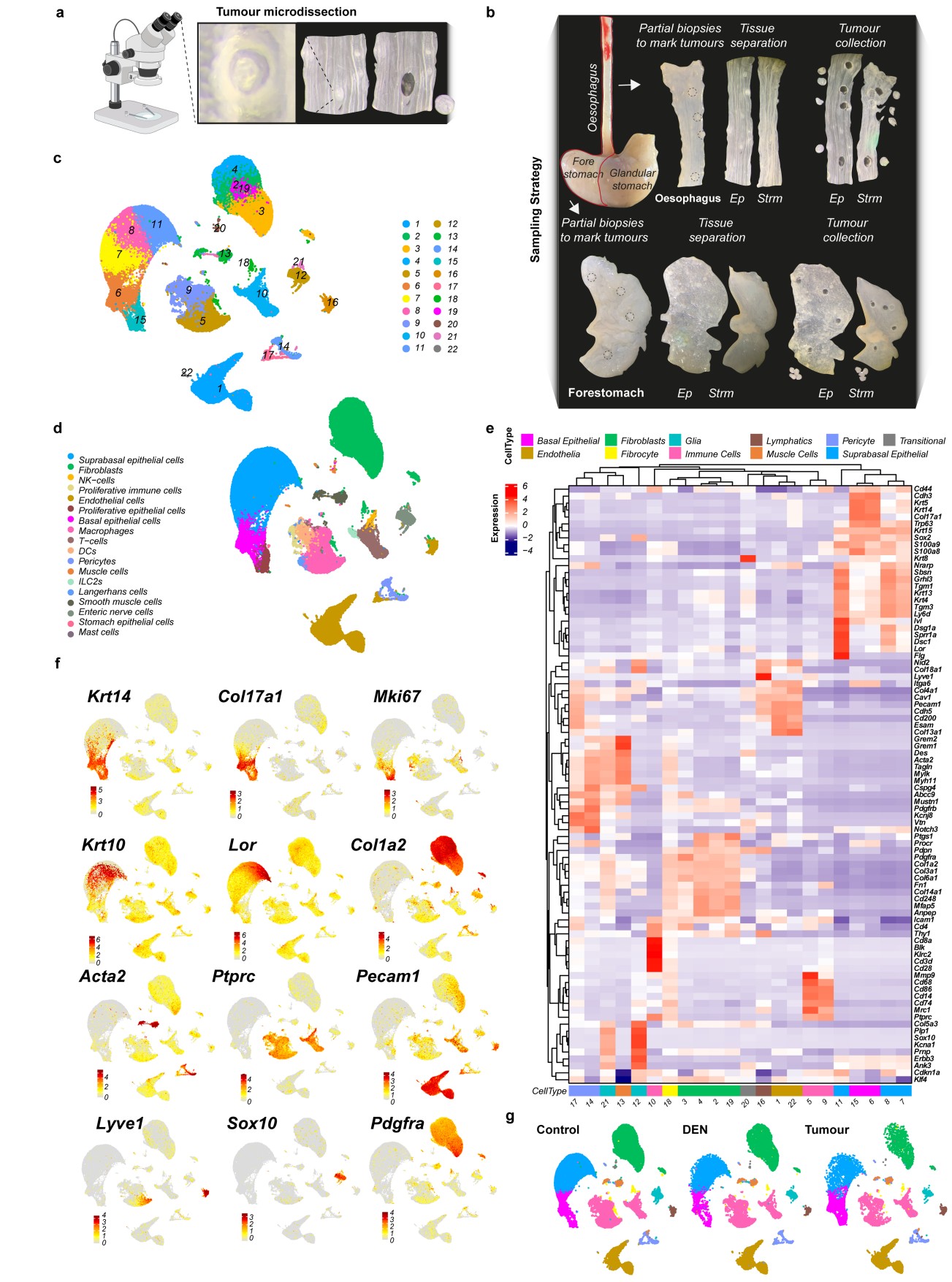

**Extended Data Fig. 5 | See next page for caption.**

**Extended Data Fig. 5 | Single cell RNA sequencing annotation; (Corresponds to Fig. 3). a**, Schematic visualisation of representative DEN-induced tumours under the dissection microscope, before and after microdissection using punch biopsy tool. **b**, Image composite depicting sample collection strategy for scRNA-seq. Squamous oesophagus and forestomach (outlined in red) first underwent tumour marking and were then peeled to separate epithelial (Ep) and Stromal (Str) compartments. This was followed by sample biopsying, where tumours and size-matched control biopsies (adjacent tissue as internal control, DEN; untreated tissue as external control, Control) were micro-dissected for scRNA-seq. **c**, UMAP representing cell cluster distribution as defined by Seurat. **d**, UMAP cell distribution of cell-types identified using label transfer from ref. 73. **e**, Heatmap showing expression of representative marker genes used for curated cell type annotation across the 22 clusters. Log-transformed normalised expression levels were averaged by cluster for each gene and scaled across all cells belonging to each group. Scale bar denotes expression range (scale: -4 to 6). **f**, UMAPs showing expression of representative genes for each cell type identified. Colour bars of UMAPs indicate log-transformed normalised expression levels. **g**, UMAP cell distribution representing identified cell types in each condition. Illustration in **a**) was created in BioRender; Alcolea, M. https://BioRender.com/l73dcie (2026).

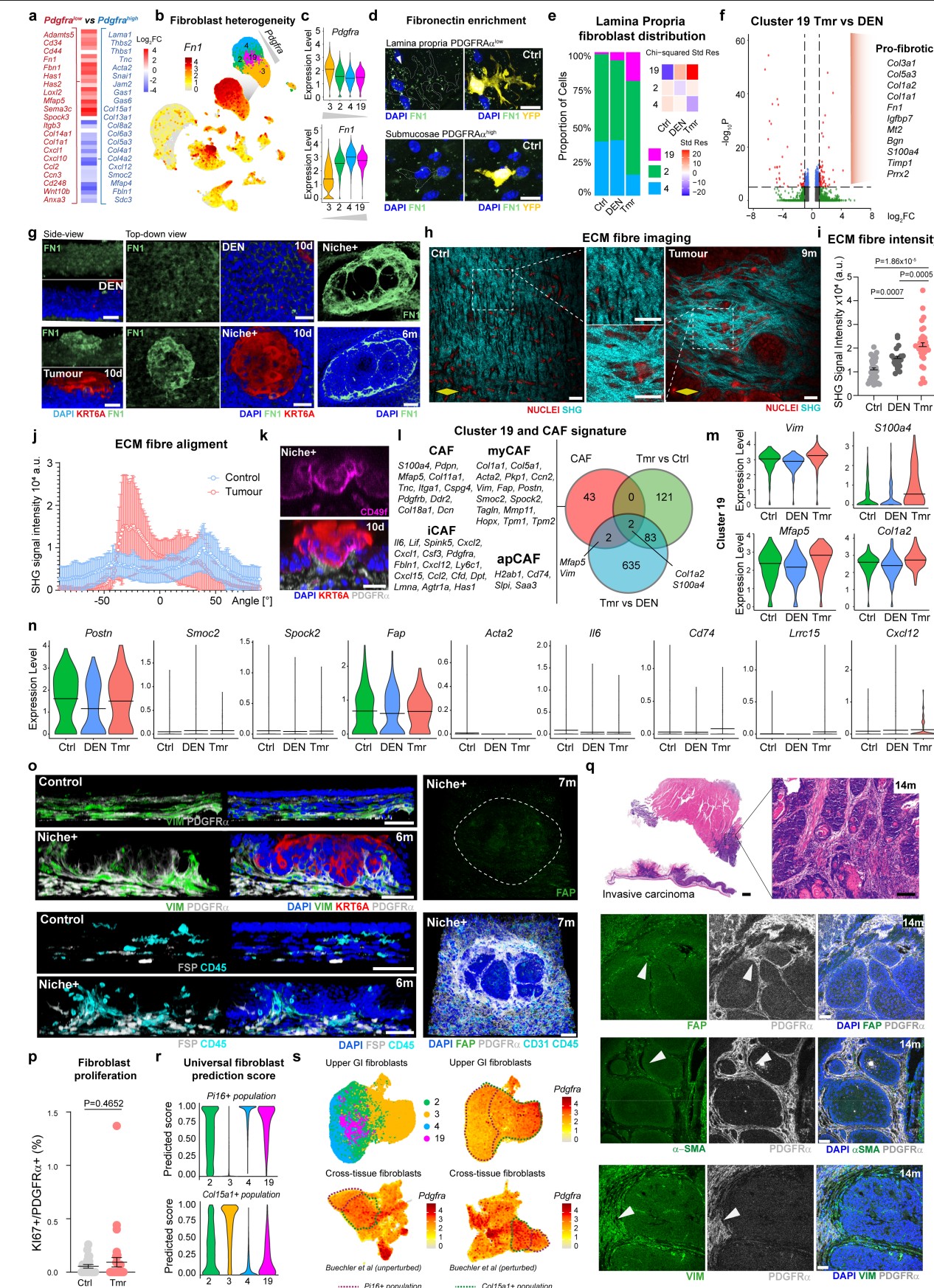

**Extended Data Fig. 6** | See next page for caption.

**Extended Data Fig. 6 | Tumour niche fibroblasts transition towards Cancer Associated Fibroblast (CAF) identity; (Corresponds to Fig. 3). a**, Heatmap showing the $\log_2$ fold-change (FC) of differentially expressed genes in $Pdgfra^{low}$ relative to $Pdgfra^{high}$ fibroblasts. Genes characterising $Pdgfra^{low}$ shown in red, $Pdgfra^{high}$ in blue. DAPI (blue). Scale bars, 10 μm. Dashed line marks fibroblast contour. **b**, UMAP projection denoting the heterogenous expression of $Fn1$ (like $Pdgfra$) in fibroblasts; inset shows fibroblast cluster distribution. **c**, Violin plots showing the levels of $Pdgfra$ and $Fn1$ expression in fibroblast clusters i.e., 3, 2, 4, and 19 in control conditions. $Fn1$ expression levels are inversely proportional to those seen of $Pdgfra$. Black line across violin plots shows is mean. **d**, Representative images of stromal whole mount showing fibronectin (FN1, green) expression in lamina propria PDGFRα$^{low}$ (white arrowhead) and submucosae PDGFRα$^{high}$ fibroblasts, labelled in yellow (from induced $Pdgfra^{CreER}Rs26^{FConfetti}$ mice; EDF 4j; n = 3 mice). **e**, Stacked bar plot (left) showing fibroblast cluster enrichment across conditions. Significance was assessed by one-sided Chi-squared test, standardised residual (Std Res) values are shown in the heatmap on the right, indicating Cluster 19 (C19) to be the most enriched cluster in tumour conditions **f**, Volcano plot showing differential gene expression between Tumour and DEN conditions in Cluster 19, $Pdgfra^{low}$ fibroblast cluster enriched in tumours. Non-significant genes ($p \geq 0.05$), green and grey. Significant genes with FC > 1, red; FC ≤ 1, blue. Genes defining the profibrotic nature of cluster 19 in tumours are labelled on the right. **g** Representative confocal images of a DEN adjacent area and nascent tumours 10 d after DEN treatment or persisting tumour 6 m after DEN treatment showing the accumulation of Fibronectin (FN1, green). DAPI (blue). Scale bars, 25 μm (10 d), 100 μm (6 m). **h**, Second harmonic generation (SHG) image of lamina propria (directly underneath the epithelium) in a control (Ctrl) and tumour areas 3 m post-DEN. Dashed lines mark insets. Extracellular matrix (ECM) fibres detected by SHG, (cyan); nuclei (red). Scale bars, 25 μm. The yellow diamonds represent the longitudinal orientation of the oesophagus. **i**, ECM fibre density was scored as SHG signal intensity 3 m after DEN treatment. N is the number of regions or tumours assessed from 6 animals; n = 36 (Ctrl), 20 (DEN), 31 (Tmr). Bars indicate mean±s.e.m. One-way Welch's ANOVA with multiple comparison was used to assess significance. **j**, Plot depicting the orientation of ECM fibres in control regions and tumours 9 m after DEN treatment. N is the number of fields of view assessed in 3 animals per condition, n = 12 (Ctrl) and 8 (Tmr). Data expressed as average ±s.d. **k**, Representative confocal image of a nascent tumour from 6 animals, showing integrin α6 (CD49f, magenta) sequestering in the early tumour niche. Samples were also labelled for KRT6A (red), PDGFRα (grey) and DAPI (blue). Image settings adjusted to upper stromal layer. **l**, Venn diagram displaying overlapping genes between known CAF markers (top left, red) and differentially expressed genes (DEGs) in DEN tumour (Tmr) fibroblasts (Cluster 19) relative to external control (Ctrl; top right, yellow) or to internal control cells (DEN; bottom, green). Overlapping genes are listed. Known subtypes of CAFs: Myofibrotic (my), immune (i), and antigen-presenting (ap) were considered. **m,n**, Violin plots showing expression of signature genes found to be enriched in Cluster 19 tumour fibroblasts (**m**) or other canonical CAF markers (**n**); external control (ctrl, green), internal control (DEN, blue), and tumours (Tmr) (red). Black line is mean. **o**, Representative 3D-rendered side-view images of control tissue and tumours 6-7 m post-DEN. Expression of CAF markers (FAP, VIM, FSP) was not detected in niche fibroblasts. KRT6A (red); CD45 (immune), and CD31 (endothelia) (cyan); Vimentin (VIM) (green); PDGFRα (top left and right) and FSP (bottom), (grey); FAP (green), DAPI, blue. Scale bars, 50 μm. Dashed line outlines tumour area. **p**, Quantification of Ki67+ EGFP+ fibroblasts from $Pdgfra^{EGFP}$ mice in control and Niche+ tumours 6-12 m after DEN treatment. N = 25 control areas (from 7 animals) and 50 tumours (from 8 animals). Data is expressed as mean±s.e.m. Two-tailed Mann Whitney test. **q**, Representative bright field (H&E) and confocal (IF) images of invasive carcinoma from 14 m, post-DEN withdrawal tissue. H&E was used to diagnose the pathology. In different panels αSMA, FAP, VIM (green) co-expression with PDGFRα (grey) show CAF emergence in DEN model. DAPI (blue). Scale bars, 100 μm (black) 50 μm (white); representative data from 5 animals >12 m after DEN treatment. **r**, Violin plots showing the prediction score from label transfer for $Pi16$+ and $Col15a1$+ cross-tissue universal fibroblast population[40] in upper GI fibroblast clusters. **s**, upper GI fibroblast clusters (top left). Remaining UMAPS show $Pdgfra$ expression in UMAP space of upper GI and cross-tissue fibroblasts[40]. Days (d), months (m). Source data (**i,p**).

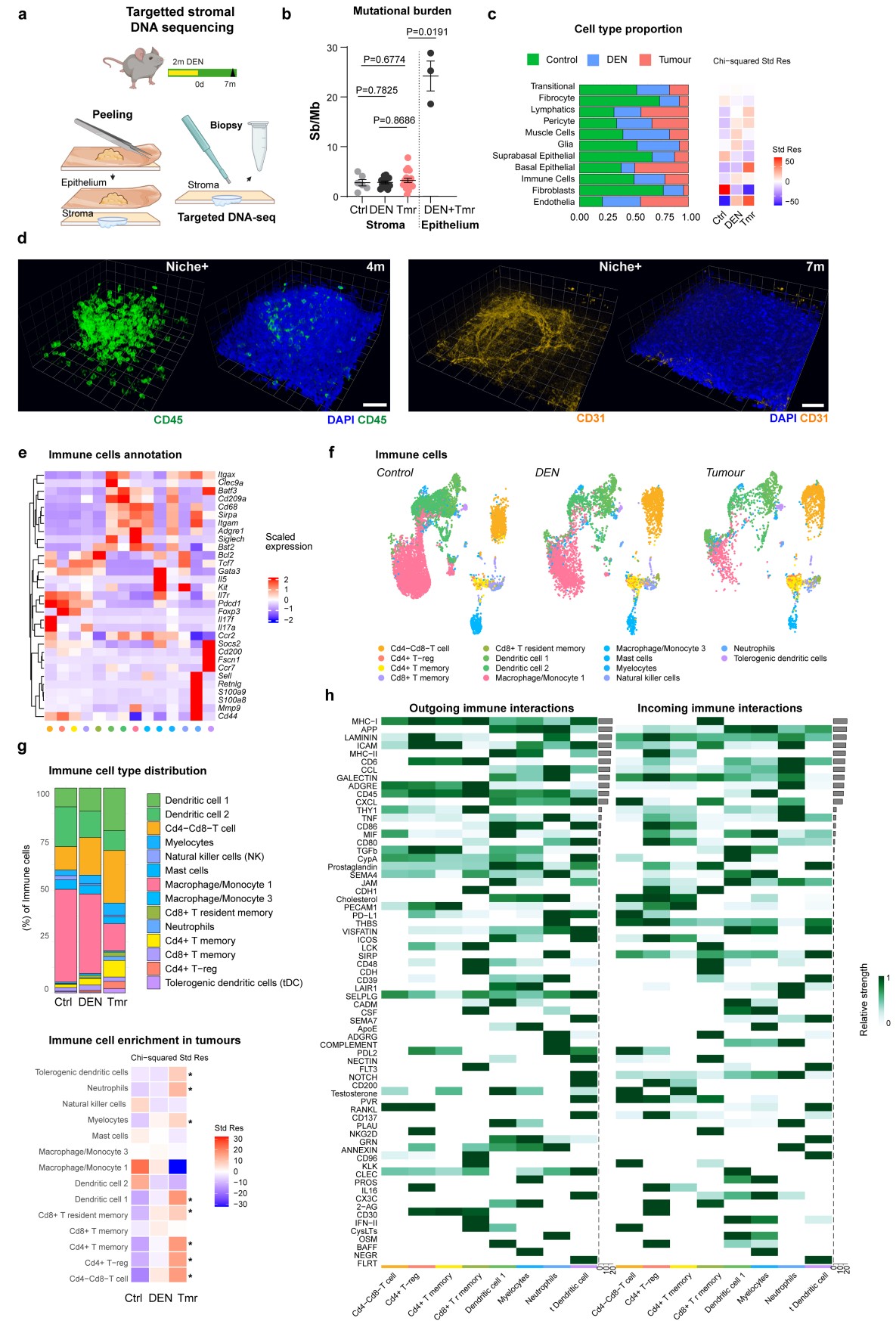

**Extended Data Fig. 7** | See next page for caption.

**Extended Data Fig. 7 | Niche of growing tumours recruits other stromal cells; (Corresponds to Fig. 3). a**, Experimental protocol for targeted DNA sequencing of stromal niche 7 m after DEN treatment. **b**, Substitutions per megabase (Sb/Mb) were calculated and used as indicative of the mutation burden in stroma across conditions. n = 6 control (from 4 animals), 13 DEN (from 6 animals) and 18 tumour (from 6 animals). Epithelial tissue (12 m after DEN treatment) was used as benchmark and includes both tumour and tumour-free tissues (DEN+Tmr; from 3 mice). Data is expressed as mean±s.e.m. Two-tailed Welch's t-test comparing tumour stroma with internal or external control stroma or with epithelium. **c**, Stacked bar plot showing scRNA-seq captured cell type distribution across conditions. **d**, Representative 3D-rendered confocal image of an early tumour (KRT6A, red) 4 m (left) and 7 m (right) post-DEN withdrawal showing the recruitment of immune cells (CD45, green) and blood vessels (CD31, orange) to the persisting tumour niche. DAPI (blue). Scale bar, 50 μm. **e**, Heatmap showing expression of representative marker genes across the 14 immune cell types identified in the scRNA-seq data. Log-transformed normalised expression levels were averaged by cluster for each gene and scaled across all cells belonging to each group. Scale bar denotes expression range (scale: -2 to 2). **f**, UMAP of identified immune cell types split by condition. **g**, Top, stacked bar plot showing immune cell type enrichment across conditions. Significance was assessed by a one-sided Chi-square test. (Bottom) heatmap showing standardised residuals values from Chi-square test. Asterisk (*) marks cell types used in CellChat analysis. **h**, Heatmaps representing outgoing (left) and incoming (right) interaction strengths as identified by CellChat and rescaled to their row maxima. Cumulative interaction strength is depicted by bars on the right. Months (m). Illustration in **a** was created in BioRender; Alcolea, M. https://BioRender.com/udeggcf (2026). Source data (**b**).

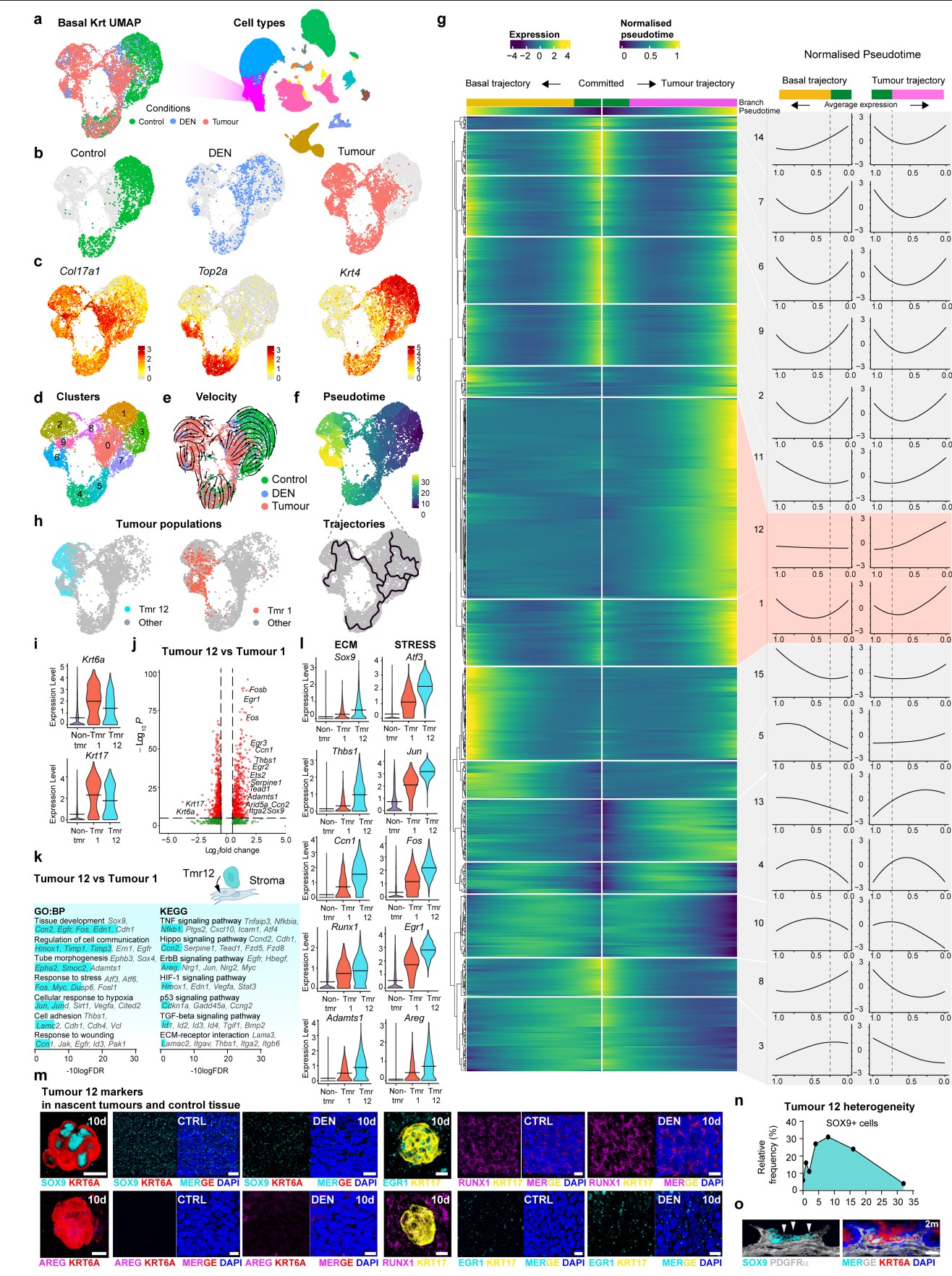

**Extended Data Fig. 8** | See next page for caption.

**Extended Data Fig. 8 | Pseudotime analysis of basal keratinocytes denotes distinctive early tumour state; (Corresponds to Fig. 3). a**, Left (inset), basal keratinocyte cell distribution in the UMAP space annotated by condition (right, UMAP consisting of all cell types). **b**, UMAP of basal keratinocyte cell distribution representing cells in each condition. **c**, UMAPs showing expression of representative basal keratinocytes genes (*Col17a1*, basal; *Top2a*, cycling basal; *Krt4*, committed). Colour bars of UMAPs indicate log-transformed normalised expression levels. **d**, UMAP showing cell clusters as defined by Seurat. **e**, Mapped RNA velocity of basal keratinocytes colour coded by condition. Black arrows are velocity vectors that show predicted cellular state change overtime. **f**, Basal keratinocyte two-dimensional pseudotime (PST) UMAP (top) and trajectories (black line) (bottom) inferred by Monocle 3. Legend shows the range of pseudotime values **g**, Heatmap (left) of the top 1500 genes identified as differentially expressed between cells in the tumour and basal trajectories. Log-transformed expression levels were scaled in a gene-wise fashion from −4 to 4 (scale) and are shown after hierarchical clustering along trajectories from committed to basal (blue to orange) and from committed to tumour (blue to magenta). The average expression patterns of the genes contained in each cluster are depicted on the right, with two modules (1 and 12) identified as tumour unique based on their preferential upregulation at the end of the tumour trajectory ($\Delta$\_tumour > 2, $\Delta$\_basal <1). (Blue, committed; Magenta, tumour; Orange, basal). Dashed vertical lines label the highest PST score of committed. **h**, Tumour cell populations representing Tumour 1 and Tumour 12 clustering in UMAP space. **i**, Violin plots highlighting *Krt6a* and *Krt17* expression between Tumour 12 and Tumour 1 states. **j**, Volcano plot showing differential gene expression between Tumour 12 and Tumour 1 states. Non-significant genes (p value higher than p = 0.05), green. Significant genes with FC $\leq$ 1, red. Genes driving stress and extracellular matrix related processes are labelled. **k**, Gene Set Enrichment Analysis of Tumour 12 versus Tumour 1 differentially expressed genes. Terms ranked by FDR (false discovery rate); representative genes listed below the bar plot. GO:BP (Gene Ontology Biological Processes); KEGG (Kyoto Encyclopedia of Genes and Genomes). Cartoon (top right) representing Tumour 12 keratinocyte communicates with stroma. **l**, Violin plots highlighting differences between Tumour 12 and Tumour 1 states. **m**, Representative confocal images from 6 mice showing expression of tumour markers: SOX9 and EGR1 (cyan); AREG and RUNX1, (magenta); KRT6A, red; KRT17 (yellow). DEN images were acquired 10 d after DEN treatment in regions adjacent to tumours. DAPI (blue). Scale bars, 10 µm. **n**, Relative frequency (%) (top) of tumours expressing cells positive for SOX9 (Tumour 12 state marker) per 1,000 µm², tumours 10 d post-DEN, n- number of tumours=119 from 3 biological replicates. **o**, Representative confocal image from 10 mice demonstrating SOX9+ (cyan) keratinocytes are situated in direct contact to stroma (PDGFRα, grey). KRT6A (red), DAPI (blue). Scale bar 50 µm. Days (d), months (m). Illustration in **k** was created in BioRender; Alcolea, M. https://BioRender.com/mhoqqbw (2026). Source data (**n**).

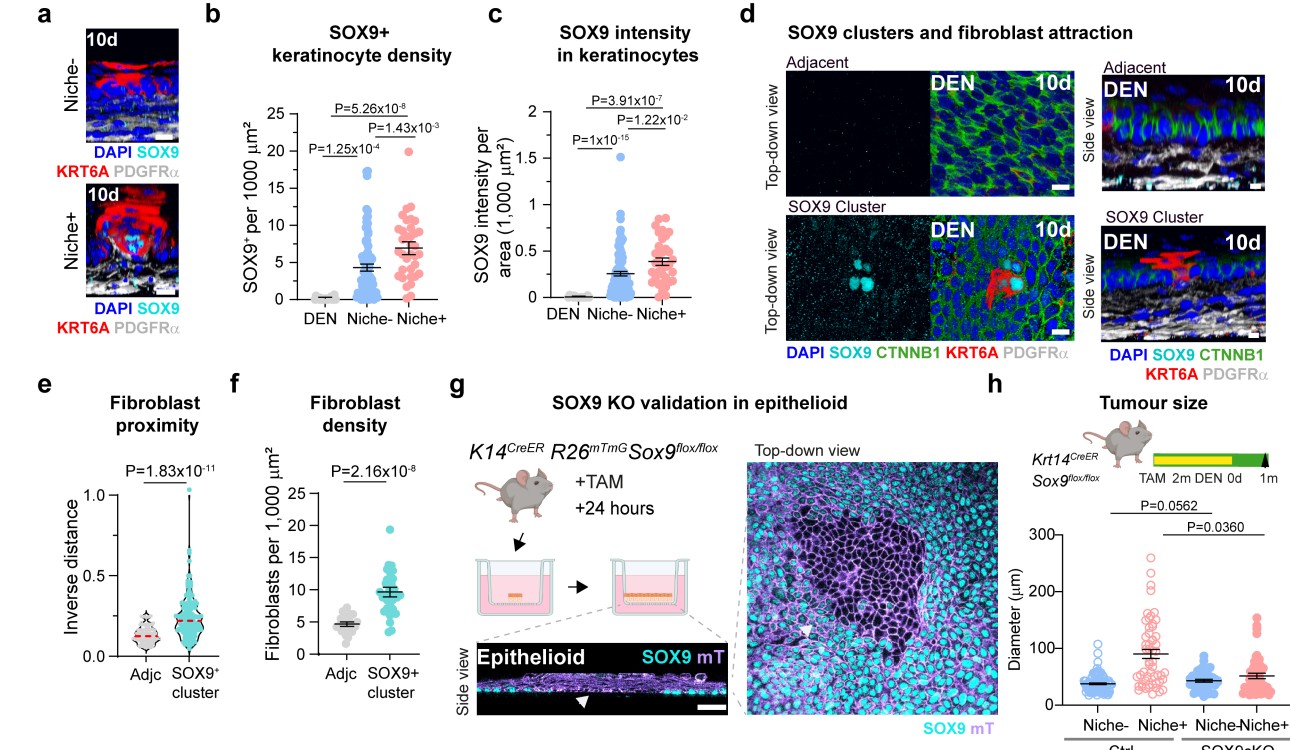

**Extended Data Fig. 9 | SOX9+ early tumour keratinocytes are associated with stromal reorganisation; (Corresponds to Fig. 3). a**, 3D-rendered confocal images of Niche+ and Niche- tumours showing KRT6A (red) and SOX9 (cyan) 10 d after DEN treatment. PDGFRα (grey), DAPI (blue). Scale bars, 10 μm. Corresponds to Fig. 3h. **b, c** Quantification of SOX9⁺ keratinocytes (krt) (**b**) and SOX9 intensity (**c**) in **a**, normalised to surface area. DEN, equivalent morphologically normal areas from DEN-treated tissues. 3 mice per condition with n = 20 DEN areas, 84 Niche- and 35 Niche+ tumours. Statistical significance assessed by one-way Welch's ANOVA with multiple comparisons. **d**, 3D-rendered confocal images (basal view, left; side-view, right) of SOX9+ keratinocyte clusters exclusively found in DEN condition. SOX9 (cyan), β catenin (green), PDGFRα (grey), DAPI (blue). Scale bar, 10 μm. Corresponds to Fig. 3i. **e**, Quantification of fibroblast proximity to basal layer from **d**, expressed as the inverse of the distance between basal keratinocytes and underlying fibroblasts. 1/distance (μm⁻¹). **f**, Number of fibroblasts directly underneath the basal layer corrected per area. The number of control (n = 27) or SOX9

expressing regions (n = 42) assessed in tissues from 4 mice. Two-tailed Welch's t-test. **g**, Strategy to validate SOX9 knock-out in basal keratinocytes (left). SOX9 expression was induced by culturing oesophageal tissue using the 3D epithelioid approach. Established cultures from tamoxifen-induced *Krt14^CreER R26^mTmG Sox9^flox/flox* mice were confocal imaged (right), confirming the presence of areas lacking SOX9 (cyan) expression. n = 3, biological replicates. White arrowheads mark the SOX9 loss. Membrane Tomato (purple), DAPI (blue). Scale bar, 25 μm. **h**, Top, experimental protocol: *Krt14^CreER Sox9^flox/flox* mice received a dose of Tamoxifen (TAM) followed by the DEN treatment. Tissues were collected 1 m after DEN treatment (from n = 3 mice) and tumour diameter was compared to DEN-treated un-induced controls (n = 3 mice). Bottom, tumour diameter (μm) of Niche+ and Niche- tumours, from the above. Data expressed as mean±s.e.m. Significance was assessed using two-tailed Welch's t-test. Days (d), months (m). Corresponds to Fig. 4j. Illustrations in **g,h** were created in BioRender; Alcolea, M. https://BioRender.com/ip2ihnq (2026). Source data (**b,c,e,f,h**).

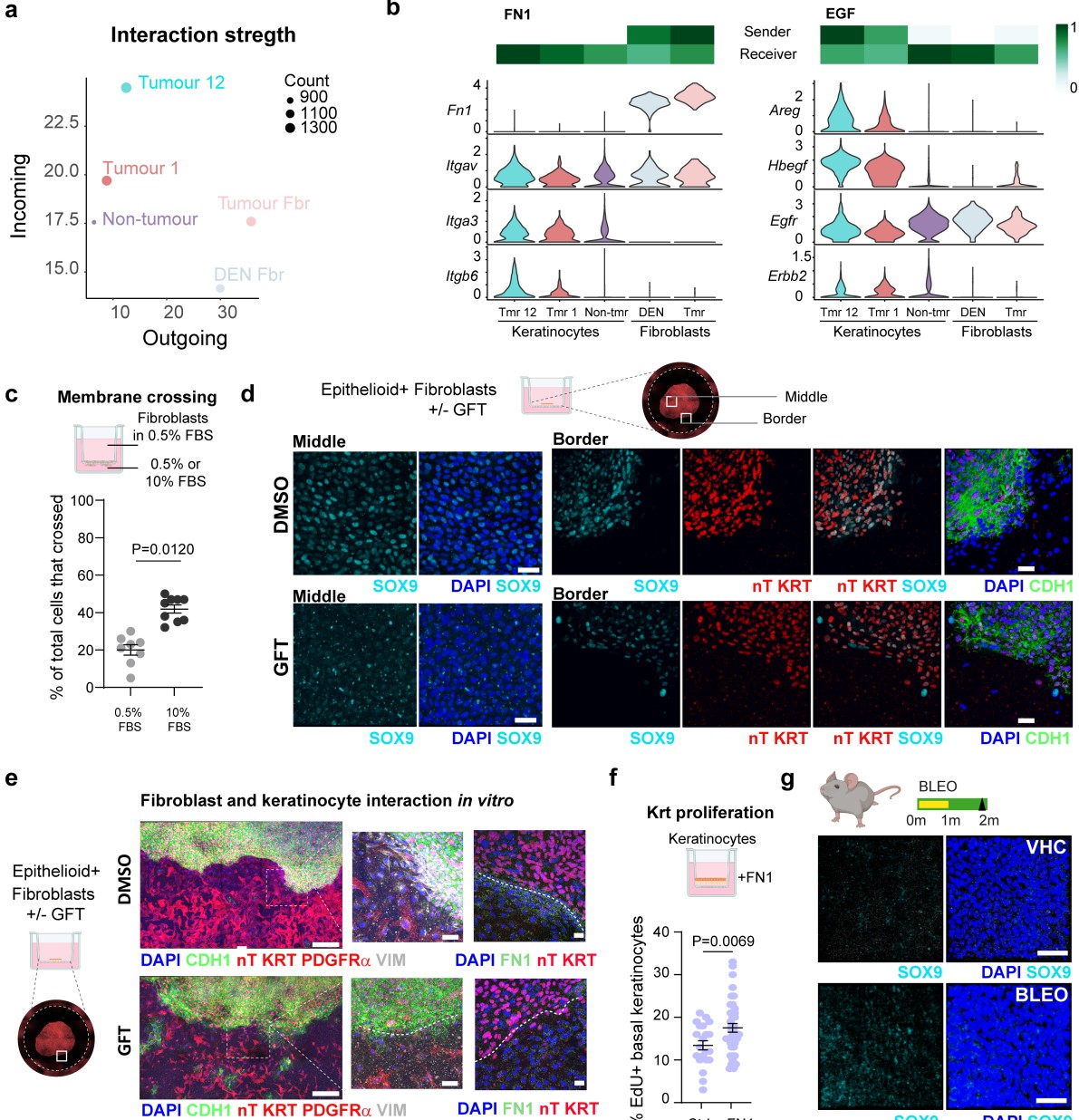

**Extended Data Fig. 10 | Tumour epithelial cells are associated with early tumour stromal remodelling via the EGF-SOX9-Fibronectin axis (Corresponds to Fig. 5). a**, Scatterplot displaying the dominant senders and receivers as identified by CellChat. Circle size represents 'communication probabilities'. **b**, Heatmaps (top) showing the relative importance of each cell group based on the computed network centrality scores for the Fibronectin (FN1) and EGF signalling network, respectively. Violin plots of relevant ligands and receptors across different cell types (bottom). These include expression of fibronectin (*Fn1*) and fibronectin-binding receptors (*Itga3, Itgav, Itgb6*), as well as EGF ligands (*Areg, Hbegf*) and binding receptors (*Egfr, Erbb2*). **c**, Migration of fibroblast through cell culture insert membrane (cartoon, top) in low (0.5%) or high (10%) serum (FBS) conditions was quantified (bottom) and expressed as a percentage of the total number of cells. Data expressed as mean±s.e.m. Each dot represents a technical replicate; data 4 biological replicates. One way Welch's ANOVA test with multiple comparison was used to assess significance between groups presented in the main Fig. 4e. **d**, schematic representation of regions imaged in epithelioid cultures from (top) and representative confocal images showing SOX9 (cyan) downregulation in keratinocytes in central epithelioid areas as well as on the border of GFT-treated epithelioids.

Representative images of the DMSO (ctrl) condition shown for comparison. Nuclear tdTomato (nT) keratinocytes (krt) (red); CDH1 (green); DAPI (blue). Scale bar, 50 µm. **e**, Schematic representation of an experiment (left) and representative confocal images (right) of expanding epithelioid cultures exposed to fibroblasts under DMSO (control) or Gefitinib (GFT) conditions. Vimentin, VIM (grey); PDGFRα (red); nuclear Tomato (nT) keratinocytes (krt) (red); DAPI (blue), E-Cadherin, CDH1 (green); Fibronectin, FN1 (green). Scale bar 500 µm (inset 50 µm). Corresponds to Fig. 4g. **f**, Top, experimental schematics, bottom, quantification of EdU incorporation (2 h chase) in confluent 3D epithelial cultures (Epithelioids) exposed to Fibronectin for 24 h. Data expressed as a percentage of the total number of cells (mean±s.e.m). Each dot represents a technical replicate; data from 3 biological replicates. Two-tailed Welch's t-test comparing EdU incorporation in control (Ctrl) and fibronectin-treated cells. **g**, Schematic representation of Bleomycin (BLEO) administration to mice (top) and representative confocal images (bottom) of SOX9 (cyan) expression in epithelia of bleomycin treated animals. DAPI (blue). Scale bars, 50 µm. N = 5 (control) 4 (BLEO) mice. Illustrations in **c,d,e,f,g** were created in BioRender; Alcolea, M. https://BioRender.com/nm3lx8y (2026). Source data (**c**, **f**).

# Reporting Summary

## Statistics

For all statistical analyses, confirm that the following items are present in the figure legend, table legend, main text, or Methods section.

| n/a | Confirmed | |
|---|---|---|
| ☐ | ☒ | The exact sample size (*n*) for each experimental group/condition, given as a discrete number and unit of measurement |
| ☐ | ☒ | A statement on whether measurements were taken from distinct samples or whether the same sample was measured repeatedly |
| ☐ | ☒ | The statistical test(s) used AND whether they are one- or two-sided<br>*Only common tests should be described solely by name; describe more complex techniques in the Methods section.* |
| ☐ | ☒ | A description of all covariates tested |
| ☐ | ☒ | A description of any assumptions or corrections, such as tests of normality and adjustment for multiple comparisons |
| ☐ | ☒ | A full description of the statistical parameters including central tendency (e.g. means) or other basic estimates (e.g. regression coefficient) AND variation (e.g. standard deviation) or associated estimates of uncertainty (e.g. confidence intervals) |
| ☐ | ☒ | For null hypothesis testing, the test statistic (e.g. *F*, *t*, *r*) with confidence intervals, effect sizes, degrees of freedom and *P* value noted<br>*Give P values as exact values whenever suitable.* |
| ☒ | ☐ | For Bayesian analysis, information on the choice of priors and Markov chain Monte Carlo settings |
| ☐ | ☒ | For hierarchical and complex designs, identification of the appropriate level for tests and full reporting of outcomes |
| ☒ | ☐ | Estimates of effect sizes (e.g. Cohen's *d*, Pearson's *r*), indicating how they were calculated |

*Our web collection on statistics for biologists contains articles on many of the points above.*

## Software and code

Policy information about availability of computer code

| Data collection | LAS X 4.7.0.28176 and 3.5.5.19976<br>Volocity 5.5.5, PerkinElmer<br>Volocity 7, Quorum technologies<br>Zen 3.2, Zeiss<br>Arivis 3.5.1, Zeiss<br>Fiji (2.0.0) |
|---|---|
| Data analysis | Confocal images were obtained and analysed using LAS X 4.7.0.28176 and 3.5.5.19976 and Zen3.2, Zeiss softwares. Further image analysis was performed using Volocity 5.5.5, PerkinElmer; Volocity 7, Quorum technologies; Arivis 3.5.1, Zeiss and Imaris Viewer 10.1.1 Cellranger (v7.0.1);<br><br>Mapping scRNAseq data to reference genomeSeurat (v5.0.3); Processing counts matrix and downstream analysisScrublet (v0.2.3); Doublet removalHarmony (v1.2.3); Integration and batch effect correctionSeuratWrappers (v0.4.0); Used to convert seurat object to cell_data_set (monocle 3)Monocle3 (v1.4.6); Pseudotime analysisCellChat (v2.2.0); Cell-Cell Communication analysesEnhancedVolcano (v1.24.0); Volcano PlotsGgplot2 (v3.5.2); PlotsGSEABase (v1.68.0); Gene list format correction for AUCellAUCell (v1.28.0); Scoring cells based on gene setsCowplot (v1.2.0); PlotsComplexHeatmaps (v2.22.0): Heatmaps.<br><br>The analysis code is available on the AlcoleaLab GitHub page https://doi.org/10.5281/zenodo.17802564 |

DNA sequencing analysis was performed using BWA-mem (0.7.17); SAMtools (v1.11); deepSNV R package (ShearwaterML; version 1.21.3); VAGrENT (3.3.0).

Statistical analysis was performed using GraphPad Prism (10.5.0)

For manuscripts utilizing custom algorithms or software that are central to the research but not yet described in published literature, software must be made available to editors and reviewers. We strongly encourage code deposition in a community repository (e.g. GitHub). See the Nature Portfolio guidelines for submitting code & software for further information.

## Data

Policy information about availability of data

All manuscripts must include a data availability statement. This statement should provide the following information, where applicable:
- Accession codes, unique identifiers, or web links for publicly available datasets
- A description of any restrictions on data availability
- For clinical datasets or third party data, please ensure that the statement adheres to our policy

Mouse reference genomes GRCm38 and mm10 2020-A were used. The single cell RNA sequencing data generated in this study have been deposited in the Gene Expression Omnibus (GEO) repository under accession code GSE271962. The DNA sequencing dataset was deposited at the European nucleotide Archive (ENA) under dataset accession number ERP134942. Source data are provided with this paper.
The analysis code is available on the AlcoleaLab GitHub page https://doi.org/10.5281/zenodo.17802564

## Research involving human participants, their data, or biological material

Policy information about studies with human participants or human data. See also policy information about sex, gender (identity/presentation), and sexual orientation and race, ethnicity and racism.

| Reporting on sex and gender | Sex and gender were not considered in this study. |
|---|---|
| Reporting on race, ethnicity, or other socially relevant groupings | Race, ethnicity or other social group was not considered in this study. |
| Population characteristics | To enable the recruitment of as many as possible early stage squamous cell carcinoma tumour samples from patients >18 years old were recruited from tissue bank. Shortly, the T1A and T1B stage chemo-naive surgical tumour samples were donated by patients who had undergone surgery at the Clinic for Visceral, Thoracic and Vascular Surgery at TU Dresden or at the Medical Department I, of the Carl Gustav Carus University Hospital. The corresponding FFPE material (tumour and normal tissue) from a total of ten characterised oesophageal squamous cell carcinomas was selected from the archive of the Institute of Pathology of the University Hospital Carl Gustav Carus (EK 59032007) by the Tumour and Normal Tissue Bank (TNTB) Dresden. Tis and ypT0 (prior to chemo T2N0M0) stage surgical tumour samples were donated by patients who had undergone surgery at Guy's and St Thomas' (London, UK) and Addenbrooke's Hospitals (Cambridge, UK), respectively. |
| Recruitment | Patient information sheet and signed consent were used to recruit prospective cases. Prospective samples for this study were collected and processed in accordance with local and national regulations with written consent and transferred using methods fully compliant with the requirements of the Human Tissue Act and the Human Tissue Authority. |
| Ethics oversight | The T1A and T1B stage chemo-naive surgical tumour samples were donated by patients who had undergone surgery at the Clinic for Visceral, Thoracic and Vascular Surgery at TU Dresden or at the Medical Department I, of the Carl Gustav Carus University Hospital. Macroscopically normal samples adjacent to the proximal resection margin were sampled from cancer resection specimens. The corresponding FFPE material (tumour and normal tissue) from a total of ten characterised oesophageal squamous cell carcinomas was selected from the archive of the Institute of Pathology of the University Hospital Carl Gustav Carus (EK 59032007) by the Tumour and Normal Tissue Bank (TNTB) Dresden. Studies presented in the manuscript involving early chemo-naïve human oesophageal tumour samples from Dresden were approved by the Ethics Committee of TU Dresden, Germany (Ref.: SR+BO-ff (Mono)-EK-161042025). Studies presenting chemo-naïve or post-chemo human oesophageal tumour samples from Guy's and St Thomas' (London, UK) and Addenbrooke's Hospital (Cambridge, UK), respectively were approved by East of Scotland Research Ethics approval committee (REC 18/ES/133). |

Note that full information on the approval of the study protocol must also be provided in the manuscript.

## Field-specific reporting

Please select the one below that is the best fit for your research. If you are not sure, read the appropriate sections before making your selection.

☒ Life sciences  ☐ Behavioural & social sciences  ☐ Ecological, evolutionary & environmental sciences

For a reference copy of the document with all sections, see nature.com/documents/nr-reporting-summary-flat.pdf

# Life sciences study design

All studies must disclose on these points even when the disclosure is negative.

| | |
|---|---|
| Sample size | Sample size was determined from pilot studies and previous published research studies as listed below: Colom, B. et al. Mutant clones in normal epithelium outcompete and eliminate emerging tumours. Nature 598, 510-514 (2021). https://doi.org:10.1038/s41586-021-03965-7 <br><br> Frede, J., Greulich, P., Nagy, T., Simons, B. D. & Jones, P. H. A single dividing cell population with imbalanced fate drives oesophageal tumour growth. Nat Cell Biol 18, 967-978 (2016). https://doi.org:10.1038/ncb3400 <br><br> A minimum of three independent mice or ex vivo cultures were used in all cases. For image analysis, a minimum of three independent samples were inspected and replicate images were taken per sample for data to be analysed both internally within said sample, and between at least 3 biological replicates. |
| Data exclusions | Exclusions were made to the data in the case of single-cell RNA sequencing analysis to exclude low quality cells as is considered best practice in such analyses. In this case, doublets were identified and removed using Scrublet, with per-sample doublet score thresholds between 0.1 and 0.2. Cells were also filtered on numbers of detected genes (nFeature_RNA) or total transcript counts (nCount_RNA) on a per-sample basis, with lower and upper limits ranging from 700–1,500 genes to 8000-10000 genes and 2,000–4,000 to 40000-160,000 transcripts per cell, respectively. Cells with more than 15% of reads mapping to mitochondrial genes were also excluded. These thresholds were determined individually for each library to account for differences in library quality. |
| Replication | A minimum of three independent mice or ex vivo cultures were used in all cases. All experiments were performed independently at least three times with similar results, unless otherwise stated. The reproducibility of all key findings was confirmed in at least three independent experiments conducted on different days and using independent biological samples. |
| Randomization | Mice and ex vivo cultures were randomly assigned to experimental groups. |
| Blinding | Blinding was performed for tumour count per condition and in vitro sample analyses by confocal microscopy. In cases where quantification was performed in tumours and the morphologically normal areas blinding was not possible due to differences in physical sample appearance. In ex vivo cultures where perturbation assay resulted in inhibition of fibroblast to keratinocyte interaction blinding was not possible due to differences in sample appearance. |

# Reporting for specific materials, systems and methods

We require information from authors about some types of materials, experimental systems and methods used in many studies. Here, indicate whether each material, system or method listed is relevant to your study. If you are not sure if a list item applies to your research, read the appropriate section before selecting a response.

## Materials & experimental systems

| n/a | Involved in the study |
|---|---|
| ☐ | ☒ Antibodies |
| ☒ | ☐ Eukaryotic cell lines |
| ☒ | ☐ Palaeontology and archaeology |
| ☐ | ☒ Animals and other organisms |
| ☒ | ☐ Clinical data |
| ☒ | ☐ Dual use research of concern |
| ☒ | ☐ Plants |

## Methods

| n/a | Involved in the study |
|---|---|
| ☒ | ☐ ChIP-seq |
| ☒ | ☐ Flow cytometry |
| ☒ | ☐ MRI-based neuroimaging |

## Antibodies

| | |
|---|---|
| Antibodies used | Primary antibodies <br> AREG; R&D Systems, Inc; AF989 <br> AREG; R&D Systems, Inc; AF262 <br> αSMA; Abcam; ab5694 <br> β-Catenin; Cell Signaling Technology; 9562 <br> CD31; Abcam; ab7388 <br> CD45; BioLegend; 103102 <br> FAP; Abcam; ab218164 <br> FINC; BD Biosciences; 610078 <br> K17; Proteintech; CL488-17516 <br> KRT6A; BioLegend; 905701 <br> KRT6A; Abcam; ab18586 <br> KI67; Abcam; ab16667 <br> PDGFRα; R&D Systems, Inc; AF1062 <br> PDGFRα; R&D Systems, Inc; AF307 |

S100A4 (FSP); Fisher Scientific Ltd; PA5-16586
SOX9; Millipore; AB5535
VIM; Abcam; ab194719
active YAP; Abcam; ab205270
EGR1, Abcam; ab300449
RUNX1, Proteintech; 25315-1-AP
Secondary antibodies
Goat IgG 647; Millipore; AP180SA6
Goat IgG 750; Abcam; ab175745
Mouse IgG 488; Invitrogen; A-21202
Mouse IgG 647; Invitrogen; A-31571
Mouse IgG 750; Abcam; ab175738
Rabbit IgG 488; Invitrogen; A-21206
Rabbit IgG 555; Invitrogen; A-31572
Rabbit IgG 647; Invitrogen; A-31573
Rabbit IgG 750; Abcam; ab175728
Rabbit IgG; Abcam; ab171870
Rat IgG 647; Abcam; ab150155
Rat IgG 750; Abcam; ab175750

| Validation | Well characterised antibodies were used in this study. All are commercially available and validated by the manufacturers. List below target molecule, host, company link<br><br>AREG, goat, https://www.rndsystems.com/products/mouse-amphiregulin-antibody_af989<br>AREG, goat https://www.rndsystems.com/products/human-amphiregulin-antibody-31221_mab262<br>A-SMA, rabbit, https://www.abcam.com/en-gb/products/primary-antibodies/alpha-smooth-muscle-actin-antibody-ab5694<br>β-Catenin, rabbit, https://www.cellsignal.com/products/primary-antibodies/b-catenin-antibody/9562<br>CD31, rat, https://www.abcam.com/en-gb/products/primary-antibodies/cd31-antibody-mec-746-ab7388<br>CD45, rat, https://www.biolegend.com/en-gb/products/purified-anti-mouse-cd45-antibody-102<br>FAP, rabbit, https://www.abcam.com/en-gb/products/primary-antibodies/fibroblast-activation-protein-alpha-antibody-ab218164<br>Fibronectin, mouse, https://www.bdbiosciences.com/en-gb/products/reagents/flow-cytometry-reagents/research-reagents/single-color-antibodies-ruo/purified-mouse-anti-fibronectin.610077<br>K17, rabbit, https://www.ptglab.com/products/Cytokeratin-17-Specific-Antibody-CL488-17516.htm<br>KRT6A, rabbit, https://www.biolegend.com/en-gb/products/purified-anti-mouse-keratin-6a-antibody-11459<br>KRT6A, mouse, https://www.abcam.com/en-us/products/primary-antibodies/cytokeratin-6-antibody-ks6ka12-ab18586<br>KI67, rabbit, https://www.abcam.com/en-gb/products/primary-antibodies/ki67-antibody-sp6-ab16667<br>PDGFRα, goat, https://www.rndsystems.com/products/mouse-pdgf-ralpha-antibody_af1062<br>PDGFRα, goat https://www.rndsystems.com/products/human-pdgf-ralpha-antibody_af-307-na<br>S100A4 (FSP); rabbit, https://documents.thermofisher.com/TFS-Assets/LSG/certificate/Certificates-of-Analysis/PA516586_SK2481855D.PDF<br>SOX9, rabbit, https://www.merckmillipore.com/GB/en/product/Anti-Sox9-Antibody,MM_NF-AB5535<br>VIM, rabbit, https://www.abcam.com/en-do/products/primary-antibodies/alexa-fluor-647-vimentin-antibody-epr3776-cytoskeleton-marker-ab194719<br>active YAP, rabbit, https://www.abcam.com/en-gb/products/primary-antibodies/active-yap1-antibody-epr19812-ab205270<br>EGR1, rabbit https://www.abcam.com/en-us/products/primary-antibodies/egr1-antibody-epr23981-46-ab300449<br>RNX1, rabbit https://www.ptglab.com/products/RUNX1-Antibody-25315-1-AP.htm?srsltid=AfmBOooRuq32Amnge4kba96D2QgywHdbscWO9HH4Fq8Z4-eEkOfggw_3 |
| --- | --- |

# Animals and other research organisms

Policy information about studies involving animals; ARRIVE guidelines recommended for reporting animal research, and Sex and Gender in Research

| Laboratory animals | Mouse, C57/bl6, transgenic, male and female, age 8 weeks to 18 months.<br>Unless otherwise specified, C57BL/6J mice (ordered from Charles River, UK; strain code 632) were used. Other mouse strains includes: cell cycle reporter line R26Fucci2aR (Fucci2a) kindly provided by Ian J. Jackson;  PDGFRαEGFP mice (stock #007669, Jackson Laboratory); Sox9flox/flox mice (obtained from MRC-Harwell, on behalf of the European Mouse Mutant Archive (https://www.infrafrontier.eu/)); K14-CreER mice (stock #005107, Jackson laboratory); R26mT-mG mice (mTmG; stock #007676, Jackson Laboratory); Col1a2CreER mice (strain #029567, Jackson Laboratory); R26FlConfetti mice (strain #017492, Jackson laboratory, kindly provided by Hans Clevers); R26nT-nG mice (nTnG, strain #023537, Jackson Laboratory); H2B-EGFP mice (CAG::H2B-EGFP; strain #006069, Jackson Laboratory); NOD scid gamma mice (NSG; NOD.Cg-Prkdcscid Il2rgtm1Wjl/SzJ; strain #005557, Jackson Laboratory). Pdgfra-CreERT mice ( stock#018280, Jackson Laboratory).<br><br>Mice were bred and maintained under specific-pathogen-free conditions at the Gurdon Institute and The Anne McLaren Building, University of Cambridge. All animals were housed between 20-24°C, 45-65% humidity and a 12-hour light-dark cycle. |
| --- | --- |
| Wild animals | No wild animals were used in this study |
| Reporting on sex | All experiments comprised a mixture of male and female mice with no gender-specific differences observed (unless specified otherwise). For RNA sequencing experiments, only male animals were used in order to avoid cofounding effects due to estrous cycle. |

| Field-collected samples | No field collected samples were used in this study |
|---|---|
| Ethics oversight | All experiments were approved by the local ethical review committees of the University of Cambridge and conducted according to Home Office project licenses PPL70/8866 and PP7037913 at the Gurdon Institute and The Anne McLaren Building, University of Cambridge. |

Note that full information on the approval of the study protocol must also be provided in the manuscript.

# Plants

| Seed stocks | *Report on the source of all seed stocks or other plant material used. If applicable, state the seed stock centre and catalogue number. If plant specimens were collected from the field, describe the collection location, date and sampling procedures.* |
|---|---|
| Novel plant genotypes | *Describe the methods by which all novel plant genotypes were produced. This includes those generated by transgenic approaches, gene editing, chemical/radiation-based mutagenesis and hybridization. For transgenic lines, describe the transformation method, the number of independent lines analyzed and the generation upon which experiments were performed. For gene-edited lines, describe the editor used, the endogenous sequence targeted for editing, the targeting guide RNA sequence (if applicable) and how the editor was applied.* |
| Authentication | *Describe any authentication procedures for each seed stock used or novel genotype generated. Describe any experiments used to assess the effect of a mutation and, where applicable, how potential secondary effects (e.g. second site T-DNA insertions, mosiacism, off-target gene editing) were examined.* |

