## [Peer Review file · Nature]

Pre-cancerous Niche Remodelling Dictates Nascent Tumour Persistence

Corresponding Author: Dr Maria Alcolea

Version 0:

Reviewer comments:

Referee #1

(Remarks to the Author)

Skrupskelyte et al. utilize a well-known carcinogen-induced esophageal squamous cell carcinoma model to investigate factors that influence the survival of early tumors. They start with a noteworthy observation: the majority of early dysplastic lesions fail to form tumors and do not survive. Their findings reveal that a stromal niche, which includes PDGFRA+ fibroblasts, is linked to tumor survival. The authors present numerous visually striking images to this effect. However, the manuscript's momentum diminishes significantly as the authors revisit well-established growth factor pathways between epithelial and mesenchymal cells. In addition to a lack of novelty, the authors describe single cell states of stromal populations in an outmoded manner without comparison to contemporary literature describing the heterogeneity of fibroblasts in particular, and stromal cells in general. Perhaps the most notable shortcoming of this study is that it almost completely overlooks the immune system's role in nascent tumor formation. Tumor immunosurveillance is a well described phenomenon in carcinogen-induced tumors, and this is not accounted for in the authors' study. I believe that this represents a fatal flaw in this study that I do not think could be corrected without a fundamental reframing of the investigation. I have enumerated specific points below.

1) The authors appear to purposely avoid a discussion of immunity. Tumor surveillance is known to be a critical element of the early tumor suppression response. The authors' fundamental question is why do these nascent tumors disappear. They strongly imply that this is due to cell/niche stress. But it is almost certainly in fact simply tumor surveillance. How do they know that the immune system is not what wipes out these early tumors?

2) That fibroblast supply growth factors for epithelial cells is not novel. And it is not surprising that a perturbed stromal microenvironment would have different properties than one that is unperturbed. Do the authors observe similar "tumor like" properties if they repeat these experiments using other stimuli, such as bleomycin-induced fibrosis or exogenous wounding?

3) Supplemental Table 3 is informative and very appreciated. However, it does seem to suggest that only three total samples were submitted for 10X processing? If this is not the case, then this point should be made very clear and within/across group comparisons performed. Otherwise, these should be repeated with at least n = 3 per group. I assume samples were pooled in order to hit the minimum cell number targets, but this can be overcome through HTO-barcoding such that cells are labeled based on their tissue/tumor of origin prior to pooling and processing at the correct scale. For a study of this magnitude, I think multiple biological replicates are critical in order to ensure reproducibility.

4) Supplemental Table 4 is inadequate and calls into serious question the authors' judgement in assigning cell types to each cluster, which is not well-described. There are numerous methods for automated classification, such as SingleR and scATOMIC, among others, but no such methods are discussed. Furthermore, based on the genes listed in Supp Table 4, it appears that the authors used a small number of "canonical" marker genes to make these decisions, which is an outdated way of thinking of scRNA-seq data. For example, they defined "myofibroblast" solely by ACTA2... despite this also being a marker for what the authors call "Myocytes", among others. It is also unclear why "myofibroblast" is a different category than "myofibroblastS". This is all highly problematic and undermines faith in the sophistication of the authors in this field. It is also not clear how the authors moved from 38 initial clusters to their final 11 "known biological cell types". Both cluster

consolidation and cell type ascription should be more methodical and described in better detail. At the very least, a complete list of DEGs for the final cell types should be provided.

5) Similarly, the three “states” ascribed based on pseudotime analysis are not convincing. Although these states themselves are described well in Supp Table 6-8, how they were arrived at, using what cutoffs, and how the number (three) was determined is not clear. In the absence of a principled heuristic, it is not clear that correction for multiple hypothesis generation would make these findings non-significant relative to those potentially available.

6) The decision to use Monocle 2, instead of Monocle 3, is also unusual, since the more recent release has been available for nearly 4 years and fixes many of the over-conclusions associated with the earlier deployment. I would ask that the reviewers repeat their analysis using the more sophisticated version of this pseudotime package.

7) The rationale for drawing distinctions around Sox9 and Krt6a is not justified in the current manuscript. Are there clearly bimodal distributions for each? Where and how were the cut points drawn? Again, this way of conceptualizing NGS data in terms of small buckets based on arbitrarily binned expression of a minority of genes is not consistent with the current understanding of transcriptomics in cellular biology. This is not FACS, where surface protein expression may be accurately assessed, but rather probabilistic sampling where measured expression of the same RNA molecule from the same cell will vary considerably over time.

8) Again, I fundamentally disagree with the manner in which the authors conceive of their cell subpopulations. For a manuscript aimed at studying the relationships between the epithelium (tumor and non-tumor) and the stromal microenvironment, there is no main-text mention of the three widely accepted “canonical” fibroblast type subtypes (iCAF, myCAF, apCAF), and these are only buried in EDF 8b-c without explanation or context. A closer interpretation based on the literature (PMC6727976, PMC5339682, etc) would be helpful.

9) In Figure 4a, instead of referring to fibroblasts based on their cluster number and relative PDGFRA expression, it would be more helpful to examine canonical iCAF markers such as IL6, CXCL12; myCAF markers such as LRRC15, ACTA2; and apCAF markers such as CD74.

10) Figure 3h, or its associated supplements, should contain representative control images.

11) I question the authors' judgement in distinguishing between high and low expression of the continuative fibroblast marker PDGFRA. Even if so, I think it is presumptuous to assume that proximity begets causal interactions.

12) For the fibroblast enrichment in Figure 4e, the authors should examine at a per-sample level the percent expression of their “PDGFRA-low” clusters and assess statistical significance.

13) The authors talk about going from 38 initial clusters to 11 final named clusters, but then on the second paragraph of Page 11 (“For this we focused on cluster 23 (C23)), they appear to be referencing back to the 38 cluster model? Is this correct? If so, they should provide some reference to where these cells are. Which of the final 11 cell types do these fall into?

14) The CellChat analysis adds almost nothing to the manuscript and is greatly over-concluded. This statement is incredibly misleading and should be removed: “Overall, our scRNA-seq and communication analyses uncovered the pro-fibrotic/wound-healing transcriptional profile characteristic of tumour niche fibroblasts”. The CellChat methods also contain insufficient detail.

15) The in vivo knockout and inhibition experiments are probably the most compelling portions of this manuscript, but are sparsely covered both in terms of text and figure panels. Additional description of these very interesting results would be warranted.

Referee #2

(Remarks to the Author)

This manuscript by Skrupskelyte et al aims to better understand how the pre cancer niche provides initial steps for tumor formation and contributes to its progression and survival. The authors use an elegant way to induce and track pre-neoplastic squamous tumors in the upper gastrointestinal mouse tract utilizing different imaging techniques such as confocal microscopy and SHG, combined with scRNA-seq and 3D cultures. They identify niche+ tumor structures that are enriched with stromal remodeling fibroblasts which affect keratinocyte proliferation and promote long term survival. Transcriptomic analysis identified a Sox9high cancer cell state that promotes fibroblast migration by EGF signaling, leading to production of fibronectin by the fibroblasts.

The model used is interesting and the notion of studying early events is important. It is not clear at all though if this has significant human relevance since there is no human validation. Moreover, the authors claim that the cells they study are

early CAFs but do not express classic CAF genes when in fact they do express what would appear to be a myCAF signature. It is true that the marker genes are not the classic. However, this could again be a consequence of the model being a model without human validation. Moreover, though the authors collected different tumor niches within each GI tract, the number of biological replicates (i.e. different mice) along the manuscript is relatively low (n=4). The authors must add human validation of at least the Sox9, AREG and FN connection, as well as their subset markers. Without these the observations are interesting, but the novelty is limited, since it is already well established that tumor development is modulated and perhaps even initiated by changes in the tumor microenvironment and specifically by fibroblasts (Sahai et al, 2020, de Visser and Joyce, 2023).

Major comments

Major comments:

1. It is well known that the TME changes at early stages. The changes shown here are still generated by the carcinogen and cancer-stroma signaling so not totally surprising or novel. Would these occur without cancer cells? Is the stroma different before the cancer cells are there? It is very hard to tell but maybe treating the stromal matrices with DEN would provide some answer.

1. Following up on this, in figure 3 the authors show that exposure to tumor stroma makes normal epithelial cells proliferate. This is not surprising and has been shown numerous times already. Would it happen similarly though from niche+ vs niche-stroma? Or when immune cells are depleted? In Figure 3o,p- The authors used mouse model to deplete Sox9 keratinocytes and measured the number of tumors 1-month post DEN treatment. On which tumors did you look at? Given the number of niche+ tumors is kept stable and low along time in DEN treated mice (Figure 1d), In order to prove Sox9 is a relevant player in epithelial-stromal communication from the most incipient stages of tumorigenesis, the authors should measure the number of niche+ tumors and niche- tumors separately, instead of measuring the number of all tumors together. Otherwise, the authors can't exclude the fact that the reduction in tumor burden originates from the decrease in niche- tumors as shown in figure 1d.

2. Much of the analysis is based on 3-4 mice per group. The experiments in Figure 1, for example, show dozens of dots in panel g but these are all from 3 mice. The authors must take into consideration the fact that different tumors in the same mouse are expected to be more similar or co-dependent than tumors from different mice.

3. The Sox9 connection is very interesting, but it is not clear what made the authors choose this factor originally. Was it one of the most DE genes? Or educated guess? Would other cancer genes also show this close to stroma pattern? Also, the authors say that Sox9 is high close to stroma but in fact most cells in fig 3m are Sox9 negative. Once again, the statistical analysis in 3n relies on many tumors albeit in very few mice.

4. How is niche+ vs niche- defined? A threshold of expression of specific factor(s)? This is not defined. Moreover, the authors use PDGFR-alpha to define fibroblasts in Figures 1-3, but then in the scRNA-seq analysis they show that in fact there is a PDGFR-alpha high and low group. So how can this be used as a pen marker for fibroblasts in the first figures?

5. The authors compare the niche fibroblasts to known CAF signatures and find very little overlap. Did they try to look at full blown tumors in their model? Or in other mouse models of esophageal cancer? It is very much possible that the specific markers here are different and that these are in fact myCAFs as also appears from the ligand-receptor analysis.

6. In the last paragraph of the results the authors claim that Sox9 exerts its effect through EGFR signaling. But they do not make the experimental link between Sox9, EGFR/AREG and FN. Is this signaling altered in the Sox9 KO mice? This should be tested in mice, and, importantly, verified in human samples. Specifically, the authors should test whether SOX9 expression is found in early tumor developmental stages or has any survival advantage.

7. Figure 4- the authors use Dvorak's theory to describe the pre cancer niche as the wound that never heals. However, they do not relate to the immune compartment that plays an integral role in wound healing processes. Supp figure 2 shows the presence of CD45 cells in nascent tumor niche 10 days post-DEN. Are there differences in immune cell composition between niche+ and niche- tumors?

Minor comments

1. Page 5: "However, at these early stages, the presence of this structure was largely heterogeneous, with the majority of epithelial lesions (~70%, 199 lesions out of 296 per esophagus, on average) showing no apparent stromal reorganization (Fig. 1c)" - where is this data shown? The authors should present this quantification.

2. Supp figure 2c-d - how did you calculate the proportion of endothelial cells and immune cells separately? They are labeled with the same fluorophore.

3. Supp figure 2e- what is the sign of invasion? What does ITAG6 mark?

4. Figure 3h - the authors don't show the expression of SOX9 in healthy control tissue but state in the text it is not expressed as opposed to nascent tumors.

5. "For this we focused on cluster 23 (C23) fibroblasts, a Pdgfralow cluster enriched in tumour cells (Fig. 4e)." I guess the authors mean stromal cells from tumors. The wording is confusing.

6. Ext Fig 1b - the significant p values for the different time points - what are they pointing to? There is only one group - what are they comparing to?

(Remarks to the Author)

Study presented by Skrupskelyte et al. aims to identify environmental changes underlying the transition of healthy epithelium towards cancer. The authors used a chemical carcinogen (DEN) to introduce genetic mutations in the squamous esophagus of mice and their observed two types of precancerous lesions. Some were surrounded by nest-like (niche+) structures composed of stromal fibroblasts. Niche+ predominantly developed into cancerous lesions. Next, using heterotypic cultures, they show that cell growth of healthy epithelia is affected by the underlying niche+ stroma. These cells can be more frequently transplanted into nude mice when exposed to niche+ stroma. Next, they performed scRNA-seq on the esophagus and squamous forestomach and investigated transcriptional patterns of the epithelium and stromal compartments. They observed a shift towards proliferating cells in the epithelium and the emergence of Sox9 cells as the disease progressed. In the stromal compartment, they identified Pdgfra low cells as potential drivers of niche+ phenotype. Subsequent cell-to-cell communication analysis identified fibronectin (FN) and EGF signalling pathways as drivers of stroma-to-epithelium interaction in the niche+ environment. They finally demonstrate that Sox9 high keratinocytes can communicate with fibroblasts using EGF induced signaling molecules and that fibroblasts signal to keratinocytes using fibronectin pathways. In general, this study aims to describe a novel mechanism of oesophageal cancer development with a strong focus on the role of microenvironment modulation during the initiation of this cancer. The elucidation of the nongenetic factors driving cancer initiation is of out-most importance and this study offers an important view on this process. The authors' use of chemical carcinogen that recapitulate normal oncogenesis is commendable. The authors further provide ample supplementary information. However, the study has technical and conceptual limitations:

1. First of all, in their single cells analysis of fibroblast, the authors avoid detailed analysis of individual clusters. It is very apparent from their analysis that the loss of cluster 8 (fig. 4a, S4e) is the most dramatic change between DEN and Ctrl samples. These cells are not well characterised by the authors and additional information on the identify of these cells (e.g. table of cluster markers) is needed.
2. Secondly, cluster 15 is also largely ignored in the course of authors analysis. These fibroblast in comparison to other fibroblast clusters, see to expand after DEN treatment and in the tumour samples. The presence of acta2 and other fibroblast markers (Fig. S4g) indicates that these fibroblasts might be contractile in nature.
3. Although the authors provide a limited set of markers per cell cluster, it appears that fibroblast studies here resemble populations S1, S2 and S3 previously annotated by Davidson et al (<https://doi.org/10.1016/j.celrep.2020.107628>). The studies are different enough (here authors investigate cancer initiation rather than injectable model) however the authors should investigate in detail the shift of clusters 8 and 15, as they resemble studies of Davidson et al and other CAF studies (<https://doi.org/10.1038/s41568-019-0238-1>).
4. The choice of Col1A2 is not obvious as a driver gene for the Pdgfra-low fibroblast is poorly justified. This gene seems to be expressed in all fibroblast clusters (S4g) and a potential flow/transcommitment of cells between compartments cannot be excluded. The authors, in addition to pdgfra-low, should also lineage-trace other fibroblast populations (especially clusters 8 and 15) to understand if these cells do not contribute to the niche+ phenotype. If only cells from cluster 23 were responsible for the phenotypes observed, lineage training using other markers should not result in the clonal clustering described by the authors.
5. Similarly, the trajectory analysis of the epithelial compartment lacks detail. The authors often switch between projections (monocle components and umap) and use different markers on different embedding types. This makes the analysis difficult to follow. It would be helpful if authors provided projects of all cell types and tissue types onto monocle projects and also the projection of states (1-3) on the Umap projects. Also, vector information in the analysis is missing. RNA velocity could provide information about the direction of cell trajectory. It is not clear to me if the cells flow from sates 3 to 2 or vice versa.
6. The validation of Sox9 genotype (figure 3o-p) only focuses on tumour burden in this model. The authors did not use this metric in any other setting in the study and additional analysis of these mice is warranted.
7. In Figure 1e, niche- cluster of cells appears significantly smaller and there is some proliferation in its nuclei (the KRT6A signal seems to be overexposed). What proportion of nuclei in the niche+/niche- regions took up EdU? Figure S2g only has counts which might be consequences of cellular density rather than higher proliferation rate. As these are representative images, it would be helpful if the authors chose regions of similar size to avoid optical illusions.
8. SOX9 staining on Figure 3k and 3m has a lot of background and it is very difficult to distinguish real cells/nuclei from the background
9. Until figure 4, authors imply that all fibroblasts are pdgfra+ however, starting in figure 4, they show fibroblasts in submucosa are actually pdgfra-high and the cells in lamina propria are pdgfra-low. This change seems to be very weak in control samples in Figure 4c but much stronger in the tumour. It would be helpful if the authors provided larger fields of view on the earlier figures to understand that pdgfra expression is not constant in all fibroblasts. Also, the figure 4c, there seems to be a continuum of fibroblasts in the normal samples, but in the tumour samples, there is a clear boundary between compartments. The authors could also provide larger fields of view to ensure that the reader can clearly see these transitions.
10. Multiple figures lack correct annotation of markers displayed. E.g. figure 1b: if the signal from multiple channels is separated then it should be done for every channel (there should be only KRT6A channel in addition to PDGFRA only channel). The PDGFRA only channel is missing annotation. The "Niche+" and "8m" labels do not indicate that these images are the same projection and all metadata is shared. This low-quality image annotation is pervasive in the figures (figure 4c tumour samples is shown twice, one with pdgfra shown in grayscale and once with this signal missing. This is not obvious from the legend or figure annotation. Similar pattern can be found on 1b, 1c, 2g, 3k, 3m, 3h, 5g, 5h .

Version 2:

Reviewer comments:

Referee #1

(Remarks to the Author)

The additional experiments and revised analyses (confirmation of phenotype in immune deficient mice, added immune profiling, Monocle3 trajectories, CellTypist/label transfer annotations, expanded functional work with EGFR inhibition and SOX9 loss, and updated statistical analysis) substantially strengthen the manuscript. My concerns have been addressed to a degree that supports acceptance in principle.

I do want to re-emphasize the importance of fibroblast phenotypic analysis in a contemporary context. The claim that fibroblast phenotypes might be specific to esophageal cancer is not supported by current literature and again raises some concern as to the authors' expertise in fibroblast biology (Gao, Li, Cheng et al., Cancer Cell, 2024 and Liu, Cui, Han et al., Sci Adv, 2025). It appears that the authors are dealing with subtle variations of a Pi16-positive population (originally described in Buechler et al., Nature, 2021 as a stemlike, or steady-state phenotype), although it is difficult to be certain. The authors have shown that the targeted fibroblast phenotype does not represent an activated myCAF or iCAF state, and we appreciate this comparison.

Our final request is that the authors annotate fibroblast phenotypes in the context of one of these cross-tissue studies. This will be important to discern whether the fibroblast phenotype in question is a considerable departure from the well described quiescent (Pi16) phenotype.

Referee #2

(Remarks to the Author)

The authors have thoroughly addressed my comments.

One minor point - in Figure 3b, the authors quantified fibroblasts in Niche⁺ vs. Niche⁻ and state that fibroblasts are enriched in Niche⁺. However, the figure appears to show enrichment in Niche⁻ instead. Could it be typo?

Referee #3

(Remarks to the Author)

The new manuscript submitted by the authors addresses the previous shortcomings very well. As urged by myself and other reviewers, the authors have introduced significant changes to their scRNA-seq analysis, added new mouse models, validated fibroblast-to-epithelial signaling discoveries, and provided a much-improved narrative that is no longer confounded by difficult-to-follow observations unrelated to the main findings. The revised manuscript elegantly demonstrates that fibroblast populations depositing matrix fibers support the survival of Niche⁺ DEN-induced precancerous lesions in their model. Furthermore, they present data supporting the role of the EGF-SOX9-FN1 axis as an important driver of Niche⁺ cell survival and selection.

As requested, the authors also demonstrate that, at least during the early stages of Niche⁺ tumor development, the immune system does not play a major role—highlighting the importance of the fibroblast population. I am not surprised that the observed fibroblast phenotypes do not resemble mature CAFs, as such cells are typically present only in tumor tissue and not in precancerous lesions.

At this point, I have only a minor comment that should not detract from the importance of this study and its significant contribution to tumorigenesis research:

It is not entirely clear to me how the Pdgfra-CreERT; R26⁺FIConfetti/WT system works. A schematic of the Rosa26 locus would be helpful. I assume that the system includes a STOP codon upstream of Confetti that is removed upon TAM administration; hence, fibroblasts expressing lower levels of Pdgfra would be expected to exhibit lower recombination efficiency and fewer fluorescent markers (as opposed to the observed phenotype). This discrepancy should be mentioned in the text.

Version 3:

Reviewer comments:

Referee #1

(Remarks to the Author)

The authors have addressed the reviewer's comments and revised the manuscript accordingly. I have no additional concerns.

Authors' point-by-point Rebuttal and associated Experimental Points (Nature manuscript 2024-07-13618)

Skrupskelyte et. al. titled "Pre-cancerous Niche Remodelling Dictates Nascent Tumour Survival"

Text from reviewers' comments presented in blue, our responses to reviewers' questions in black, references to text in the manuscript in amber and proposed experimental plan/actions points in green. References shown as hyperlinks.

Referee #1 (Remarks to the Author):

Skrupskelyte et al. utilize a well-known carcinogen-induced esophageal squamous cell carcinoma model to investigate factors that influence the survival of early tumors. They start with a noteworthy observation: the majority of early dysplastic lesions fail to form tumors and do not survive. Their findings reveal that a stromal niche, which includes PDGFR α + fibroblasts, is linked to tumor survival. The authors present numerous visually striking images to this effect. However, the manuscript's momentum diminishes significantly as the authors revisit well-established growth factor pathways between epithelial and mesenchymal cells. In addition to a lack of novelty, the authors describe single cell states of stromal populations in an outmoded manner without comparison to contemporary literature describing the heterogeneity of fibroblasts in particular, and stromal cells in general. Perhaps the most notable shortcoming of this study is that it almost completely overlooks the immune system's role in nascent tumor formation. Tumor immunosurveillance is a well described phenomenon in carcinogen-induced tumors, and this is not accounted for in the authors' study. I believe that this represents a fatal flaw in this study that I do not think could be corrected without a fundamental reframing of the investigation. I have enumerated specific points below.

1) The authors appear to purposely avoid a discussion of immunity. Tumor surveillance is known to be a critical element of the early tumor suppression response. The authors' fundamental question is why do these nascent tumors disappear. They strongly imply that this is due to cell/niche stress. But it is almost certainly in fact simply tumor surveillance. How do they know that the immune system is not what wipes out these early tumors?

Reviewer #1 Authors' response: We would like to clarify that the fundamental question of our study is not “**why do these nascent tumors disappear**” as stated by the reviewer. This intriguing question represented the focus of our previous collaborative work (Colom et al. Nature 2021). In the current manuscript, instead, we study a distinct open question: what are the unique features of the few tumours that are not outcompeted from the tissue and that succeed in persisting long-term? In other words, our focus is on the surviving tumours that remain capable of progressing further towards advanced stages. We alluded to this in the original version of the manuscript: “**Previous studies using an oesophageal early-tumour model demonstrated that not all nascent tumours have the same chance of survival, with most being cleared from the tissue soon after formation through competition with neighbouring mutant clones. [...] Here, we [...] study the unique features of the few tumours that manage to escape the pre-neoplastic tumour clearing process in the upper gastrointestinal tract.**” We will make sure to emphasize this point in a revised version of the manuscript.

Regarding the concern about the immune system, we recognise that we did not elaborate on the immune component and sincerely apologise for the lack of clarity on this. We completely agree with the reviewer that the immune compartment is a critical player contributing to carcinogenesis. However, this aspect was explored in depth in our previous collaborative work (Colom et al., Nature 2021). There, as indicated above, we used the same model presented here (the DEN model) to study the mechanisms involved in the disappearance of nascent tumours. Our results demonstrated that most incipient tumours were rapidly lost with no indication of an anti-tumour immune response, including **no differences in early tumour**

persistence in immunodeficient mice (Extended data Figure 6 from Colom et.al. 2021). Instead, early tumour clearance was shown to be largely driven by the presence of neighbouring mutant clones with a competitive advantage. In light of these findings, in the current study, we decided to limit our focus on the contribution of immune cells.

Action points:

- To address this concern, we will include scRNA-seq data on the interactions between tumour and immune cells in a revised version of the manuscript.
- Experimentally, we plan to perform additional DEN carcinogen experiments in immunodeficient mice to explore whether the niche+/- phenotype is altered.
- The new manuscript will also contain new data revealing that, at day 10 post-DEN, the immune infiltrate in niche+ and niche- tumours show no significant differences, reinforcing our previous observations (Colom et al., Nature 2021). See preliminary results in Rebuttal Figure 1 (below).

Rebuttal Figure 1: Immune cell surveillance in nascent niche- and niche+ tumours. **a**, Representative confocal images showing the distribution of CD45⁺ cells (immune, cyan) in niche- and niche+ tumours (KRT6A, red) 10d post-DEN. PDGFR α , greyscale; DAPI, blue; scale bar, 10 μ m. **b**, Number of immune cells per 1000 μ m² in control, niche- and niche+ tissues; only the epithelial and lamina propria layers were considered. Data is presented as mean+s.e.m. A non-parametric Kruskal-Wallis test was used to assess significance, n=3.

2) That fibroblast supply growth factors for epithelial cells is not novel. And it is not surprising that a perturbed stromal microenvironment would have different properties than one that is unperturbed. Do the authors observe similar “tumor like” properties if they repeat these experiments using other stimuli, such as bleomycin-induced fibrosis or exogenous wounding?

There is no doubt that epithelial cells and fibroblasts communicate with each other in response to tissue perturbations (such as during wound healing, cancer and in vitro conditions (PMIDs: 15549095, 30737373, 1052771) in an attempt to restore tissue integrity. And, although there are physiological studies showing active epithelial-stromal interactions in early dysplastic tumours (PMIDs: 20080664, 20138012), no previous report has investigated the contribution of the mesenchymal compartment during tumour inception when tumours are in their nascent stages and only ~10 cells in size. In this sense, our study is, to our knowledge, the first one to use a holistic mutagen approach to dissect the contribution of mesenchymal cells to tumour heterogeneity and survival at nascent pre-neoplastic stages.

Indeed, as eluded by the reviewer, in our study we show that, upon lesion formation, the initial epithelial-mesenchymal interactions closely resemble those of an injury response. These include stromal fibronectin enrichment (PMID: 7240787), epithelia migration and hyperproliferation (PMID: 26174765), EGF ligand secretion (PMIDs: 8483908, 15482488), as well as the role of SOX9 in epithelial-mesenchymal communication (PMID: 38386758), among others. However, given that fibroblasts in non-tumour areas do not show the features of those forming the tumour niche, we reason that the observed fibroblast changes are unlikely to be the direct result of the stress induced by the carcinogen treatment. This suggests that distinctive mechanisms govern cell behaviour in the areas where tumour lesions emerge. Further reinforcing this notion, we also show that, unlike epithelia cells, the mutational burden of fibroblasts is not affected by our carcinogen treatment (Extended Data Fig. 8i). We will revise our manuscript and include new data to highlight this point.

Action points:

- **We propose to revise our scRNA-seq analysis to establish distinctive versus unique features of epithelial-mesenchymal communication in nascent tumours, surrounding tissue and carcinogen-free controls to assess the carcinogen-stress response.**
- **Additionally, to visualise the changes induced by tissue-wide fibrosis in a non-tumour context, we will conduct an in vivo bleomycin experiment, as suggested by the reviewer.**

3) Supplemental Table 3 is informative and very appreciated. However, it does seem to suggest that only three total samples were submitted for 10X processing? If this is not the case, then this point should be made very clear and within/across group comparisons performed. Otherwise, these should be repeated with at least $n = 3$ per group. I assume samples were pooled in order to hit the minimum cell number targets, but this can be overcome through HTO-barcoding such that cells are labeled based on their tissue/tumor of origin prior to pooling and processing at the correct scale. For a study of this magnitude, I think multiple biological replicates are critical in order to ensure reproducibility.

A full description of the samples submitted for 10x processing can be found in the "Supplemental Table 3" file. The "pre-processed" tab shows the number of biological replicates and libraries (namely 2-3 libraries per condition, each of which contains samples pooled from 8 animals).

We agree that HTO-barcoding is a powerful technique that would allow for samples to be pooled while retaining information on the tumour of origin. However, our micro-dissected tumours are formed by very few cells, many of which are lost throughout downstream processing. Therefore, any additional processing (antibody labelling, FACS, or HTO-barcoding) poses the risk for a complete loss of the sample. To put this into perspective, for our scRNA-seq analysis, we captured between 50-650 cells per micro-dissected tumour. Overall, this represents an inevitable technical constraint inexorably linked to studying tumours at incipient stages. Hence, sample pooling was the only viable option to recover a satisfactory number of cells for subsequent analyses.

Action points:

- **A more detailed breakdown of the data sample processing pipeline will be included in “Supplemental Table 3”.**

4) Supplemental Table 4 is inadequate and calls into serious question the authors’ judgement in assigning cell types to each cluster, which is not well-described. There are numerous methods for automated classification, such as SingleR and scATOMIC, among others, but no such methods are discussed. Furthermore, based on the genes listed in Supp Table 4, it appears that the authors used a small number of “canonical” marker genes to make these decisions, which is an outdated way of thinking of scRNA-seq data. For example, they defined “myofibroblast” solely by ACTA2... despite this also being a marker for what the authors call “Myocytes”, among others. It is also unclear why “myofibroblast” is a different category than “myofibroblastS”. This is all highly problematic and undermines faith in the sophistication of the authors in this field. It is also not clear how the authors moved from 38 initial clusters to their final 11 “known biological cell types”. Both cluster consolidation and cell type ascription should be more methodical and described in better detail. At the very least, a complete list of DEGs for the final cell types should be provided.

We apologise for the lack of clarity on this point. During our initial scRNA-seq analysis, we performed automated cell type annotation using PanglaoDB (4,909 markers for 163 cell types available in the database; PMID: 30951143). This approach was satisfactory to annotate the major epithelial and stromal cell types. However, this annotation strategy failed to capture the full breadth of biological heterogeneity identified by manual annotation. Hence, the manual annotation strategy used in the current version of the manuscript.

Action points:

- **To address this point, we propose to use a combined approach, where major cell types will be annotated using an automated classification pipeline and refined manually (based on literature mining) to add less frequently present cell types such as pericytes, lymphatic endothelial cells, fibrocytes, among others.**

Apologies for the unintentional repetition of the myofibroblast marker Acta2, and for the confusion in relation to the marker gene list used for annotation in “Supplementary Table 4”. A number of markers are present in more than one annotation list, given that certain cell types cannot be assigned by the sole expression of specific genes. Instead, annotation in this case is based on the co-expression of a subset of genes (e.g. Acta2 in myocytes, pericytes and myofibroblasts). We will include additional UMAP and Violin plots to exemplify this as well as a detailed supplementary table with DEGs across the final cell types assigned. Preliminarily, we have improved the content of “Supplementary Table 4” to provide further clarity.

5) Similarly, the three “states” ascribed based on pseudotime analysis are not convincing. Although these states themselves are described well in Supp Table 6-8, how they were arrived at, using what cutoffs, and how the number (three) was determined is not clear. In the absence of a principled heuristic, it is not clear that correction for multiple hypothesis generation would make these findings non-significant relative to those potentially available.

In methods in “Cell transition trajectory analysis” we specify that the formula differentialGeneTest and the top 1200 DEGs ranked by FDR-corrected p-value were used. Based on publications (PMID: 32709844) where a similar method was applied, similar biological relevance was identified, reassuring our choice of 1200 DEGs. Regarding the three states, the number of branching points was automatically determined by the pipeline, without input from the user.

6) The decision to use Monocle 2, instead of Monocle 3, is also unusual, since the more recent release has been available for nearly 4 years and fixes many of the over-conclusions associated with the earlier deployment. I would ask that the reviewers repeat their analysis using the more sophisticated version of this pseudotime package.

We appreciate the expert advice of reviewer and are glad to incorporate this suggestion in a revised analysis of our data. We are confident that our observations are robust, as they have been extensively validated, and we are happy to improve the presentation of our data to enhance the clarity of our message.

7) The rationale for drawing distinctions around Sox9 and Krt6a is not justified in the current manuscript. Are there clearly bimodal distributions for each? Where and how were the cut points drawn? Again, this way of conceptualizing NGS data in terms of small buckets based on arbitrarily binned expression of a minority of genes is not consistent with the current understanding of transcriptomics in cellular biology. This is not FACS, where surface protein expression may be accurately assessed, but rather probabilistic sampling where measured expression of the same RNA molecule from the same cell will vary considerably over time.

We thank the reviewer for raising this point. Below we describe the rationale behind the choice of markers defining the early tumour state, as well as the approach to define thresholds for them. Additionally, please see the associated **Rebuttal Figure 2** below (Panels **e** and **i** have been added; **i** shows the expression levels of both markers, Krt6a and Sox9, across conditions; a horizontal line is overlaid on the value used as threshold for each). **This information will be included in a revised version of the manuscript.**

- As indicated in the manuscript, Krt6a and Krt17 were used as “two established early tumour markers in the DEN carcinogen model” (PMID: 25551772, Supplementary Figure 3c; PMID: 34646013, Figure 1e and Extended Data Figure 2e; our manuscript, Figure 1b,c; Extended Data Figure 1c-e). These two markers, together with gene enrichment analysis, were used to assign the transcriptional state defining early tumour cells both in the pseudotime trajectory and in the UMAP projection (state 2; Rebuttal Figure 2a-e). Please note that since tumours are largely microscopic, the punch biopsy approach used to enrich for tumour cells prior to scRNA-seq library preparation also contains adjacent non-tumour cells. Hence, the use of markers was extremely important to ensure that the scRNA-seq analysis focuses on the right cell population (early tumour cells).
- Sox9, in turn, was identified as a result of the gene expression analysis along the pseudotime trajectory. Sox9 represented one of the top transcription factors showing an increased expression in state 2 tumour cells (Rebuttal Figure 2f,h), among other genes such as Junb and Fosl1 (Rebuttal Figure 2f). Based on the literature, which

describes the role of Sox9 in cancer and ECM remodelling, and taking advantage of the fact that there are good antibodies to further assess the SOX9^{high} population, we decided to focus on the Sox9^{high} Krt6a^{high} population. Indeed, SOX9 protein expression was found to be enriched in nascent tumours (g; 10 days post-carcinogen treatment) and showed a heterogeneous expression profile that positively correlated with the presence of the stromal niche, which made it an ideal candidate marker.

- The double positive population for these marker genes (Sox9^{high} Krt6a^{high}) was determined based on each individual gene expression distribution. Cells over the inflection point above background expression levels were assigned as positive for both markers (Rebuttal Figure 2i). The assignment of Sox9^{high} Krt6a^{high} as nascent tumour cells was further validated by differential gene expression, which revealed that this double positive population recapitulates the transcriptional signature of the tumour state (state 2; Supplementary Table 9, “DEGs” tab).

Action points:

- **To address the reviewer’s point, we propose to use a combined “tumour state” gene signature (as defined by our scRNA-seq analysis) to reinforce the SOX9^{high} tumour cell selection for downstream cell communication analysis.**

Rebuttal Figure 2: Tumour-associated epithelial cell markers. **a**, UMAPs showing expression of the early tumour marker genes *Krt6a* and *Krt17*. Insets depict keratinocyte clusters split by condition: external control (Ctrl), internal control (DEN) and tumours. **b**, Basal keratinocytes ordered by Monocle2 along a pseudotime (PST) trajectory. The identified states (S1, S2, S3) and the expression of tumour cell markers *Krt6a* and *Krt17* are shown. **c**, Gene enrichment analysis of genes representing state 2, compiled from Gene Ontology Biological processes (GO:BP) and reactome (REAC) terms and grouped based on biological knowledge; terms with p -value $< 1 \times 10^{-16}$ were ranked by their $-\log_{10}(\text{FDR})$ value. Representative genes are listed below each plot. **d, e**, UMAPs showing the cell distribution from different conditions (**d**) and distribution of the different states identified by PST analysis of basal keratinocytes (**e**; see **b**). **f**, Expression of transcription factors along the two pseudotime trajectories: from state 1 to either state 2 or state 3. Increased expression of stress-related transcription factors is observed in the tumour state (S2). **g**, Representative confocal images showing heterogeneous SOX9 expression (cyan) among KRT6A^{high} cells (red) in nascent tumours (10d post-DEN). DAPI, blue; cyan arrowheads mark SOX9-positive cells; scale bar, 10 μm . **h**, UMAP showing expression of Sox9. Insets depict Sox9 expression in keratinocytes split by condition: external control (Ctrl), internal control (DEN) and tumours. **i**, Violin plots showing *Krt6a* and Sox9

expression in basal keratinocytes across conditions. Blue lines indicate thresholds for defining the Krt6a^{high} and Sox9^{high} (above the line), and Krt6a^{low} and Sox9^{low} (below the line) populations.

8) Again, I fundamentally disagree with the manner in which the authors conceive of their cell subpopulations. For a manuscript aimed at studying the relationships between the epithelium (tumor and non-tumor) and the stromal microenvironment, there is no main-text mention of the three widely accepted “canonical” fibroblast type subtypes (iCAF, myCAF, apCAF), and these are only buried in EDF 8b-c without explanation or context. A closer interpretation based on the literature (PMC6727976, PMC5339682, etc) would be helpful.

We realize that we have not been sufficiently clear in relation to this point. Indeed, as the reviewer indicates, cancer associated fibroblasts (CAFs) represent a key component to be examined when investigating the relationship between tumour epithelium and stroma. Accordingly, as part of our initial fibroblast characterization in tumour samples, we analysed our scRNA-seq data in search of a CAF-associated signature. We described such CAF (iCAF, myCAF, apCAF) analysis in the main text of our manuscript: “We next assessed whether the niche-forming fibroblasts recapitulated CAF features by comparing the gene expression profile of C23 tumour cells to DEN and Ctrl conditions. Despite a subtle upregulation of CAF-associated genes, including S100a4, Vim, Fap, Postn and Ctgf, their protein levels (VIM, FAP, FSP1) remained largely unaltered or non-detectable in the niche of surviving tumours (Extended data Fig. 8b-e; Supplementary Table 12). Similarly, we did not observe the typical pro-tumorigenic increase in fibroblast proliferation (Extended data Fig. 8f), suggesting that niche+ fibroblasts do not present a fully established CAF phenotype at incipient tumour stages. However, niche+ fibroblasts from nascent tumours did express nuclear YAP (Extended data Fig. 8g), indicating the presence of pre-CAFs in the niche, i.e., fibroblasts in a transition state prior to becoming CAFs.”

We thank the reviewer for providing literature on CAF subtypes in pancreatic ductal adenocarcinoma (PDAC). While there is a large field of work around CAFs on PDAC, this cancer type is known to be particularly fibrotic in comparison to other solid tumours (PMID: 37190281) and hence may not be the best reference for our upper GI tumour model. Consequently, to characterise the fibroblasts present in our scRNA-seq data, we focussed on studies of normal oesophageal tumours transitioning towards dysplasia and finally to invasive carcinoma (PMID: 32709844 (mouse) and PMID: 36963399 (human)), which we believe would more closely reflect the CAF-associated phenotype expected in our samples of interest. The tumour fibroblast annotations was further informed by the overall literature on CAFs, covering markers identified across different models (PMIDs: 31980749, 36139552). Noteworthy, in oesophageal tumour studies, CAF subtypes have only been found in moderate/severe dysplasia (PMID: 36963399). Hence, since our study focuses on even earlier stages, i.e. hyperplastic/low-grade dysplastic stages (pre-neoplastic/pre-cancer tumours) well before cancer per se, it was not entirely surprising that CAF features were not fully established in our model. Please refer to text description “despite a subtle upregulation of CAF-associated genes, including S100a4, Vim, Fap, Postn and Ctgf, their protein levels (VIM, FAP, FSP1) remained largely unaltered or non-detectable in the niche of surviving tumours (Extended data Fig. 8b-e; Supplementary Table 12).” Precisely here is where the relevance of our study

resides; the fact that fibroblasts start contributing to the promotion of tumour survival even before acquiring a full-blown CAF phenotype is not only novel but rather remarkable.

Action points:

- We will elaborate on our CAF characterization in the text. This was initially placed in supplementary material given that the CAF phenotype is not fully established; however, we will gladly move part of this analysis to the main figures, if deemed necessary or useful.
- To demonstrate that the lack of an established CAF phenotype in early tumours is stage-dependent, we will assess the expression of established CAF markers (FAP, α SMA, VIM, FSP) on more advanced cancer stages (invasive carcinomas) in the DEN model. Newly obtained preliminary data revealing the presence of CAFs in DEN-induced invasive carcinomas can be found below (Rebuttal Figure 3). In particular, we detect fibroblasts that express Vimentin, α -smooth muscle actin (α SMA), and fibroblast activated protein (FAP) in our DEN tumour model. Note that, given the slow progression of the DEN model, advanced tumours as the ones shown in the Rebuttal Figure 3 are rare. Nevertheless, more samples are being collected and will be included in the revised manuscript.

Rebuttal Figure 3: Cancer-associated fibroblast (CAF) markers in invasive carcinoma of upper gastrointestinal tract after treatment with DEN carcinogen. a, H&E image of a transversal cross-section from the upper-gastrointestinal tract displaying invasive carcinoma in samples collected 12 months after DEN treatment. Scale bars: 1cm (left) and 500 μ m (right; inset). **b-d,** Representative confocal images of a transversal cross-section of an invasive tumour labelled for the typical CAF markers α SMA (**b**), VIM (**c**), and FAP (**d**) (green) that overlap (white arrow head) with cells positive for the resident fibroblast marker PDGFR α expression in tumour stroma. DAPI (blue), Scale bars, 30 μ m.

9) In Figure 4a, instead of referring to fibroblasts based on their cluster number and relative PDGFRA expression, it would be more helpful to examine canonical iCAF markers such as IL6, CXCL12; myCAF markers such as LRRC15, ACTA2; and apCAF markers such as CD74.

We agree with the reviewer that this would have been the ideal way to present the different fibroblast subtypes had they been represented in our dataset. However, as stated in response to remark #8, our pre-malignant tumours do not show evidence of an established CAF signature (Extended data Fig. 8). In **Rebuttal Figure 4** (below), the expression levels of the marker genes proposed by the reviewer are shown across conditions as violin plots, with individual plots for $Pdgfra^{low}$ and $Pdgfra^{high}$ fibroblast populations.

Given that the results of our experiments revealed different contributions of the $Pdgfra^{low}$ and $Pdgfra^{high}$ fibroblast populations to the early tumour niche (as shown by lineage tracing, histology, and the analysis of our scRNA-seq data; Fig. 4), we consider that it is justified focusing on these two populations for our downstream analyses (Extended data Fig. 7e,f; Supplementary Table 10).

Action points:

- For further clarity we will provide additional violin plots showing the gene expression profile of canonical CAF markers across conditions in $Pdgfra^{low}$ and $Pdgfra^{high}$ fibroblasts (as in Rebuttal Figure 4).

Rebuttal Figure 4: Typical cancer-associated fibroblast subtype markers in $Pdgfra^{low}$ and $Pdgfra^{high}$ fibroblast populations across conditions. Violin plots illustrating \log_2 -transformed normalised expression levels of typical CAF subtype markers (myCAF: Acta2, Lrrc15; iCAF: Il6, Cxcl12; apCAF: Cd74) in control, DEN (internal control) and tumour conditions across the two defined fibroblast populations.

10) Figure 3h, or its associated supplements, should contain representative control images.

We thank the reviewer for pointing this out. Tumour-adjacent, tumour-free control areas immunolabeled for SOX9 were originally found in Figure 3m and Extended data Figure 6l. **We will refer to this more clearly in the revised version of the manuscript.** We apologise for missing to incorporate an image showing SOX9 expression in non-DEN treated control samples. Please find this image below in **Rebuttal Figure 5. This new figure will be added to the revised manuscript.**

Rebuttal Figure 5: SOX9 expression is absent in homeostatic oesophageal squamous epithelium. a, Representative confocal image showing normal (control) area and absence of SOX9 expression in epithelial cells. DAPI, blue; scale bar 10 μm .

11) I question the authors' judgement in distinguishing between high and low expression of the continuative fibroblast marker PDGFRA. Even if so, I think it is presumptuous to assume that proximity begets causal interactions.

We appreciate the reviewer concern. Our scRNA-seq analysis revealed clear differences in the expression level of PDGFR α across fibroblast clusters. We went on to validate this in tissue wholemounts and cross-sections by immunofluorescence (Extended Data Fig. 7b,c). Our histological characterization not only confirmed the presence of these two distinct populations of fibroblasts, but also revealed their compartmentalization in the tissue, with PDGFR α^{high} fibroblasts residing in the submucosa and PDGFR α^{low} fibroblasts in the lamina propria. Additionally, we assessed the distinctive role that these subpopulations play in tumorigenesis by performing unbiased lineage tracing of fibroblasts (Extended data Fig. 7g-j; Supplementary Table 11), as well as by assessing the differences in their transcriptional signatures through differential gene expression (Extended Data Fig. 7e,f, Supplementary Table 10). Our observations are in line with previous studies in the gut, where PDGFR α^{high} and PDGFR α^{low} compartmentalization in stromal fibroblasts has also been reported in connection with their role in epithelial morphogenesis and cross-compartment communication (PMIDs: 38781967, 33306673).

Although we believe that the results of our study provide evidence of the genuine existence of these subpopulations, we plan to address the reviewer's comment as suggested below.

Action points:

- **To reveal the distinctive nature of the PDGFR α^{high} and PDGFR α^{low} fibroblast populations, we propose to immunostain for additional markers of these two populations, as identified by scRNA-seq (including Has1, Cd34, Col3a1, Lama1). We anticipate that these markers will show a similar compartmentalization in the lamina propria and submucosa as the one observed for PDGFR α .**

12) For the fibroblast enrichment in Figure 4e, the authors should examine at a per-sample level the percent expression of their “PDGFRA-low” clusters and assess statistical significance.

We appreciate the expert advice of the reviewer. We will incorporate the proposed suggestions in a revised version of the manuscript.

13) The authors talk about going from 38 initial clusters to 11 final named clusters, but then on the second paragraph of Page 11 (“For this we focused on cluster 23 (C23)), they appear to be referencing back to the 38 cluster model? Is this correct? If so, they should provide some reference to where these cells are. Which of the final 11 cell types do these fall into?

Apologies for the lack of clarity, we did provide a correspondence between fibroblast cell type (one of the 11 identified cell types) and the original 38 clusters in the manuscript (Figure 4a). Additionally, we will include a cluster-assigned cell type table for further reference.

14) The CellChat analysis adds almost nothing to the manuscript and is greatly over-concluded. This statement is incredibly misleading and should be removed: “Overall, our scRNA-seq and communication analyses uncovered the pro-fibrotic/wound-healing transcriptional profile characteristic of tumour niche fibroblasts”. The CellChat methods also contain insufficient detail.

In our view, the CellChat analysis was critical to further discern how nascent tumour epithelial cells and fibroblasts communicate. In this sense, EGFR-mediated communication, which has gained prominence as an important signalling pathway modulating mutant cell competition even before tumours emerge (PMID: 37344586), was identified in our CellChat analysis as one of the strongest pathways by which tumour cells (Sox9^{high} Krt6^{high}) and niche fibroblasts (Pdgfra^{low}) communicate. Such conclusions could not be reached by differential gene expression analysis alone.

Action points:

- **We regret that our statement was found to be misleading. We will add further details to the CellChat section and improve the presentation of our data, which will certainly improve our manuscript.**
- **Apologies for the lack of detail in our methodology, only published pipelines were used for the analysis (no new scripts were developed). We will revise the methods section to clarify this.**

15) The in vivo knockout and inhibition experiments are probably the most compelling portions of this manuscript, but are sparsely covered both in terms of text and figure panels. Additional description of these very interesting results would be warranted.

We appreciate that the Reviewer finds our functional studies to be “very interesting” and “compelling”.

Action points:

- We propose to provide additional functional data, where we will be inhibiting EGFR using Gefitinib in vivo to determine whether the activation of this pathway mediates fibroblast recruitment, early tumour niche formation and/or long-term tumour survival.
- Additionally, we propose to combine EGFR inhibition (via Gefitinib) with an inhibitor of fibronectin matrix assembly (FUD; already used in the current manuscript as shown in Figure 6e) to assess the synergistic effect of both processes in early tumour survival.
- Finally, we propose to induce high levels of recombination in $Krt14^{CreER}Sox9^{flox/flox}$ mice ($SOX9^{cKO}$) and to subsequently assess for the presence/absence of the niche in tumours 10d post-carcinogen treatment, matching the timepoint originally used to characterise tumour survival. This will offer deeper insights into the mechanisms by which SOX9 influences the formation of the niche in nascent tumours.

Referee #2 (Remarks to the Author):

This manuscript by Skrupskelyte et al aims to better understand how the pre cancer niche provides initial steps for tumor formation and contributes to its progression and survival. The authors use an elegant way to induce and track pre-neoplastic squamous tumors in the upper gastrointestinal mouse tract utilizing different imaging techniques such as confocal microscopy and SHG, combined with scRNA-seq and 3D cultures. They identify niche+ tumor structures that are enriched with stromal remodeling fibroblasts which affect keratinocyte proliferation and promote long term survival. Transcriptomic analysis identified a Sox9high cancer cell state that promotes fibroblast migration by EGF signaling, leading to production of fibronectin by the fibroblasts.

The model used is interesting and the notion of studying early events is important. It is not clear at all though if this has significant human relevance since there is no human validation. Moreover, the authors claim that the cells they study are early CAFs but do not express classic CAF genes when in fact they do express what would appear to be a myCAF signature. It is true that the marker genes are not the classic. However, this could again be a consequence of the model being a model without human validation. Moreover, though the authors collected different tumor niches within each GI tract, the number of biological replicates (i.e. different mice) along the manuscript is relatively low (n=4). The authors must add human validation of at least the Sox9, AREG and FN connection, as well as their subset markers. Without these the observations are interesting, but the novelty is limited, since it is already well established that tumor development is modulated and perhaps even initiated by changes in the tumor microenvironment and specifically by fibroblasts (Sahai et al, 2020, de Visser and Joyce, 2023).

Reviewer #2 Authors' response: We thank the reviewer for the overall positive comments, highlighting the “elegant”, “interesting” and “important” nature of our work. We are also grateful to the reviewer for directing our attention towards aspects of the data that need further consideration to strengthen the claims of our study.

Major comments:

1a. It is well known that the TME changes at early stages. The changes shown here are still generated by the carcinogen and cancer-stroma signaling so not totally surprising or novel. Would these occur without cancer cells? Is the stroma different before the cancer cells are there? It is very hard to tell but maybe treating the stromal matrices with DEN would provide some answer.

We appreciate the reviewer's comment, and agree that the relevance of the TME in early cancer stages is well established. However, to our knowledge, no previous reports exist in which the nature of epithelial-mesenchymal interactions at nascent stages of tumorigenesis was investigated. By this, we refer to tumours ~10 cells in size. Part of the novelty of our work stems in the heterogeneous nature of nascent tumours. In particular, the observation that nascent tumour fibroblasts contribute to the formation of the niche and effectively prime

tumours for survival even before acquiring a cancer-associated fibroblast (CAF) phenotype paves the way to understand nascent tumour-stromal interactions. We apologise for failing to convey this message in the current form of the manuscript. **The point will be further elaborated and clarified in a revised version.**

Regarding the reviewer's concern that the carcinogen (DEN) treatment may directly impact the stromal compartment, we agree that this aspect needs to be considered when studying early tumorigenesis, particularly because humans are exposed to carcinogens systemically. Our data, however, argues that fibroblast changes are unlikely to be induced by the carcinogen treatment directly. We observe that, while the tumour epithelium presents mutations in cancer genes (Colom et.al. Nature. 2021), there are no significant changes in the mutational burden of fibroblasts neither in tumours nor in adjacent tumour-free DEN-treated tissue (Extended Data Figure 8h,i). Additionally, if DEN induced a stress response in fibroblasts, one would expect stromal remodelling to also occur in adjacent tumour-free regions. However, we show that the fibroblast-associated remodelling occurs directly underneath nascent tumours, with the surrounding tissue showing a completely normal phenotype (Fig. 1c, 3m). This suggests that distinctive mechanisms govern cell behaviour in the areas where tumour lesions emerge, and that the stromal remodelling we observe is likely induced by the mutant epithelium. **This will be clarified in the text and additional zoomed-in figures of adjacent tumour-free stroma added.**

Action points:

- **To further demonstrate that stromal remodelling occurs as a consequence of tumour formation, the revised version of the manuscript will include the characterization of the stroma in samples collected immediately after carcinogen treatment (i.e., before tumours are formed). Stromal wholemounts stained for early tumour fibroblast markers as identified in the relevant clusters of the scRNA-seq data will additionally reveal potential phenotypic changes in fibroblasts before tumour formation.**
- **Additionally, we will experimentally assess the impact of DEN in the stroma. To this end, keratinocytes from healthy tissues will be exposed to a tumour-free DEN-treated stroma. Epithelial cell proliferation will be assessed by EdU incorporation.**

1b. Following up on this, in figure 3 the authors show that exposure to tumor stroma makes normal epithelial cells proliferate. This is not surprising and has been shown numerous times already. Would it happen similarly though from niche+ vs niche- stroma? Or when immune cells are depleted? In Figure 3o,p- The authors used mouse model to deplete Sox9 keratinocytes and measured the number of tumors 1-month post DEN treatment. On which tumors did you look at? Given the number of niche+ tumors is kept stable and low along time in DEN treated mice (Figure 1d), In order to prove Sox9 is a relevant player in epithelial-stromal communication from the most incipient stages of tumorigenesis, the authors should measure the number of niche+ tumors and niche- tumors separately, instead of measuring the number of all tumors together. Otherwise, the authors can't exclude the fact that the reduction in tumor burden originates from the decrease in niche- tumors as shown in figure 1d.

We thank the reviewer for these insightful comments. Multiple points are discussed here:

a) In relation to the pro-proliferative effect of tumour fibroblasts on epithelial cells, we acknowledge that the effect of the tumour stroma on epithelial cells is known. However, very little is known about this the early tumour stages investigated here and, in particular, the heterogeneous nature of this communication, which is what we have demonstrated throughout the manuscript. The tissue recombination assay represents one of several experiments exploring this question (Fig. 2). Regarding the suggested experiment comparing niche+ vs niche- stroma, we would like to clarify that in our experiments only niche+ tumours were used. Technically, it is not possible to perform this experiment with niche- tumours. This requires the visualization of early (minuscule) tumours under a dissecting microscope. And, unfortunately, given their less apparent morphology, niche- tumours cannot be visualized under a dissecting microscope, but only a confocal microscope in fixed and immune-stained tissues. Notwithstanding, our results indicate that niche+ tumour cells possess an increased proliferative capacity in vivo (Fig. 1e-f; Extended data Fig. 2g-j; Supplementary Table 2), which is associated with a greater tumour growth compared to niche- tumours (Fig. 1g). Hence, in tissue recombination experiments, we would anticipate that niche- tumours would not confer any tumour properties to control epithelial cells (i.e., from animals not treated with the DEN carcinogen).

Action points:

- **We recognise that we cannot dissect the contribution of the immune compartment in our tissue recombination experiments. To assess this, we propose to perform a new experiment in which DEN is administered to immunocompromised mice (specifically NOD.Cg-Prkdc^{scid}Il2rg^{tm1Wjl} SzJ mice; PMID: 15879151) and niche+ and niche- tumours are quantified.**

b) Regarding the conditional SOX9 deletion experiment, we are grateful to the reviewer for pointing this out. To clarify the impact of SOX9 on the tumour niche heterogeneity, we have included new data below (Rebuttal Figure 6). Our results show that SOX9 deletion reduces the size of niche+ tumours to values matching those of niche- tumours (Panel d). Please note that, as previously reported, SOX9 expression is necessary for initial tumour formation (PMID: 26095047). We found that the latter also holds true in the DEN model.

Rebuttal Figure 6: Sox9 conditional KO in epithelial cells reduces early tumour burden. **a**, Experimental protocol: $Krt14^{CreER} Sox9^{flox/flox}$ mice (SOX9cKO) received Tamoxifen (TAM) followed by DEN treatment. Tissues were collected 1 month-post DEN and compared to DEN-treated uninduced controls (Ctrl). **b**, Quantification of mice subjected to the experimental protocol depicted in (a); tumour burden decreased by 45.6% in induced mice. Results from $n=3$ mice per group; data is presented as mean \pm s.e.m; significance was assessed by a one-tailed Mann-Whitney test. **c**, Quantification of SOX9+ tumours in induced and uninduced mice from (a). $n=3$ mice per group; significance was assessed by a two-tailed Mann-Whitney test. Data is presented as mean \pm s.e.m. **d**, Tumour diameter in niche- and niche+ tumours from induced and uninduced mice (as described in a) 1 month after DEN treatment. Data is expressed as mean \pm s.e.m; $n=3$ mice per group. Significance was assessed by a two-tailed Mann-Whitney test.

Action points:

- The new data presented in Rebuttal Figure 6 will be included and discussed in the manuscript, highlighting the role of SOX9 in tumour niche formation and heterogeneity.
- Additionally, we propose to highly induce $Krt14^{CreER} Sox9^{flox/flox}$ mice (SOX9cKO) and to quantify the number of niche+ and niche- tumours 10d after carcinogen treatment, to match the timepoint originally used to characterise tumour survival. This will offer deeper insights into how SOX9 influences niche formation and heterogeneity in nascent tumours.

2. Much of the analysis is based on 3-4 mice per group. The experiments in Figure 1, for example, show dozens of dots in panel g but these are all from 3 mice. The authors must take into consideration the fact that different tumors in the same mouse are expected to be more similar or co-dependent than tumors from different mice.

We appreciate the reviewer's concern about biological replicates. However, we would like to stress that the presented approach enabled the detection of hundreds of localised and well-defined 'micro-tumours' in each oesophagus (average surface area of 30mm² analysed) ten days after DEN withdrawal (as characterised in Colom et.al. Nature. 2021). These tumours undergo a selection process and grow over time. The large number of tumours present in each animal, as well as their well-defined localization, allows us to reduce the number of animals used (see details in Colom et.al. Nature. 2021). This constitutes an important 3R

(Replacement, Reduction and Refinement) improvement compared to other approaches where mice develop diffuse rapid growing tumours across the tissue.

Action points:

- We have split the plot mentioned by the reviewer in Figure 1 to show each animal individually. The plot clearly illustrates the reproducibility across biological replicates in terms of the number of niche- and niche+ tumours. This data will be added to a revised version of the manuscript.

Rebuttal Figure 7: Niche- and niche+ tumour diameter in animals of distinct ages. Tumour diameter (μm) at the indicated timepoints; tumours from each animal are plotted separately for each timepoint. Blue dots, niche-tumours; red dots, niche+ tumours. Data is expressed as mean+s.e.m; n=3 mice per group. Each dot represents a tumour.

3. The Sox9 connection is very interesting, but it is not clear what made the authors choose this factor originally. Was it one the most DE genes? Or educated guess? Would other cancer genes also show this close to stroma pattern? Also, the authors say that Sox9 is high close to stroma but in fact most cells in fig 3m are Sox9 negative. Once again, the statistical analysis in 3n relies on many tumors albeit in very few mice.

We thank the reviewer for raising this point. Please see our answer below.

a) Sox9 was identified as a result of our gene expression analysis along the pseudotime trajectory. Sox9 represented one of the top transcription factors showing an increased expression in state 2 tumour cells (Figure 2f,h). Other genes identified included Egr1, Junb, Fosl1, and Atf3 (Figure 2f). Based on the literature (PMID: 38386758, 26095047), which describes the established role of Sox9 in cancer and ECM remodelling, and taking advantage on the existence of good antibodies to label and examine the SOX9^{high} cells, we decided to focus on the Sox9^{high} Krt6a^{high} population. Indeed, the expression of SOX9 at the protein level was found to be enriched in nascent tumours 10 days after DEN (Figure 2g; 10 days post-carcinogen treatment) and showed a heterogeneous expression profile that positively correlated with the presence of the stromal niche, which made it an ideal candidate marker.

b) Regarding Figure 3m, as originally stated: “we took advantage of sporadic clusters of KRT6A+/SOX9+ cells found in phenotypically normal (non-tumour) areas of DEN-treated tissue, likely marking prospective tumour cells prior to lesion formation,” to determine whether there was any evidence of stromal reorganization in the vicinity of SOX9 high cells in a tumour-free tissue context; in other words, to assess whether SOX9 high cells were promoting the formation of the niche. For this reason, most cells in this figure are SOX9 negative, these do not represent tumours (at least, yet). The clusters are stress areas showing sporadic cells expressing high levels of SOX9. Indeed, the results demonstrated that SOX9 expression appears to elicit changes in mesenchymal cells, independent on whether they belong to tumours or not. We concluded “Isolated SOX9+ cell clusters showed signs of fibroblast attraction, presenting an increased density of fibroblasts directly underneath them (Fig. 3m; Extended Data Fig. 6l,m)”. This can be further clarified in a revised version of the text.

In relation to whether any other markers would show the same profile of proximity to the stroma, we can confirm that SOX9+ clusters were also positive for the early tumour marker KRT6A, known to denote tissue stress (PMID: 38299111). Please refer to Extended Data Fig. 6l.

[Text Redacted]

[Figure Redacted]

[Text Redacted]

c) As per the statistical analysis comment, a two-tailed Mann-Whitney test comparing the regions was performed considering the biological replicates (n=4 per condition; individual data points were not the source for statistical data).

4. How is niche+ vs niche- defined? A threshold of expression of specific factor(s)? This is not defined. Moreover, the authors use PDGFR-alpha to define fibroblasts in Figures 1-3, but then in the scRNA-seq analysis they show that in fact there is a PDGFR-alpha high and low group. So how can this be used as a pen marker for fibroblasts in the first figures?

Apologies for the lack of clarity. Please see comments below:

a) Niche+ and niche- tumours were defined by the presence/absence of a “**nest-like structure composed of stromal fibroblasts (PDGFR α +).**” Such structure emerges under nascent tumours 10 days post carcinogen treatment and becomes immediately apparent upon confocal inspection. Niche fibroblasts extend cytoplasmic protrusions that reach the epithelial surface, a feature not observed in healthy tissues. Thus, the presence/absence of a niche is not determined based on the expression level of specific factors, but on prominent morphological changes in the stroma. **This will be clarified in the text.**

b) Regarding the expression of PDGFR α , we would like to clarify that although all fibroblasts express this marker (PMIDs: 38781967, 33306673, 24336287, 30737373), the expression level is compartmentalised across the two distinct stromal layers, i.e. the lamina propria (PDGFR α ^{low}) and the submucosa (PDGFR α ^{high}; see Extended Data Figure 7b-d). High magnification images of the niche directly underlying early tumours, i.e. localized in the upper stroma (lamina propria), were acquired with exposure settings optimised to the expression levels of PDGFR α in that tissue layer. This enabled us to properly show and illustrate changes in the stromal architecture and niche remodelling. All images including both stromal compartments (lamina propria and submucosa) reveal the heterogeneous expression of this marker. **We will make a note in the text, figure legends and methods to indicate this imaging setting adjustment.**

5. The authors compare the niche fibroblasts to known CAF signatures and find very little overlap. Did they try to look at full blown tumors in their model? Or in other mouse models of esophageal cancer? It is very much possible that the specific markers here are different and that these are in fact myCAFs as also appears from the ligand-receptor analysis.

We fully concur with the reviewer in that tumour-associated fibroblasts mostly resemble the myCAF signature in our scRNA-seq data (Figure 5, Extended Data Figure 8). There is a subtle increase in myCAF-associated genes, including S100a4, Vim, Fap, Postn and Ctgf, compared to control conditions (please see Rebuttal Figure 9 below). However, such changes are rather modest. While the ligand-receptor analysis suggests ECM production as their main function, again resembling the myCAF function, the protein levels of these markers (VIM, FAP, FSP1) remain largely unaltered or non-detectable in the niche of surviving pre-neoplastic tumours (Extended Data Figure 8d-e). The latter suggests that although these fibroblasts might be primed to become myCAFs, such phenotype has not yet been fully established.

Interestingly, we found that early tumour fibroblasts show a transcriptional profile that closely eludes to that of wound-associated fibroblasts (see Rebuttal Figure 9 below; PMID: 30737373). This suggests that at early tumour stages, before the CAF phenotype becomes fully established, fibroblasts exhibit a mixed transcriptional profile dominated by a wound-

like signature, most likely as a result of the mutant epithelium, but seemingly transitioning towards an early myCAF state.

Action points:

- To address this point, we will elaborate on our CAF characterization in the main text.
- To demonstrate that the lack of an established CAF phenotype in early tumours is stage-dependent, we will conduct experiments on more advanced cancer stages (invasive carcinomas) found in the DEN model and assess the expression of established CAF markers (FAP, α SMA, VIM, FSP). Newly obtained preliminary data denoting the presence of CAFs in DEN-induced invasive carcinomas can be found in Rebuttal Figure 3, see Reviewer#1 point 8. Here we show that, in advanced cancer stages, fibroblasts do express Vimentin, α -smooth muscle actin (α SMA) and fibroblast activated protein (FAP) in the DEN model. Note that advanced DEN tumours are rare given its slow progression, which more closely recapitulates human disease. Additional samples are being collected and will be included in the revised version of the manuscript.

Rebuttal Figure 9: Expression level of typical wound fibroblast and cancer-associated fibroblast markers in Cluster 23 fibroblast population across conditions. Violin plots illustrating log₂-transformed normalised expression levels of typical myCAF subtype markers and wound-associated fibroblast markers in control, DEN (internal control) and tumour conditions within Cluster 23 (C23) fibroblast population.

6. In the last paragraph of the results the authors claim that Sox9 exerts its effect through EGFR signalling. But they do not make the experimental link between Sox9, EGFR/AREG and FN. Is this signalling altered in the Sox9 KO mice? This should be tested in mice, and, importantly, verified in human samples. Specifically, the authors should test whether SOX9 expression is found in early tumor developmental stages or has any survival advantage.

We thank the reviewer for this suggestion. We aim to deepen our understanding of the connection between EGF signalling, fibronectin expression, and SOX9 by performing the following experiments:

Action points:

In mouse:

- **In vitro: Tumour-derived fibroblasts will be treated with an EGFR inhibitor (Gefitinib) in the presence or absence of AREG (EGFR ligand) to assess whether EGFR-mediated signalling modulates fibronectin expression and fibroblast migration.**
- **In vitro: Tumour-derived keratinocytes will be treated with Gefitinib in the presence or absence of AREG to assess whether EGFR-mediated signalling modulates the expression of SOX9 and of other tumour-associated genes, including KRT6A, KRT17, JUNB and FOSL1.**
- **In vivo: DEN-treated mice will be treated with Gefitinib to assess niche+/- tumour formation, as well as SOX9 expression.**
- **In vivo: DEN-treated mice will be treated with Gefitinib and FUD (fibronectin assembly inhibiting peptide) to assess niche+/- tumour formation and SOX9 expression. This will reveal whether EGFR-mediated signalling and fibronectin expression cooperate during early tumorigenesis.**

In human

- **Given that upper GI cancers have a late clinical presentation (diagnosed at advanced stages), there is a lack of availability for relevant samples. As a compromise, we propose to explore the consequences of treating healthy oesophageal cells with DEN, and to assess the expression of SOX9, AREG and fibronectin in keratinocytes and fibroblasts in in vitro co-cultures.**
- **Additionally, we propose to assess the expression of SOX9, AREG and fibronectin at advanced cancer stages to determine whether these early tumour features persist long-term in tumours.**

7. Figure 4- the authors use Dvorak's theory to describe the pre cancer niche as the wound that never heals. However, they do not relate to the immune compartment that plays an integral role in wound healing processes. Supp figure 2 shows the presence of CD45 cells in nascent tumor niche 10 days post-DEN. Are there differences in immune cell composition between niche+ and niche- tumors?

Regarding the immunity point, we apologise for the lack of background. The role of the immune compartment in early tumour elimination in the DEN model was already explored in our previous collaborative work (Colom et al., Nature 2021). There, as indicated above, we studied the mechanisms involved in the disappearance of nascent tumours in the DEN model.

Our results demonstrated that most incipient tumours were rapidly lost with no indication of an anti-tumour immune response, including no differences in early tumour persistence in immunodeficient mice (**Extended data Figure 6 in Colom et al., Nature 2021**). Instead, early tumour clearance was shown to be largely driven by the competitive advantage of neighbouring mutant clones.

Action points:

- The new manuscript will contain the new data presented here (see Reviewer #1, point 1, and Rebuttal Figure 1) revealing that, at day 10 post-DEN, the immune infiltrates in niche+ and niche- tumours are not significantly different from each other.
- We plan to perform additional DEN carcinogen experiments to explore whether the niche+/- phenotype is altered in an immunodeficient context compared to what is seen in wild-type immunocompetent animals.

Minor comments:

1. Page 5: “However, at these early stages, the presence of this structure was largely heterogeneous, with the majority of epithelial lesions (~70%, 199 lesions out of 296 per esophagus, on average) showing no apparent stromal reorganization (Fig. 1c)” – where is this data shown? The authors should present this quantification.

The 70% statement comes from Extended Data Figure 2f. Apologies for not including the numerical data (please see Rebuttal Table 1 below). **This will be included in a revised version of the manuscript.**

Rebuttal Table 1. Number of oesophageal tumours detected at different time-points after DEN administration.

	10d post DEN			2m post DEN			8m post DEN			1y post DEN		
	Total	Niche -	Niche+	Total	Niche -	Niche+	Total	Niche -	Niche+	Total	Niche -	Niche+
Animal1	310	202	108	116	43	72	63	17	47	103	57	47
Animal2	245	168	77	133	40	94	103	15	88	53	5	48
Animal3	334	226	107	75	17	58	72	11	61	81	45	36
Average	296	199	97	108	33	75	79	14	65	79	36	44

Total tumours are the sum of Niche⁻ and Niche⁺ tumours.

2. Supp figure 2c-d – how did you calculate the proportion of endothelial cells and immune cells separately? They are signalling with the same fluorophore.

As exemplified in the image below (Rebuttal Figure 10I), endothelial cells and immune cells can be easily distinguished due to their different staining profile. Endothelial cells from vascular networks and are flattened and elongated, while immune cells appear as distinct individual cells with round, amoeboid-like shape). **Additional images where endothelial and immune cells are labelled using different fluorophores will be provided.**

Top down 3D Projection

Rebuttal Figure 10: Capillary network and immune cells in nascent tumours. A representative confocal image of nascent tumours 10d post-DEN. Markers: CD31 (white arrows) and CD45 (yellow arrows) (cyan), PDGFR α (white), KRT6A (red), DAPI (blue). Scale bar, 20 μ m.

3. Supp figure 2e- what is the sign of invasion? What does ITGA6 mark?

We thank the reviewer for highlighting the lack of background here. ITGA6 is an epithelial to basement membrane adhesion integrin. By staining for ITGA6, we show that the basement membrane is not breached by niche+ tumours. In the absence of such breaching, we concluded that there were no signs of tumour invasion. **The text will be amended accordingly to clarify this point.**

4. Figure 3h – the authors don't show the expression of SOX9 in healthy control tissue but state in the text is it not expressed as opposed to nascent tumors.

Apologies for not referring to the relevant control images in the text. A representative control area labelled for SOX9 can be found in Figure 3m and in Extended data Figure 6l, as well as in Rebuttal Figure 5.

5. "For this we focused on cluster 23 (C23) fibroblasts, a Pdgfra^{low} cluster enriched in tumour cells (Fig. 4e)." I guess the authors mean stromal cells from tumors. The wording is confusing.

We are grateful for the reviewer's attention to detail. **This was indeed an error and will be corrected.**

6. Ext Fig 1b – the significant p values for the different time points – what are they pointing to? There is only one group – what are they comparing to?

Apologies for this, the obtained p values are the result of pairwise comparisons between two consecutive timepoints (indicated by the line underneath the p value).

Referee #3 (Remarks to the Author):

Study presented by Skrupskekyte et al. aims to identify environmental changes underlying the transition of healthy epithelium towards cancer. The authors used a chemical carcinogen (DEN) to introduce genetic mutations in the squamous esophagus of mice and their observed two types of precancerous lesions. Some were surrounded by nest-like (niche+) structures composed of stromal fibroblasts. Niche+ predominantly developed into cancerous lesions. Next, using heterotypic cultures, they show that cell growth of healthy epithelia is affected by the underlying niche+ stroma. These cells can be more frequently transplanted into nude mice when exposed to niche+ stroma. Next, they performed scRNA-seq on the esophagus and squamous forestomach and investigated transcriptional patterns of the epithelium and stromal compartments. They observed a shift towards proliferating cells in the epithelium and the emergence of Sox9 cells as the disease progressed. In the stromal compartment, they identified Pdgfra low cells as potential drivers of niche+ phenotype. Subsequent cell-to-cell communication analysis identified fibronectin (FN) and EGF signalling pathways as drivers of stoma-to-epithelium interaction in the niche+ environment. They finally demonstrate that Sox9 high keratinocytes can communicate with fibroblasts using EGF induced signalling molecules and that fibroblasts signal to keratinocytes using fibronectin pathways. In general, this study aims to describe a novel mechanism of oesophageal cancer development with a strong focus on the role of microenvironment modulation during the initiation of this cancer. The elucidation of the nongenetic factors driving cancer initiation is of out-most importance and this study offers an important view on this process. The authors' use of chemical carcinogen that recapitulate normal oncogenesis is commendable. The authors further provide ample supplementary information. However, the study has technical and conceptual limitations:

Reviewer #3 Authors' response: We are grateful for the reviewer's insightful feedback and suggestions, which will certainly enhance the presentation of our manuscript. We are also pleased that the reviewer for considering that our study "offers an important view" in cancer initiation, and that the use of the DEN model in our study is "commendable".

1. First of all, in their single cells analysis of fibroblast, the authors avoid detailed analysis of individual clusters. It is very apparent from their analysis that the loss of cluster 8 (fig. 4a, S4e) is the most dramatic change between DEN and Ctrl samples. These cells are not well characterised by the authors and additional information on the identify of these cells (e.g. table of cluster markers) is needed.

We apologise for missing to elaborate on cluster 8. Given the increased fibroblast density and mesenchymal reorganisation observed in the stroma underlying early tumours, we focused our analysis on fibroblast clusters that became particularly enriched in early tumour samples compared to controls. As correctly pointed by the reviewer, cluster 8 is instead lost both in tumour samples and in adjacent DEN tissue (internal control), suggesting that this change is not specific of tumours per se.

Action points:

- **We appreciate the reviewers' feedback. In a revised version of the manuscript, we will include a more detailed characterisation of the different fibroblast clusters, as well as the list of marker genes for each cluster.**

2. Secondly, cluster 15 is also largely ignored in the course of authors analysis. These fibroblast in comparison to other fibroblast clusters, see to expand after DEN treatment and in the tumour samples. The presence of *acta2* and other fibroblast markers (Fig. S4g) indicates that these fibroblasts might be contractile in nature.

Following from the previous point and given the heterogeneity of fibroblast populations, we decided to use empirical data to guide our scRNA-seq analysis. That is, fibroblasts enriched specifically in tumours, but not in surrounding tumour-free tissue (Fig. 4d), and expressing low levels of PDGFR α (as established in our lineage tracing experiments; Fig. 4c, Extended data Fig. 7k-p) were the focus of our study. Cluster 15 did not form part of our central analysis because it represented a *Pdgfra*^{high} subset, which was not identified as the major component of the tumour niche. However, we did recognise that C15 was a cluster enriched in tumour samples. Hence, we performed a differential gene expression analysis for this cluster (*Pdgfra*^{high}) against the rest of fibroblast clusters (*Pdgfra*^{low}), the results of which were shown in Extended Data Figure 7e,f.

Action points:

- **In light of the reviewer's comments, we will include a more detailed characterisation of the different fibroblast clusters (including C15), and the list of marker genes for each cluster.**

3. Although the authors provide a limited set of markers per cell cluster, it appears that fibroblast studies here resemble populations S1, S2 and S3 previously annotated by Davidson et al (<https://doi.org/10.1016/j.celrep.2020.107628>). The studies are different enough (here authors investigate cancer initiation rather than injectable model) however the authors should investigate in detail the shift of clusters 8 and 15, as they resemble studies of Davidson et al and other CAF studies (<https://doi.org/10.1038/s41568-019-0238-1>).

We thank the reviewer for their expert insights on this matter. We appreciate that tumour-associated fibroblasts show a transcriptional signature that partially aligns with myCAFs (Figure 5, Extended Data Figure 8). Indeed, there is a subtle increase in myCAF-associated genes, including *S100a4*, *Vim*, *Fap*, *Postn* and *Ctgf*, in tumour samples compared to control conditions (see Rebuttal Figure 9 under Reviewer #2, point 5). However, changes are rather modest. Additionally, our ligand-receptor analysis suggests ECM production as their main function (again, resembling the myCAF function). Despite these observations, the protein levels (*VIM*, *FAP*, *FSP1*) remain largely unaltered or non-detectable in the niche of surviving pre-neoplastic tumours (Extended Data Figure 8d-e), indicating that the fibroblasts are primed to become myCAFs, but that the phenotype is not fully established yet. For this reason, we refrained from using the term "CAF" to describe the fibroblast populations identified in our study.

Interestingly, we observed that early tumour fibroblasts show a transcriptional profile that eludes to that of wound associated fibroblasts (see Rebuttal Figure 9 in Reviewer #2, point 5; PMID: [30737373](https://pubmed.ncbi.nlm.nih.gov/30737373/)). This suggests that, at early stages, before the CAF phenotype becomes fully

established, fibroblasts show a mixed transcriptional profile presenting a wound-like signature, most likely responding to the stressed caused by the mutant epithelium, that appears to transition towards an early myCAFs state as tumours progress.

Action points:

- To address this point, we will provide additional details on the heterogeneity of tumour fibroblasts in line with the CAF literature, including a comparison to the research of Davidson et.al., as suggested by the reviewer.
- To demonstrate that the lack of an established CAF phenotype in early tumours is stage-dependent, we will conduct experiments on more advanced cancer stages (invasive carcinomas) found in the DEN model and assess the expression of established CAF markers (FAP, α SMA, VIM, FSP). Newly obtained preliminary data denoting the presence of CAFs in DEN-induced invasive carcinomas can be found below (Rebuttal Figure 3, see Reviewer#1 point 8). The data show that, at advanced cancer stages, fibroblasts do express Vimentin, α -smooth muscle actin (α SMA) and fibroblast activated protein (FAP) in the DEN tumour model.

4. The choice of Col1A2 is not obvious as a driver gene for the Pdgfra-low fibroblast is poorly justified. This gene seems to be expressed in all fibroblast clusters (S4g) and a potential flow/transcommitment of cells between compartments cannot be excluded. The authors, in addition to pdgfra-low, should also lineage-trace other fibroblast populations (especially clusters 8 and 15) to understand if these cells do not contribute to the niche+ phenotype. If only cells from cluster 23 were responsible for the phenotypes observed, lineage training using other markers should not result in the clonal clustering described by the authors.

We fully agree with the reviewer that performing lineage tracing on the different fibroblast subpopulations, namely Pdgfra^{high} (cluster 15) and Pdgfra^{low} (clusters 8, 7, 23), would have been ideal. However, from our scRNA-seq analysis, we were only able to identify a few distinctive markers across these two populations (such as Has1/2 for PDGFR α ^{low} and Lama1 for PDGFR α ^{high}). Unfortunately, we are not aware of any existing mouse lines enabling the tracing of cells expressing these markers. And even if they existed, these markers are also expressed in other cell types. Together, these constraints render such an approach not feasible.

We selected instead an unbiased approach for lineage tracing. By using Col1a2, we were able to trace all the fibroblast populations present, as rightfully pointed out by the reviewer, including Pdgfra^{low} and Pdgfra^{high} (See UMAP). This strategy allowed us to study the cellular dynamics associated with tumour niche formation.

We agree that, on the other hand, our strategy makes it challenging to identify fibroblast transcommitment events, such as those from Pdgfra^{low} to Pdgfra^{high} cells, and vice versa.

However, given that the PDGFR α ^{low} and PDGFR α ^{high} populations are compartmentalized in the tissue (lamina propria and submucosal stromal compartments, respectively; Extended Data Fig. 7c,d), we anticipate that this is unlikely. In this sense, we did not observe any clonal events spanning both the lamina propria and submucosal compartment (Figure 4c). While it

cannot be excluded that all cells from a clone may migrate together, given the low cohesion of these cells, such a scenario is highly improbable.

Action points:

- **To address the reviewer's concern, we propose to use the *Pdgfra*-CreERT line (PMID: 29752282) to trace niche formation. We hope that tamoxifen titration may allow us to distinctively trace PDGFR α ^{low} and PDGFR α ^{high} subpopulations. However, this will need to be subject to assessment.**

5. Similarly, the trajectory analysis of the epithelial compartment lacks detail. The authors often switch between projections (monocle components and umap) and use different markers on different embedding types. This makes the analysis difficult to follow. It would be helpful if authors provided projects of all cell types and tissue types onto monocle projects and also the projection of states (1-3) on the Umap projects. Also, vector information in the analysis is missing. RNA velocity could provide information about the direction of cell trajectory. It is not clear to me if the cells flow from states 3 to 2 or vice versa.

We appreciate the expert advice of the reviewer and apologise for the confusing presentation. We will gladly incorporate the suggestions in a revised analysis of our scRNA-seq data. We believe the proposed improvements to the data presentation will enhance the clarity of our message.

Please note, a projection of the 3 pseudotime states we identified onto UMAP space is already provided in Rebuttal Figure 2 (Reviewer #1, point 7).

6. The validation of Sox9 genotype (figure 3o-p) only focuses on tumour burden in this model. The authors did not use this metric in any other setting in the study and additional analysis is of these mice is warranted.

We are grateful to the reviewer for pointing this in relation to the conditional SOX9 deletion experiment. To clarify the impact of SOX9 on the tumour niche heterogeneity, we have included new data (Rebuttal Figure 6 under Reviewer #2, point 1). In the Figure, we provide evidence that SOX9 deletion leads to a reduction in the size of niche+ tumours, specifically. Below, we propose additional experiments to strengthen this point.

Action points:

- **The new data presented in Rebuttal Figure 6 will be included and discussed in the manuscript, highlighting the role of SOX9 in tumour formation and niche heterogeneity.**
- **Additionally, we propose to highly induce *Krt14*^{CreER}*Sox9*^{flox/flo} mice (SOX9^{CKO}) and assess niche tumour heterogeneity at 10d post-carcinogen treatment to match the timepoint originally used to characterise tumour survival. This will offer deeper insights into how SOX9 influences niche heterogeneity in nascent tumours.**

7. In Figure 1e, niche- cluster of cells appears significantly smaller and there is some proliferation in its nuclei (the KRT6A signal seems to be overexposed). What proportion of nuclei in the niche+/niche- regions took up EdU? Figure S2g only has counts which might be

consequences of cellular density rather than higher proliferation rate. As these are representative images, it would be helpful if the authors chose regions of similar size to avoid optical illusions.

We thank the reviewer for their attention to detail. Apologies for the lack of clarity, but the data presented in Extended Data Figure 2g are normalised to total basal cell number. Thereby, variations in cell density and tumour size are accounted for. **This will be clarified in the revised version of the manuscript.**

Apologies for the overexposure of the images; a recognised feature of squamous tissue is that cornified layers on the surface of the tissue typically show misleadingly high immunolabelling signal compared to other layers. Similarly, we recognise that we did not select the most representative images for this figure. Please see below (Rebuttal Figure 11) showing a revised version of the Figure 1e. **These images, which were produced omitting the final (top) luminal cell layer to avoid overexposure, will be included in a revised version of the manuscript.**

Rebuttal Figure 11: Increased proliferation in niche+ nascent tumours. Representative confocal images of niche+ and niche- nascent tumours 10d after DEN withdrawal. Dashed line is marking tumour area. Markers: EdU (green), KRT6A (red), DAPI (blue). Scale bars, 25 μ m.

8. SOX9 staining on Figure 3k and 3m has a lot of background and it is very difficult to distinguish real cells/nuclei from the background

We appreciate the reviewers' concern. Unfortunately, staining tissue wholemounts (epithelial and stromal composites of around 200 μ m) can result in a grainy background when using certain antibodies; this is the case for SOX9 antibodies.

Action points:

- **To address this, we will stain new samples to obtain clearer images for Figure 3k and 3m.**

9. Until figure 4, authors imply that all fibroblasts are pdgfra+ however, starting in figure 4,

they show fibroblasts in submucosa are actually pdgfra-high and the cells in lamina propria are pdgfra-low. This change seems to be very weak in control samples in Figure 4c but much stronger in the tumour. It would be helpful if the authors provided larger fields of view on the earlier figures to understand that pdgfra expression is not constant in all fibroblasts. Also, the figure 4c, there seems to be a continuum of fibroblasts in the normal samples, but in the tumour samples, there is a clear boundary between compartments. The authors could also provide larger fields of view to ensure that the reader can clearly see these transitions.

We are thankful to the reviewer for raising this point. Regarding the PDGFR α expression, all fibroblasts express this marker (PMID: 24336287). However, we found that the expression level is compartmentalised across the two distinct stromal layers: PDGFR α^{low} fibroblasts reside in the lamina propria, while PDGFR α^{high} ones are found in the submucosa (Extended Data Fig. 7c,d). High magnification images of the niche directly under early tumours, i.e. of the upper stroma (lamina propria), were acquired with exposure settings optimised to the expression levels of PDGFR α in that layer. This enabled us to properly show and illustrate changes in the stromal architecture and niche remodelling. All images including both stromal compartments (lamina propria and submucosa) reveal the heterogeneous expression of this marker. **We will make a note in the text, figure legends and methods describing this imaging setting.**

In relation to the differences in PDGFR α compartmentalization across conditions, we apologise for not choosing the most representative images to illustrate this. Given that the thickness of the stroma

changes along the upper gastrointestinal tract, the stromal layers may be more or less apparent depending on the region of the tissue one looks at. For reference, see below and example of a control area where compartmentalization can be better appreciated: PDGFR α^{low} fibroblasts residing in the lamina propria, closest to the basal layer of epithelial cells, and separated from PDGFR α^{high} cells by a layer of αSMA^+ cells (green). This compartmentalization is typically more apparent in tumours due to the pronounced thickening of the lamina propria compared to controls.

Action points:

- **We will provide a clearer description on the stromal compartments, together with relevant images, in control and tumour areas.**
- **We will select a more representative control for Figure 4c and provide lower magnification images where the heterogeneous nature of the stromal compartment can be easily visualized. A preliminary example of such a figure, with a more representative control image, is shown below (Rebuttal Figure 12).**

Rebuttal Figure 12. PDGFR α^{low} fibroblasts form the early tumour niche. **a**, Experimental protocol for fibroblast lineage tracing: Col1a2^{CreER}R26^{Confetti/wt} mice received a dose of Tamoxifen (TAM) followed by DEN treatment. Samples were collected 6 months (m) after DEN withdrawal. **b-d**, Representative top-down (top) and side (bottom) views of tumour (b) and control tissues (b) collected as described in (a). Samples were immunolabeled for PDGFR α (greyscale). Lineage traced confetti⁺ cells are shown in yellow and red. Scale bars, 25 μ m. Dashed lines label the separation between the PDGFR α^{low} and PDGFR α^{high} stromal compartments. **d**, Split channel from **b** and **c** shows marked difference in PDGFR α expression across these two stromal compartments both in control and tumour samples; lamina propria (underlying the epithelium), PDGFR α^{low} ; submucosa (deeper stromal layer), PDGFR α^{high} .

10. Multiple figures lack correct annotation of markers displayed. E.g. figure 1b: if the signal from multiple channels is separated then it should be done for every channel (there should be only KRT6A channel in addition to PDGFRA only channel). The PDGFRA only channel is missing annotation. The “Niche+” and “8m” labels do not indicate that these images are the same projection and all metadata is shared. This low-quality image annotation is pervasive in the figures (figure 4c tumour samples is shown twice, one with pdgfra shown in grayscale and once with this signal missing. This is not obvious from the legend or figure annotation. Similar pattern can be found on 1b, 1c, 2g, 3k, 3m, 3h, 5g, 5h.

We thank the reviewer for pointing this out. Given the space constraints, we decided to optimize our approach by presenting the most relevant channel combinations (merged or split channel images), rather than showing all individual channels. Similarly, not to crowd the figures, when an image is part of a composite, annotations were only presented once per composite. This includes time point, niche +/-, scale bar and colour channel annotation (including all colours present in the composite). Any information that may be different across composite images would be repeated for every image (for example, time point). We regret that this has caused confusion. We will revise the figures to improve the annotation and presentation while keeping with the space constraints.

Authors' point-by-point response to reviewers (2024-07-13618A-Z)

Skrupskelyte et. al. entitled "Pre-cancerous Niche Remodelling Dictates Nascent Tumour Persistence"

*Text from reviewers' comments presented in **blue italics**, our responses to reviewers' questions in **black**, and quoted references are highlighted in **orange italics**.*

REVIEWER COMMENTS

Reviewer #1 (Remarks to the Author):

Skrupskelyte et al. utilize a well-known carcinogen-induced esophageal squamous cell carcinoma model to investigate factors that influence the survival of early tumors. They start with a noteworthy observation: the majority of early dysplastic lesions fail to form tumors and do not survive. Their findings reveal that a stromal niche, which includes PDGFRa+ fibroblasts, is linked to tumor survival. The authors present numerous visually striking images to this effect. However, the manuscript's momentum diminishes significantly as the authors revisit well-established growth factor pathways between epithelial and mesenchymal cells. In addition to a lack of novelty, the authors describe single cell states of stromal populations in an outmoded manner without comparison to contemporary literature describing the heterogeneity of fibroblasts in particular, and stromal cells in general. Perhaps the most notable shortcoming of this study is that it almost completely overlooks the immune system's role in nascent tumor formation. Tumor immunosurveillance is a well described phenomenon in carcinogen-induced tumors, and this is not accounted for in the authors' study. I believe that this represents a fatal flaw in this study that I do not think could be corrected without a fundamental reframing of the investigation. I have enumerated specific points below.

Reviewer #1 - Author response: We thank the reviewer for their insightful comments, and for highlighting the "*noteworthy*", "*striking*" nature of the data presented in our manuscript. We are also grateful to the reviewer for pointing us towards aspects of the data that needed further consideration in order to strengthen the claims of our study. We have addressed each of the comments as described below.

1) The authors appear to purposely avoid a discussion of immunity. Tumor surveillance is known to be a critical element of the early tumor suppression response. The authors' fundamental question is why do these nascent tumors disappear. They strongly imply that this is due to cell/niche stress. But it is almost certainly in fact simply tumor surveillance. How do they know that the immune system is not what wipes out these early tumors?

Question 1. Reviewer #1

We fully agree with Reviewer #1 on the importance of the immune system on tumour biology.

We apologise for not adequately stating the primary aim of our study. The aim is not to study "*why do these nascent tumors disappear*" as indicated by the reviewer. This intriguing question represented the focus of our previous collaborative work (Colom et al., Nature 2021)¹. In this previous publication, we established that emerging oesophageal tumours can be outcompeted by adjacent mutant clones, with no apparent contribution from the immune system.

Response to reviewers' Nature ms 2024-07-13618A-Z

In the current manuscript, instead, we focus on a distinct open question: what are the unique features of the few tumours that are not outcompeted from the tissue and that succeed in persisting long-term? In other words, here we study the surviving nascent tumours that remain capable of progressing further towards advanced stages. This has now been further reinforced in the revised version of the manuscript:

“As a result of this tumour sorting process, only a subset of the original tumours survives long-term, enabling us to study the mechanisms modulating pre-cancerous tumour persistence.”

In relation to the concern about the immune system, we recognise that we did not elaborate on the immune component and sincerely apologise for the lack of clarity on this. We completely agree with the reviewer that the immune compartment is a critical player contributing to carcinogenesis. Precisely for this reason, as indicated above, this aspect was explored in previous collaborative work (*Colom et al., Nature 2021*)¹. There, the same DEN model system presented here was used to study the mechanisms involved in the disappearance of nascent tumours, demonstrating that most incipient tumours were rapidly lost with no indication of an anti-tumour immune response. This included **no differences in early tumorigenesis in immune-deficient mice** (Extended data Figure 6 from *Colom et al. 2021*)¹. Instead, early tumour clearance was shown to be largely driven by the presence of neighbouring mutant clones with a competitive advantage. In light of these findings, in the current study, we decided to limit our focus on the contribution of immune cells.

However, given that our new work identifies the existence of two different early tumour subtypes (Niche+ and Niche-), we fully appreciate the importance of revisiting this point. Hence, we now provide new data describing the immune compartment in nascent Niche+ and Niche- tumours in the DEN model. These include:

- a. **Immune cells in Niche+/- tumours: Fig 3b, c and Extended Data Fig 4g** reveal no significant differences in the immune infiltrate in Niche+ and Niche- tumours at day 10 post-DEN treatment, reinforcing our previous observations (*Colom et al., Nature 2021*)¹.
- b. **Immune-deficient mice:** additional experiments show no difference in number of Niche+ and Niche- tumours that arise upon DEN carcinogen treatment in immune-deficient mice compared to wild-type (**Extended Data Fig 4h-l**). This indicates that immune response may not be the central player determining early tumour heterogeneity, i.e. nascent niche formation. We, however, noted a marginal decrease in tumour size at 10d post-DEN; an effect that lost relevance at later time points (2m post-DEN; **Extended Data Fig. 4l**).
- c. **Immune signature in surviving tumours: Fig 4b and Extended Data Fig 8** now include the immune cell characterisation of surviving tumours (8 months after DEN treatment). We defined immune cell subtypes that are enriched in surviving tumours compared to surrounding tissue. Our results align with previous work in that area². Shortly, T cell status in surviving tumours showed an increase in T effector and resident memory cells compared to surrounding control tissue. The increase in T regulatory cells hints an immunosuppressive environment. Monocyte activation resulting in dendritic cell infiltration as well as neutrophil recruitment indicates a transition to Type III immunity (**Extended Data Fig 8 e,f,g and Supplementary Table 10**).

2) That fibroblast supply growth factors for epithelial cells is not novel. And it is not surprising that a perturbed stromal microenvironment would have different properties than one that is unperturbed. Do the authors observe similar “tumor like” properties if they repeat these experiments using other stimuli, such as bleomycin-induced fibrosis or exogenous wounding?

Question 2. Reviewer #1

We fully appreciate the comment. As alluded by the reviewer, there is no doubt that epithelial cells and fibroblasts communicate with each other in response to tissue perturbations (such as during wound healing, cancer and *in vitro* conditions³⁻⁵ in an attempt to restore tissue integrity. Although previous studies have focused on epithelial-stromal interactions in early-stage tumours^{6,7}, the role of the mesenchymal compartment during tumour inception remains poorly understood. Our study is the first one to use a holistic mutagen approach to investigate the contribution of mesenchymal cells to tumour heterogeneity and survival at nascent pre-neoplastic stages (tumours of ~10-20 cells) on the verge of histological detection.

Indeed, in line with the classical Dvorak notion that tumours are wounds that never heal⁸, in our study we show that, upon lesion formation, the initial epithelial-mesenchymal interactions closely resemble those of an injury response. These include epithelia migration and hyperproliferation⁹, EGF ligand secretion^{10,11}, the role of SOX9 in epithelial-mesenchymal communication¹², as well as ECM remodelling and stromal fibronectin deposition¹³. However, given that fibroblasts in non-tumour areas do not recapitulate the features of nascent Niche+ tumours, we reasoned that the observed fibroblast changes are unlikely to be the direct result of the stress induced by the carcinogen treatment. This suggests there is a distinctive local response in the areas where tumour lesions emerge. We have revised our manuscript and included new data to highlight this point. These include:

- a. **Stromal changes restricted to tumour areas:** Quantifications of stromal cells in Niche+/- tumours as well as DEN-treated tissue **Fig 3b** (together with existing data **Extended Data Fig 2b** of the original manuscript) show that the stromal changes observed are specific to nascent tumours, in particular to Niche+ tumours. The stromal cell density in tumour-free DEN-treated tissue remains low (**Fig 3b**) and shows no significant increase in fibroblast proliferation (as marked by Ki67) (**Extended Data Fig 1j**).
- b. **Transcriptional analysis of adjacent tumour tissue:** scRNA-seq analysis comparing the transcriptional signature of nascent tumours, surrounding tissue and carcinogen-free controls shows that the stromal stress response observed in early tumours is different to that of the adjacent tissue (tumour-free carcinogen-treated tissue). For further specifics, a new **Supplementary Table 8** has been included to reflect stromal changes upon carcinogen exposure.
- c. **Fibrotic perturbation:** New bleomycin treatment experiments, known to promote fibrosis in epithelial tissues¹⁴, reveal that this fibrotic perturbation only leads to a marginal, almost negligible, increase in SOX9 expression in the oesophagus of DEN-treated mice **Extended Data Fig 10g**. These data indicate that the epithelial-stromal cross-talk observed in early tumours is not merely due to the presence of a fibrotic perturbed environment, but it is rather specific to pre-neoplastic lesions.

3) Supplemental Table 3 is informative and very appreciated. However, it does seem to suggest that only three total samples were submitted for 10X processing? If this is not the case, then this point should be made very clear and within/across group comparisons performed. Otherwise, these should be repeated with at least n = 3 per group. I assume samples were pooled in order to hit the minimum cell number targets, but this can be overcome through HTO-barcoding such that cells are labeled based on their tissue/tumor of origin prior to pooling and processing at the correct scale. For a study of this magnitude, I think multiple biological replicates are critical in order to ensure reproducibility.

Question 3. Reviewer #1

Response to reviewers' Nature ms 2024-07-13618A-Z

We thank the reviewer for raising this point and apologise for the lack of clarity. The information the reviewer is referring to is in the new **Supplementary Table 2**. This table shows the number of biological replicates and libraries (namely 5-7 libraries per condition).

We agree that HTO-barcoding is a powerful technique that would allow for samples to be pooled while retaining information on the tumour of origin. However, our micro-dissected tumours are formed by very few cells, many of which are lost throughout downstream processing. Therefore, any additional processing (antibody labelling, FACS, or HTO-barcoding) poses the risk for a complete loss of the sample. To put this into perspective, for our scRNA-seq analysis, we captured 50-650 cells per micro-dissected tumour. Overall, this represents an inevitable technical constraint inexorably linked to studying tumours at nascent stages. Hence, sample pooling was the only viable option to recover a satisfactory number of cells for subsequent analyses.

4) Supplemental Table 4 is inadequate and calls into serious question the authors' judgement in assigning cell types to each cluster, which is not well-described. There are numerous methods for automated classification, such as SingleR and scATOMIC, among others, but no such methods are discussed. Furthermore, based on the genes listed in Supp Table 4, it appears that the authors used a small number of "canonical" marker genes to make these decisions, which is an outdated way of thinking of scRNA-seq data. For example, they defined "myofibroblast" solely by ACTA2... despite this also being a marker for what the authors call "Myocytes", among others. It is also unclear why "myofibroblast" is a different category than "myofibroblastS". This is all highly problematic and undermines faith in the sophistication of the authors in this field. It is also not clear how the authors moved from 38 initial clusters to their final 11 "known biological cell types". Both cluster consolidation and cell type ascription should be more methodical and described in better detail. At the very least, a complete list of DEGs for the final cell types should be provided.

Question 4. Reviewer #1

We thank the reviewer and acknowledge the importance of their comments and suggestions. During our initial scRNA-seq analysis, we performed automated cell type annotation using PanglaoDB¹⁵ (4,909 markers for 163 cell types available in the database). This approach was satisfactory to annotate the major epithelial and stromal cell types. However, this annotation strategy failed to capture the full breadth of biological heterogeneity identified by our manual annotation.

Following the reviewer's suggestion, we have now revised our annotation strategy using CellTypist. This enabled us to provide an independent, classifier-based annotation using published reference atlases. This was combined with a LabelTransfer approach based on Seurat functions from previously curated publications^{2,16}. The new manuscript includes the revised automated annotation results in **Supplementary Table 3** and a description of the pipeline used in the methods section **Data Clustering, annotation and enrichment**. However, we have experienced that transferring cell-type labels from an annotated reference atlas to an unannotated dataset led to a lack of cell-type granularity, driven by experiment-specific nuisance factors and technology-dependent sparsity of gene expression. Hence, manual inspection based on prior knowledge remains an essential step in the annotation process, particularly considering that there are no previous scRNA-seq studies investigating incipient tumorigenesis in the gastrointestinal tract of the mouse at the single-cell level.

Response to reviewers' Nature ms 2024-07-13618A-Z

Regarding the specific annotation discrepancy highlighted by the reviewer, we apologise for the unintentional repetition of the myofibroblast marker *Acta2*, and for the confusion in relation to the marker gene list used for annotation in “**Supplementary Table 4**” of the original manuscript. A number of markers are present in more than one annotation list, given that most cell types cannot be assigned by the sole expression of specific genes. Instead, annotation is based on the co-expression of a subset of genes (e.g. *Acta2* in myocytes, pericytes and myofibroblasts). The revised version of the manuscript now includes a revised and improved presentation of the markers used for the visualisation of the cell annotation. This can be found under new **Supplementary Table 3**. Additionally, we have included the DEGs of each cell type; this was used to confirm annotation assignments (new **Supplementary Table 4**).

5) Similarly, the three “states” ascribed based on pseudotime analysis are not convincing. Although these states themselves are described well in Supp Table 6-8, how they were arrived at, using what cutoffs, and how the number (three) was determined is not clear. In the absence of a principled heuristic, it is not clear that correction for multiple hypothesis generation would make these findings non-significant relative to those potentially available.

Question 5. Reviewer #1

We thank the reviewer for their insightful comment. In the original manuscript, the formula DifferentialGeneTest and the top 1200 DEGs ranked by FDR-corrected p-value were used for the trajectory analysis (Monocle2). This information was included in methods in the “Cell transition trajectory analysis” section, and it is in line previous reports in oesophageal tumorigenesis². Regarding the three states, the number of branching points was automatically determined by the pipeline, without input from the user.

For a more thorough and up-to-date trajectory analysis, following the reviewer’s request (see **Question 6. Reviewer #1**), the revised manuscript uses Monocle3 to define the different cell “states” marking the transition of epithelial cells towards tumorigenesis (**Fig 4h,i, Extended Data Fig 9e-f**). The updated section **Cell transition trajectory analysis** can be found in **Methods** page 39, line 1-34.

6) The decision to use Monocle 2, instead of Monocle 3, is also unusual, since the more recent release has been available for nearly 4 years and fixes many of the over-conclusions associated with the earlier deployment. I would ask that the reviewers repeat their analysis using the more sophisticated version of this pseudotime package.

Question 6. Reviewer #1

We appreciate the expert advice of the reviewer, which has improved the presentation of our data and enhanced the clarity of our message. To address the reviewer’s point, Monocle 3 is now used to perform scRNA-seq trajectory analysis of epithelial cells, replacing the previous Monocle 2 version (**Fig 4h,i; Extended Data Fig 9e-f**; analysis description in “**Cell transition trajectory analysis**” **Methods** section).

The revised Monocle 3 analysis supports our original observations, reflecting their robustness. In line with the original manuscript, the new data also show a distinctive and unique basal cell trajectory in tumour samples that partially overlaps with control samples in differentiating cells. This trajectory analysis has enabled us to establish the heterogeneous nature of epithelial tumour cells in surviving (**Fig 4i-k; Extended Data Fig 9f-k**) and nascent tumours (**Fig 4l,**

Extended Data Fig 9j), as well as their cross-talk with stromal cells (**Fig 5a-c; Extended Data Fig 10a-b**). By combining Monocle 3 with CellChat analysis, we identify the EGF-SOX9-FN1 axis as a potential regulator of epithelial-mesenchymal communications in nascent tumours.

7) The rationale for drawing distinctions around Sox9 and Krt6a is not justified in the current manuscript. Are there clearly bimodal distributions for each? Where and how were the cut points drawn? Again, this way of conceptualizing NGS data in terms of small buckets based on arbitrarily binned expression of a minority of genes is not consistent with the current understanding of transcriptomics in cellular biology. This is not FACS, where surface protein expression may be accurately assessed, but rather probabilistic sampling where measured expression of the same RNA molecule from the same cell will vary considerably over time.

Question 7. Reviewer #1

We thank the reviewer for raising this point, and we apologise for not conveying the rationale for marker selection in a clearer manner. For the reviewer's reference, below we elaborate on how these markers were chosen in the original manuscript, primarily based on data analysis and prior knowledge. However, we fully acknowledge the concerns of the reviewer regarding the use of a minority of genes to establish relevant cell populations/states for downstream analysis. Hence, the original rationale has been revised in the current manuscript to establish relevant tumour cell populations/states more robustly. See details on former and current rationale below:

Original manuscript epithelial tumour cell selection:

- As indicated in the original manuscript, **Krt6a and Krt17** were used as “two established early tumour markers in the DEN carcinogen model” (*Alcolea et al., Nat Cell Biol 2014; Frede et al., 2026; Colom et al., Nature 2021*)^{1,17,18}. These two markers, together with gene enrichment analysis, were used to assign the transcriptional state defining early tumour cells both in the pseudotime trajectory and in the UMAP projection (**state 2; Rebuttal Figure 1a-e**). Please note that since tumours are largely microscopic, the punch biopsy approach used to enrich for tumour cells prior to scRNA-seq library preparation also contains adjacent non-tumour cells. Hence, the use of known markers, such as *Krt6a*, was extremely important to ensure that the scRNA-seq analysis focused on the right cell population, i.e., early tumour cells).
- **Sox9**, in turn, was identified by gene expression analysis along the original pseudotime trajectory (Monocle 2). Sox9 represented one of the top transcription factors showing an increased expression in state 2 tumour cells (**Rebuttal Figure 1f,h**), among other genes such as *Junb* and *Fosl1* (**Rebuttal Figure 1f**). Based on the literature, which describes the role of SOX9 in cancer and ECM remodelling¹⁹, and taking advantage of the fact that good antibodies to label the SOX9^{high} population were available, we decided to focus on the Sox9^{high} Krt6a^{high} population. Indeed, SOX9 protein expression was found to be enriched in nascent tumours (**10 days post-carcinogen treatment**) and showed a heterogeneous expression profile that positively correlated with the presence of the stromal niche, which made it an ideal candidate marker (**Rebuttal Figure 1g**).
- The **double positive population** for these marker genes (Sox9^{high} Krt6a^{high}) was determined based on each individual gene expression distribution. Cells over the inflection point above background expression levels were assigned as positive for both markers (**Rebuttal Figure 1i**). The assignment of Sox9^{high} Krt6a^{high} as nascent tumour cells was further validated by differential gene expression, which revealed that this double positive population recapitulates the transcriptional signature of the tumour state (**state**

2 in original manuscript), as well as by immunofluorescence staining (**Rebuttal Figure 1g**).

Revised epithelial tumour cell selection:

- **Tumour gene modules:** Following the reviewer's suggestion, we re-analysed the trajectory of the basal cell population using Monocle 3 (**Fig 4h,i; Extended Data Fig 9e,f**). This new pseudotime analysis not only resolved the main cell trajectories but also allowed us to identify specific gene modules for epithelial tumour cells (**Extended Data Fig 9f-g**). These modules, representing genes enriched along the pseudotime "Tumour branch," further revealed two distinct subpopulations within tumour sample. These populations were visualised in a basal cell UMAP space, as described in the revised **Methods** section **Cell transition trajectory analysis**. Please find quoted text below:

“To identify cells enriched for these programmes, we applied AUCCell (v1.28.0) to the raw count matrix from basal epithelial cells, ranking cells by their expression of Module 12 and Module 1 gene sets. Cells above the 80th percentile of the corresponding AUC distributions were selected, with Module 1 cells restricted to those not overlapping with Module 12. Populations assigned from the modules were then visualised on UMAP embeddings.”

- **Epithelial tumour cell heterogeneity:** The tumour specific gene modules resulting from the new pseudotime trajectory analysis revealed the presence of two major epithelial tumour sub-populations with a distinctive transcriptional signature (referred to as **Tumour 1** and **Tumour 12; Fig 4h-l and Extended Data Fig 9g-k**). Reassuringly, these two populations, now selected in an unsupervised manner, have been found to recapitulate features of those previously selected based on prior knowledge (i.e. *Krt6a* and *Sox9* expression) as per the original manuscript (**Rebuttal Figure 2**). In particular, the **Tumour 12** module is enriched in *Sox9*, as well as other developmental and stress-related genes, recapitulating the signature of the original *Sox9^{high}Krt6a^{high}* population. Accordingly, CellChat analysis predicts a higher interaction strength in the **Tumour 12** population.

Overall, our revised cell selection strategy established **Tumour 12** as the relevant epithelial population to study cell communication in persistent tumours. **Tumour 12** recapitulates the original *Sox9^{high} Krt6a^{high}*, showing an enriched expression of developmental and stress-related genes, including *Krt6a*, *Krt17*, *Sox9*, *Runx1*, *Egr1*, and *Areg*, also validated in nascent tumours. We believe that the scRNA-seq reanalysis and subsequent CellChat analysis provide further evidence that enhances the robustness of our results and strengthens the manuscript's conclusions.

Rebuttal Figure 1: Tumour-associated epithelial cell markers. **a**, UMAPs showing expression of the early tumour marker genes *Krt6a* and *Krt17*. Insets depict keratinocyte clusters split by condition: external control (Ctrl), internal control (DEN) and tumours. **b**, Basal keratinocytes ordered by Monocle2 along a pseudotime (PST) trajectory. The identified states (S1, S2, S3) and the expression of tumour cell markers *Krt6a* and *Krt17* are shown. **c**, Gene enrichment analysis of genes representing state 2, compiled from Gene Ontology Biological processes (GO:BP) and Reactome (REAC) terms and grouped based on biological knowledge; terms with p -value $< 1 \times 10^{-16}$ were ranked by their $-\log_{10}(\text{FDR})$ value. Representative genes are listed below each plot. **d, e**, UMAPs showing the cell distribution from different conditions (**d**) and distribution of the different states identified by PST analysis of basal keratinocytes (**e**; see **b**). **f**, Expression of transcription factors along the two pseudotime trajectories: from state 1 to either state 2 or state 3. Increased expression of stress-related transcription factors is observed in the tumour state (S2). **g**, Representative confocal images showing heterogeneous SOX9 expression (cyan) among *KRT6A*^{high} cells (red) in nascent tumours (10d post-DEN). DAPI, blue; cyan arrowheads mark SOX9-positive

cells; scale bar, 10 μm . **h**, UMAP showing expression of Sox9. Insets depict Sox9 expression in keratinocytes split by condition: external control (Ctrl), internal control (DEN) and tumours. **i**, Violin plots showing Krt6a and Sox9 expression in basal keratinocytes across conditions. Blue lines indicate thresholds for defining the Krt6a^{high} and Sox9^{high} (above the line), and Krt6a^{low} and Sox9^{low} (below the line) populations.

Rebuttal Figure 2: Tumour populations. Krt6a^{high} Sox9^{high} and Krt6a^{high} Sox9^{low} (left) and **Tumour 1** and **Tumour 12** (right) population distribution in the UMAP space.

8) Again, I fundamentally disagree with the manner in which the authors conceive of their cell subpopulations. For a manuscript aimed at studying the relationships between the epithelium (tumor and non-tumor) and the stromal microenvironment, there is no main-text mention of the three widely accepted “canonical” fibroblast type subtypes (iCAF, myCAF, apCAF), and these are only buried in EDF 8b-c without explanation or context. A closer interpretation based on the literature (PMC6727976, PMC5339682, etc) would be helpful.

Question 8. Reviewer #1

We thank the reviewer for their valuable feedback. We agree that the role of CAFs was not originally covered in sufficient detail. Our initial focus was driven by the fact that fibroblasts did not present a full CAF phenotype at pre-neoplastic tumour stages, consistent with previous reports²⁰. Considering the reviewer's suggestions, the new version of the manuscript revises this important point.

Despite our efforts, neither scRNA-seq nor immunofluorescence analyses enabled us to detect fully established CAF signatures in early DEN tumours. Although we observed a subtle upregulation of certain CAF-associated genes (including *Mfap5*, *S100a4*, *Vim*, and *Col1a2*; **Extended Data Fig 7k,l**), other canonical CAF marker genes (*Postn*, *Smock2*, *Spock2*, *Fap*, *Acta2*, *Il6*, *Cd74*, *Lrrc15*, *Cxcl12*; **Extended Data Fig 7m**) and their associated protein levels (FAP, α -SMA, VIM and FSP1: **Extended Data Fig 4f and 7n,o**) remained largely unaltered or non-detectable in surviving tumour fibroblasts. Accordingly, we did not observe the typical pro-tumorigenic increase in fibroblast proliferation (**Extended Data Fig 7p**).

Nevertheless, the characterisation of nascent and early tumour niche fibroblasts did reveal their marked pro-fibrotic nature (**Fig 4f and Extended Data Fig 7d-i**). This finding, together with the expression of nuclear YAP in niche fibroblasts of emerging tumours (**Extended Data Fig 4e**),

Response to reviewers' Nature ms 2024-07-13618A-Z

suggested the presence of a pre-CAF state²¹ that may be transitioning towards a myCAF identity. This has now been emphasised in the revised manuscript.

This CAF transition was further supported by the expression of CAF markers in DEN-derived invasive carcinomas. We show that the presence of an established CAF phenotype in upper GI tract tumours is stage-dependent. DEN-derived invasive carcinomas, sporadically emerging 9 months post-carcinogen treatment (**Extended Data Fig 1h**), express CAF markers (FAP, α -SMA, VIM new data in **Extended Data Fig 7q**). This newly presented data confirms that, although CAFs do not appear to be central players at nascent stages, cancer associated phenotypes become apparent as tumours progress.

Finally, we also thank the reviewer for providing literature on CAF subtypes in pancreatic ductal adenocarcinoma (PDAC). While there is a large field of work around CAFs on PDAC, this cancer type is known to be particularly fibrotic in comparison to other solid tumours²² and hence may not be sensible to compare it to our upper GI tumour model. Consequently, to characterise the fibroblasts present in our scRNA-seq data, we focussed on oesophageal cancer studies^{2,20}, which we believe would more closely reflect the CAF-associated phenotype expected in our model system. Our tumour fibroblast annotation was further informed by the overall literature on CAFs, covering markers identified across different models^{23,24}. Notably, in oesophageal tumour studies, CAF subtypes have only been found in moderate/severe dysplasia²⁰. Hence, since our study focuses on even earlier stages, i.e. hyperplastic/low-grade dysplastic stages (pre-neoplastic/pre-cancer tumours; **Extended Data Fig 1g,h**) well before cancer *per se*, it was not entirely surprising that CAF features were not found to be fully established. Precisely, we believe that here is where the relevance of our study resides; the fact that fibroblasts promote nascent tumour survival even before acquiring a full-blown CAF phenotype is not only novel but rather remarkable.

9) In Figure 4a, instead of referring to fibroblasts based on their cluster number and relative PDGFRA expression, it would be more helpful to examine canonical iCAF markers such as IL6, CXCL12; myCAF markers such as LRRC15, ACTA2; and apCAF markers such as CD74.

Question 9. Reviewer #1

We thank the reviewer for this suggestion. We agree with the reviewer that this would have been the ideal way to present the different fibroblast subtypes if they had been represented in our dataset. However, as stated in response to remark #8, our pre-malignant tumours did not show evidence of an established CAF signature (**Extended Data Fig 7**). In **Rebuttal Figure 3** (below), we provide plots showing the expression of canonical CAF markers, as requested by the reviewer, across different conditions in each fibroblast cluster.

Since we did not identify a distinct CAF population, but rather observe differential contributions of the *Pdgfra*^{low} and *Pdgfra*^{high} fibroblast populations to the early tumour niche (as shown by lineage tracing, histology, and the analysis of our scRNA-seq data), we considered it justified to focus on these two populations for our downstream communication analyses (**Fig 5a-c**; **Extended Data Fig 10a,b**). For further transparency, the expression levels of the marker genes proposed by the reviewer are individually shown in *Pdgfra*^{low} and *Pdgfra*^{high} fibroblast populations and across conditions (**Rebuttal Figure 4**).

Rebuttal Figure 3: Typical cancer-associated fibroblast subtype markers in fibroblast clusters across conditions. Violin plots illustrating log-transformed normalised expression levels of typical CAF subtype markers (myCAF: *Acta2*, *Lrrc15*; iCAF: *Il6*, *Cxcl12*; apCAF: *Cd74*) in control, DEN (internal control) and tumour conditions across the two defined fibroblast populations.

Rebuttal Figure 4: Typical cancer-associated fibroblast subtype markers in *Pdgfra*^{low} and *Pdgfra*^{high} fibroblast populations across conditions. Violin plots illustrating log-transformed normalised expression levels of typical CAF subtype markers (myCAF: *Acta2*, *Lrrc15*; iCAF: *Il6*, *Cxcl12*; apCAF: *Cd74*) in control, DEN (internal control) and tumour conditions across the two defined fibroblast populations.

10) Figure 3h, or its associated supplements, should contain representative control images.

Question 10. Reviewer #1

We thank the reviewer for pointing this out. Tumour-adjacent, and tumour-free control areas showing undetectable SOX9 expression (as per immunolabelling) were originally included in the manuscript (current figures: **Extended Data Fig 9n**). We apologise for the lack of clarity on this point. To better reflect the tumour marker specificity, we have now revisited this point and added new data showing expression of SOX9 and KRT6A, as well as of additional early tumour markers (KRT17, AREG, EGR1 and RUNX1) in morphologically normal DEN-treated tissue and untreated controls (new **Extended Data Fig 9j**).

11) I question the authors' judgement in distinguishing between high and low expression of the continuative fibroblast marker PDGFRA. Even if so, I think it is presumptuous to assume that proximity begets causal interactions.

Question 11. Reviewer #1

We appreciate the reviewer's concern. The existence of these two discrete populations became apparent upon immunolabelling tissue wholemounts and cross-sections of both control and tumour samples, before performing our scRNA-seq analysis (**Fig 3a,e,f; Extended Data Fig 4a-d**). Our histological characterisation not only revealed the presence of these two distinct populations of fibroblasts, but also their compartmentalisation in the tissue, with PDGFR α ^{high} fibroblasts residing in the submucosa (lower stroma) and PDGFR α ^{low} fibroblasts in the lamina propria (upper stroma; **Extended Data Fig 4c,d**). Hence, we were not entirely surprised when our scRNA-seq analysis revealed clear differences in the expression level of *Pdgfra* across fibroblast clusters (**Fig 4c**).

The text has been amended to reflect that the distinctive PDGFR α expression was a histological empirical observation that guided our scRNA-seq analysis. In our opinion, the clear histological segregation of these two fibroblast populations, together with their distinctive morphological and transcriptional signature, as well as the enrichment of *Pdgfra*^{low} fibroblasts in early DEN tumours (**Fig 4c,d; Extended Data Fig 7a-d**), warranted further investigation. Even more so, given that our data did not segregate fibroblasts based on a CAF signature.

This choice was further informed by unbiased lineage tracing of fibroblasts showing that the uppermost PDGFR α ^{low} fibroblast population of the lamina propria primarily contributed to the formation of the early tumour niche (**Fig 3d-f; Extended Data Fig 5**). Our observations on heterogeneous PDGFR α fibroblast populations are in line with previous studies in the gastrointestinal tract, where PDGFR α ^{high} and PDGFR α ^{low} compartmentalisation in stromal fibroblasts has also been reported in connection to their role in epithelial morphogenesis and cross-compartment communication²⁵⁻²⁸.

Response to reviewers' Nature ms 2024-07-13618A-Z

Since increased fibronectin expression was found to be one of the key differences between PDGFR α ^{low} and PDGFR α ^{high} populations (**Fig 4d, Extended Data Fig 7a,b**), we have now included new data validating fibronectin expression in *Pdgfra-CRE^{ERT} Rosa26^{Confetti/wt}* labelled fibroblasts from lamina propria and submucosa, respectively (**Fig 4e**).

12) For the fibroblast enrichment in Figure 4e, the authors should examine at a per-sample level the percent expression of their “PDGFRA-low” clusters and assess statistical significance.

Question 12. Reviewer #1

We apologise for missing to add the statistics for the *Pdgfra*^{low} clusters proportions across conditions. The revised version of the manuscript includes a detailed statistical analysis in **Extended Data Fig 7c**. In particular, we used one-sided chi-squared test to determine if a cell cluster is significantly enriched in a specific condition.

13) The authors talk about going from 38 initial clusters to 11 final named clusters, but then on the second paragraph of Page 11 (“For this we focused on cluster 23 (C23)), they appear to be referencing back to the 38 cluster model? Is this correct? If so, they should provide some reference to where these cells are. Which of the final 11 cell types do these fall into?

Question 13. Reviewer #1

Apologies for the lack of clarity, in the original manuscript, we did provide a correspondence between cells annotated as fibroblasts (1 of 11 cell types identified) and the initial 38 clusters. This can be found in **Fig 4c** (original **Fig 4a** for the reviewer's reference). For further clarity, we have now included a table detailing the cluster-to-cell type assignment (**Supplementary Table 3**). Additionally, we elaborated on how this was performed in the methods section, as well as **Question 4 Reviewer #1**.

14) The CellChat analysis adds almost nothing to the manuscript and is greatly over-concluded. This statement is incredibly misleading and should be removed: “Overall, our scRNA-seq and communication analyses uncovered the pro-fibrotic/wound-healing transcriptional profile characteristic of tumour niche fibroblasts”. The CellChat methods also contain insufficient detail.

Question 14. Reviewer #1

In our view, the CellChat analysis was critical to further discern how nascent tumour epithelial cells and fibroblasts communicate. Reassuringly, after our comprehensive scRNA-seq reanalysis, the CellChat analysis has also been revised, supporting our original observations. In this sense, EGFR-mediated communication, which has gained prominence as an important signalling pathway modulating mutant cell competition even before tumours emerge²⁹, has been identified in our CellChat analysis as one of the strongest pathways by which tumour cells (**Tumour 12** signature; Sox9^{high}) and niche fibroblasts (Cluster 19 *Pdgfra*^{low}; profibrotic signature) communicate. Such conclusions could not be reached by differential gene expression analysis alone nor it would have highlighted the heterogenous nature of epithelial tumour cell communication.

15) *The in vivo knockout and inhibition experiments are probably the most compelling portions of this manuscript, but are sparsely covered both in terms of text and figure panels. Additional description of these very interesting results would be warranted.*

Question 15. Reviewer #1:

We appreciate that the Reviewer finds our functional studies to be “*very interesting*” and “*compelling*”.

We apologise for the lack of functional depth. A more detailed description on functional aspects has been added to the revised version of the manuscript. Additionally, new data now validates the original claim that EGF-SOX9-Fibronectin signalling governs epithelial-stromal communication and tumour persistence at the most incipient stages of tumorigenesis. Please find below a description of the new experiments included in the revised manuscript:

- a. **EGFR contributes to fibroblast recruitment and stromal remodelling:** since our CellChat analysis predicted epithelial-fibroblast communication via EGFR in tumours, we decided to validate this more extensively. We originally proposed that AREG (an EGFR ligand) acts as a fibroblast chemoattractant, potentially promoting ECM deposition and niche formation. To further support this, new data now show that signals from regenerative 3D epithelial organ cultures³⁰ favour the self-organisation of adjacent fibroblasts and ECM remodelling in an EGFR dependent manner. In particular, we found that, when co-cultured with 3D epithelial cells, fibroblasts segregate into PDGFR α ^{low} (Fibronectin producing fibroblasts) adjacent to the epithelia, and PDGFR α ^{high} (Fibronectin low) in areas distant from the tissue (**Fig 5g; Extended Data Fig 10e**). Remarkably, this *in vitro* assay recapitulated the architecture of the *in vivo* stromal compartment just by exposing fibroblasts to expanding epithelial cells (SOX9+). Fibroblast self-organisation, as well as ECM deposition and remodelling were prevented by EGFR inhibition with Gefitinib (**Fig 5h; Extended Data Fig 10e**). Overall, this *ex vivo* experiment demonstrates that epithelial cells stimulate fibroblast pro-fibrotic nature and ECM remodelling via EGFR signalling.
- b. ***In vivo* EGFR inhibition reduced niche+ tumour burden:** the revised manuscript includes new data showing that *in vivo* treatment with Gefitinib (EGFR inhibitor) leads to a reduction in early tumour burden (10 days post-DEN) (**Fig 5i,j**)
- c. **EGFR modulates SOX9 expression:** Interestingly, Gefitinib treatment was found to reduce SOX9 expression in epithelial cells (**Extended Data Fig10d**), suggesting that either EGFR signalling may be upstream of SOX9, or responsible for perpetuating SOX9 expression.
- d. **SOX9 expression impacts niche+ tumours:** previously, we had shown that SOX9 depletion in *Krt14-Cre^{ER}; Sox9^{fllox/fllox}* DEN treated mice led to a significant reduction in tumour burden (**Fig 4r; Extended Data Fig 9o**). New experiments included in the revised manuscript now show that SOX9 deletion also leads to a reduction of nascent Niche+ tumour size (10 days post-DEN treatment), reaching a similar size to Niche- tumours. Moreover, SOX9 deletion rendered significant changes in Niche- tumours (**Extended Data Fig 9p**).

Referee #2 (Remarks to the Author):

This manuscript by Skrupskelyte et al aims to better understand how the pre cancer niche provides initial steps for tumor formation and contributes to its progression and survival. The authors use an elegant way to induce and track pre-neoplastic squamous tumors in the upper gastrointestinal mouse tract utilizing different imaging techniques such as confocal microscopy and SHG, combined with scRNA-seq and 3D cultures. They identify niche+ tumor structures that are enriched with stromal remodeling fibroblasts which affect keratinocyte proliferation and promote long term survival. Transcriptomic analysis identified a Sox9high cancer cell state that promotes fibroblast migration by EGF signaling, leading to production of fibronectin by the fibroblasts.

The model used is interesting and the notion of studying early events is important. It is not clear at all though if this has significant human relevance since there is no human validation. Moreover, the authors claim that the cells they study are early CAFs but do not express classic CAF genes when in fact they do express what would appear to be a myCAF signature. It is true that the marker genes are not the classic. However, this could again be a consequence of the model being a model without human validation. Moreover, though the authors collected different tumor niches within each GI tract, the number of biological replicates (i.e. different mice) along the manuscript is relatively low (n=4). The authors must add human validation of at least the Sox9, AREG and FN connection, as well as their subset markers. Without these the observations are interesting, but the novelty is limited, since it is already well established that tumor development is modulated and perhaps even initiated by changes in the tumor microenvironment and specifically by fibroblasts (Sahai et al, 2020, de Visser and Joyce, 2023).

Reviewer #2 Authors' response: We thank the reviewer for the overall positive comments, highlighting the “*elegant*”, “*interesting*” and “*important*” nature of our work. We are also grateful to the reviewer for directing our attention towards aspects of the data that need further consideration to strengthen the claims of our study.

Major comments:

1a. It is well known that the TME changes at early stages. The changes shown here are still generated by the carcinogen and cancer-stroma signaling so not totally surprising or novel. Would these occur without cancer cells? Is the stroma different before the cancer cells are there? It is very hard to tell but maybe treating the stromal matrices with DEN would provide some answer.

Question 1a. Reviewer #2:

We appreciate the reviewer's comment and agree that the relevance of the TME in early tumorigenesis has been recognised. However, to our knowledge, the nature of tumour-stromal interactions at nascent tumour stages is currently not known. The immense majority of scientific work in early cancer has been done using grafting assays of established and advance cancers and/or by inducing potent cancer driver mutations throughout a significant proportion of the tissue^{31,32}. These models, although important to advance our knowledge, are far from truly recapitulating the initial stages of tumour formation. Our model, instead, studies the emergence of sporadic tumours in a mutant field created by treating the tissue with a tobacco-derived

carcinogen, which we have shown recreates the mutational burden found in aged human oesophagus (Colom et.al. Nature. 2021)¹. Thereby, this approach recapitulates early tumorigenesis in a more physiologically relevant context and allows us to focus on early tumours from their inception, when they are only ~10 cells in size. The novelty of our work lies in the heterogeneity of nascent tumours detected in the DEN model, which are distinguished by the presence or absence of a supporting stromal niche. In particular, the observation that nascent tumour fibroblasts contribute to the formation of the tumour niche (at pre-neoplastic stages) and effectively prime tumours for persistence/survival even before acquiring a fully established cancer-associated fibroblast (CAF) phenotype paves the way to understand the interactions that occur between a nascent tumour and its surrounding stroma. We apologise for failing to convey this message in the previous form of the manuscript. The point has been further elaborated and clarified in the revised manuscript.

Regarding the reviewer's concern that the carcinogen (DEN) treatment may directly impact the stromal compartment, our data argue that the fibroblast changes that we observe in early tumours are unlikely to be induced by the direct effect of the carcinogen treatment on fibroblasts/stroma alone (see details below). Since humans are exposed to carcinogens systemically for most of our lives, we agree that this is an important point. We have amended the manuscript to reflect and emphasise this.

See below a detailed account of new and existing data demonstrating that DEN does not appear to exert its niche remodelling effect by directly impacting the stromal compartment:

- a. **No impact on stromal mutational burden:** In-depth DNA sequencing of tumour epithelium and stroma (separately) shows that, while the tumour epithelium presents mutations in cancer genes (Colom et.al. Nature. 2021), there are no significant changes in the mutational burden of stromal cells neither in tumours nor in adjacent tumour-free DEN-treated tissue (**Extended Data Fig 7rs**).
- b. **Stromal remodelling is exclusive of tumour areas:** we show that the fibroblast-associated remodelling occurs specifically underneath nascent tumours, with the surrounding tissue (referred to as DEN) showing a normal phenotype (**Fig 1c, 4f; Extended Data Fig 7e**). This reflects a distinctive cell behaviour in the areas where tumour lesions emerge. If DEN was to directly induce fibroblast remodelling, one would expect stromal remodelling to also occur in tumour-free regions. For further evidence:
 - i. **Fibroblast density:** We have added new data showing the quantification of fibroblasts in Niche+ *versus* Niche- tumours, as well as in adjacent non-tumour tissue (10 days post-DEN). The data show that fibroblasts are specifically enriched in Niche+ tumours, without significantly impacting other areas of the tissue also exposed to the DEN carcinogen (**Fig 3b**).
 - ii. **Fibroblast proliferation:** **Extended Data Fig 1j** also shows that there is no proliferative advantage in fibroblasts treated with DEN as opposed to control tissue, arguing against their potential activation upon DEN treatment alone.
 - iii. **Lineage tracing:** Fibroblast lineage tracing data (**Fig 3d,e; Extended Data Fig 5e-i**) shows no significant differences between control (untreated tissues) and DEN (non-tumour areas in treated tissue); in contrast, the tumour niche is considerably different.
 - iv. **Fibronectin expression:** **Fig 4f** and **Extended Data Fig 7e** show fibronectin enrichment is localised to the niche when compared to DEN-treated (tumour-free) stroma.
- c. **Niche formation is promoted by the tumour epithelium:** Since mutations in the epithelium are the primary cause of tumour emergence, we hypothesise that early

Response to reviewers' Nature ms 2024-07-13618A-Z

tumour stromal remodelling is likely induced by the early tumour epithelium. Additional data supporting this claim include:

- i. New and existing experiments showing that EGF signalling from perturbed epithelial cells directly impacts fibroblast remodelling (**Fig 5f-j; Extended Data Fig 10e**).
- ii. New and existing experiments showing that the expression of SOX9 promotes the Niche+ tumour phenotype (**Fig 4m; Extended Data Fig 10d,e**).

Overall, although no morphological, histological or genetic changes were observed in the stroma of tumour-free DEN-treated tissue, we acknowledge that the DEN treatment, as any systemic treatment, is deemed to impact both the epithelium and the stroma to some extent. For reference, the revised manuscript now includes detailed data reporting changes in stroma treated with DEN. Differential gene expression can be found in newly provided **Supplementary Tables 5,6**.

Finally, we would like to note that DEN treatment is not amenable for *in vitro* experiments, as this compound needs to be metabolised to be active. Experiments in the lab have been attempted before confirming that his approach is not viable as of yet.

1b. Following up on this, in figure 3 the authors show that exposure to tumor stroma makes normal epithelial cells proliferate. This is not surprising and has been shown numerous times already. Would it happen similarly though from niche+ vs niche- stroma? Or when immune cells are depleted? In Figure 3o,p- The authors used mouse model to deplete Sox9 keratinocytes and measured the number of tumors 1-month post DEN treatment. On which tumors did you look at? Given the number of niche+ tumors is kept stable and low along time in DEN treated mice (Figure 1d), In order to prove Sox9 is a relevant player in epithelial-stromal communication from the most incipient stages of tumorigenesis, the authors should measure the number of niche+ tumors and niche- tumors separately, instead of measuring the number of all tumors together. Otherwise, the authors can't exclude the fact that the reduction in tumor burden originates from the decrease in niche- tumors as shown in figure 1d.

Question 1b. Reviewer #2:

We thank the reviewer for these insightful comments. Below, we address individually each of the points raised:

-Question 1b (i) [first sub-question]: In relation to the pro-proliferative effect of tumour fibroblasts on epithelial cells, we acknowledge that the general effect of the tumour stroma on epithelial behaviour is well-established. However, little is currently understood about how these interactions operate in nascent tumour stages. In our study, we address this gap by showing that the heterogeneous nature of epithelial-mesenchymal communication during early tumorigenesis can shape tumour's fate at later stages. The tissue recombination assay represents one of several experiments exploring this question (**Fig 2; Extended Data Fig 3**).

Regarding the suggestion to compare Niche+ vs Niche- stroma, we would like to clarify that in our experiments only Niche+ tumours were used. Technically, it is not possible to perform this experiment with Niche- tumours, given their lack of apparent morphology. Niche- tumours can only be visualised under a confocal microscope in fixed and immunostained tissues, and not in fresh tissues under a dissecting microscope. Notwithstanding, our results indicate that Niche+ tumour cells possess an increased proliferative capacity *in vivo* (**Fig 1e; Extended Data Fig 2b,c,e**), which is associated with a greater tumour growth compared to Niche- tumours (**Fig 1f,g**;

Extended Data Fig 2f). Hence, in tissue recombination experiments, we would anticipate that Niche- tumours would not confer any tumour properties to control epithelial cells (i.e., from untreated animals).

Regarding immune cell contribution, the revised manuscript includes new data to address the reviewer's request:

- a. **Immune cells in tumours: Fig 3b** shows no immune cell enrichment in Niche+ compared to Niche- nascent tumours.
- b. **Tumour formation in immune-deficient mice:** In a new experiment, immune-deficient animals³³ (*NOD.Cg-Prkdc^{scid}Il2rg^{tm1Wjl}* SzJ mice) were exposed to the DEN carcinogen to assess early tumour formation 10 days and 2 months post-DEN treatment. Neither tumour phenotype, nor size or number were significantly different between immune-deficient and wild-type age-matched mice (**Extended Data Fig 4 h-l**).

-Question 1b (ii) [second sub-question]: To clarify the impact of SOX9 on tumour niche heterogeneity, we have now split our quantifications into Niche+ and Niche- tumours, and included new data showing that SOX9 deletion in basal epithelial cells indeed reduced the size of Niche+ tumours to values matching those of Niche- tumours (**Extended Data Fig 9p**). These observations reinforce the claim that SOX9 expression in epithelial tumour cells promote the Niche+ tumour phenotype. Please note that, as previously reported, SOX9 expression is necessary for initial tumour formation¹⁹. Our observations indicate that the latter also holds true in the DEN model.

2. Much of the analysis is based on 3-4 mice per group. The experiments in Figure 1, for example, show dozens of dots in panel g but these are all from 3 mice. The authors must take into consideration the fact that different tumors in the same mouse are expected to be more similar or co-dependent than tumors from different mice.

Question 2. Reviewer #2:

We appreciate the reviewer's concern about biological replicates. However, we would like to stress that the presented DEN model enabled the detection of hundreds of localised and well-defined 'micro-tumours' in each oesophagus (average surface area analysed: 30mm²) after DEN withdrawal (as characterised in Colom et.al. Nature. 2021)¹. The number of tumours assessed for each dataset can be found in the associated source data. **Please refer to detailed rationale below for further specifics.** Overall, assessing a large number of discrete and independent tumours within the same native environment has allowed us to gain a deeper understanding of the true heterogeneous nature of nascent tumours. However, to address the reviewer's concern, the updated manuscript includes a larger number of replicates for our *in vivo* validation experiments (**Fig 5i,j**). Please note that co-dependency has always remained a consideration; for this reason we use both internal control groups (DEN non-tumour areas) within each mouse, as well as untreated control animals. Statistical tests are always conducted across biological replicates, as indicated in the methods section (**Statistics and reproducibility**).

Biological variability considerations in early tumours

Since early tumours undergo a dynamic selection process (elimination *versus* persistence), to understand this process in-depth, it is critical to control for their native environment. This can be achieved by analysing the large number of well-localised, independent tumours that emerge within each animal in this particular model (as seen in new **Extended Data Fig 1i**). This reduces

the inter-animal variability that is inevitably present in our experiments, enabling us to understand the evolution of nascent tumours within the same environment (see details in Colom et.al. Nature. 2021)¹. This analysis would resemble that of individual clones in lineage tracing analysis, where every emerging clone is an independent entity, whose behaviour is influenced by the systemic environment of each individual animal (see **Rebuttal Figure 5** depicting reproducibility across biological replicates). This approach cannot be compared to other cancer/tumour models driven by a specific experimental approach, such as a given driver mutation or grafts of advanced cancer cells. Such models tend to develop rapidly growing and merging tumours across the tissue, without well-defined margins. We hope the reviewer appreciates that part of the uniqueness of this model resides on its early tumour evolution nature: mutations are first induced by a clinically relevant cigarette smoke carcinogen (a major risk factor for GI cancers), and independent tumours are longitudinally assessed thereafter as they emerge and evolve. Overall, based on our observations across multiple independent cohorts (see **source data** for details), we can confirm that the tumorigenesis process driven by the DEN carcinogen is highly reproducible and are confident that our conclusions are well supported.

Rebuttal Figure 5: Niche- and Niche+ tumour diameter in animals of distinct ages. Tumour diameter (μm) at the indicated timepoints; tumours from each animal are plotted separately for each timepoint. Blue dots, Niche- tumours; red dots, Niche+ tumours. Data is expressed as mean+s.e.m; n=3 mice per group. Each dot represents a tumour.

3. The Sox9 connection is very interesting, but it is not clear what made the authors choose this factor originally. Was it one the most DE genes? Or educated guess? Would other cancer genes also show this close to stroma pattern? Also, the authors say that Sox9 is high close to stroma but in fact most cells in fig 3m are Sox9 negative. Once again, the statistical analysis in 3n relies on many tumors albeit in very few mice.

Question 3. Reviewer #2:

We thank the reviewer for pointing us to the need for further clarifying our marker selection process. Please see answers to the different points raised by the reviewer below:

- a. **Original SOX9 marker selection:** In the original manuscript SOX9 was identified as a result of our gene expression analysis along the pseudotime trajectory. There, *Sox9* represented one of the top differentially expressed transcription factors (TFs). Other differentially expressed TFs included *Egr1*, *Junb*, *Fosl1*, and *Atf3*. Based on our observations, together with the literature^{12,19}, which describes the relevance of SOX9 in cancer and stroma/ECM remodelling, we selected SOX9 as a proxy to mark the relevant tumour population for further validation. Indeed, at the protein level, SOX9 was found to be enriched in nascent tumours 10 days after DEN (**Fig 4l**); it also showed a heterogeneous expression profile that positively correlated with the presence of the stromal niche, which made it an ideal candidate marker (**Fig 4m,n**). In the revised manuscript, we have now revisited this to establish relevant tumour cell populations/states more robustly (**see point b**).
- b. **Revised epithelial tumour cell selection:** We have now reanalysed the pseudotime trajectory using Monocle 3 (**Fig 4h,i; Extended Data Fig 9e,f**). The revised trajectory analysis has enabled us to identify two gene modules specific to epithelial tumour cells, based on the top 1500 most variable genes across the trajectory (See updated **Methods** section **Cell transition trajectory analysis**). Further mining of these tumour specific gene modules to identify corresponding populations revealed the presence of two major epithelial tumour sub-populations with a distinctive transcriptional signature (referred to as **Tumour 1** and **Tumour 12**; new and revised data **Fig 4 j-l** and **Extended Data Fig 9e-k**). Reassuringly, these two populations were found to recapitulate features of the populations previously selected based on prior knowledge (i.e. *Krt6a* and *Sox9* expression) as per the original manuscript (**Rebuttal Figure 2**). In particular, the **Tumour 12** population is enriched in *Sox9*, as well as other developmental and stress-related genes, recapitulating the signature of the original *Sox9*^{high}*Krt6a*^{high} population. We believe that our new scRNA-seq analysis and assignment of populations for downstream CellChat further enhance the robustness of our results and strengthen the manuscript. Given the availability of reliable antibodies to label and examine SOX9^{high} cells, we have chosen to continue using the same proxy markers (*Sox9*^{high} *Krt6a*^{high}) to define the relevant tumour population in tissue immunostainings, thereby facilitating downstream validation experiments.
- c. **Clarification on Figure 3m** (current **Fig 4o; Extended Data Fig 9n**): We apologise for the lack of clarity. This figure aims to assess whether SOX9^{high} epithelial cells promote stromal reorganisation in a tumour-free context. To this end, we took advantage of sporadic clusters of KRT6A+/SOX9+ cells found in phenotypically normal (non-tumour) areas of DEN-treated tissue, likely marking prospective tumour cells prior to lesion formation. For this reason, most cells in this figure are SOX9 negative. The clusters are stress areas showing sporadic cells expressing high levels of SOX9. Indeed, the results indicated that SOX9 expression elicits changes in mesenchymal cells, irrespective of whether they belong to tumours or not. This has been clarified in our revised version of the text.

“...we took advantage of sporadic clusters of KRT6A+/SOX9+ cells found in phenotypically normal (non-tumour) areas of DEN-treated tissue, potentially marking prospective tumour cells prior to lesion formation. Isolated SOX9+ cell clusters showed signs of fibroblast attraction, presenting closer proximity to fibroblasts.”

New functional *in vitro* experiments further reinforce this claim and show that interfering with the EGFR-SOX9 signalling axis impacts fibroblast reorganisation (**Fig 5f-h; Extended Data Fig 10d,e**).

- d. **Would other cancer genes also show a similar pattern?** As outlined in our previous response, SOX9 was initially used as a proxy to identify the relevant tumour cell population. However, these cells (both the current **Tumour 12** population, as well as the original SOX9^{high} population) also show elevated expression of developmental and stress-associated genes, including *Runx1*, *Egr1*, and *Areg*, among others (**Fig 4j,l; Extended Data Fig 9h-j and Supplementary Table 13**).

4. How is niche+ vs niche- defined? A threshold of expression of specific factor(s)? This is not defined. Moreover, the authors use PDGFR-alpha to define fibroblasts in Figures 1-3, but then in the scRNA-seq analysis they show that in fact there is a PDGFR-alpha high and low group. So how can this be used as a pen marker for fibroblasts in the first figures?

Question 4. Reviewer #2:

We thank the reviewer for raising this point and apologise for the confusion. Please see answers below:

- a. **Niche+ and Niche- definition:** Niche+ and Niche- tumours were defined by the presence/absence of a nest-like structure composed of stromal fibroblasts (PDGFR α +). From our observations, we determined that a structure emerges beneath nascent tumours 10 days post carcinogen treatment and becomes immediately apparent upon confocal inspection of immuno-labelled tissues. Compared to control or Niche- tumours, where the basement membrane is smooth, the tumour niche displays marked irregularities as niche fibroblasts extend cytoplasmic protrusions that reach the epithelial surface; a feature not observed in normal surrounding tissue (**Fig 1c; 3a,f; Extended Data Fig 4c**). Thus, the presence/absence of a niche is not determined based on the expression level of specific factors, but on prominent morphological changes in the stroma. For clarity, this has been described in the new version of the manuscript.
- b. **PDGFR α expression in stromal fibroblasts:** We apologise for not clearly conveying this point. The revised manuscript now includes a better description of the use of PDGFR α as fibroblast marker.

Despite being a well-established pan-fibroblast marker^{4,25,26,34}, PDGFR α expression levels showed an apparent compartmentalisation in tissue wholemounts and cross-sections (**Fig 3a,e,f; 4m; Extended Data Fig 4a-d**). As indicated in the original and revised manuscripts, we observed that PDGFR α ^{high} fibroblasts reside in the submucosa (lower stromal layer) and PDGFR α ^{low} fibroblasts in the lamina propria (stromal layer underneath the epithelium; **Extended Data Fig 4d**). Our observations of fibroblast populations with distinct expression levels of PDGFR α are in line with previous studies in the upper and lower GI, where PDGFR α ^{high} and PDGFR α ^{low} compartmentalisation in stromal fibroblasts has also been reported in connection with their role in epithelial morphogenesis and cross-compartment communication²⁵⁻²⁷.

Hence, we were not entirely surprised when our scRNA-seq analysis revealed clear differences in the expression level of *Pdgfra* across fibroblast clusters (**Fig 4c**). The text has been amended to reflect that we first observed differences in PDGFR α levels empirically in our histological analyses, which in turn guided our scRNA-seq analysis. We apologise for not clarifying in the original version.

Please note, that, albeit heterogeneous expression, PDGFR α still represents a pan-fibroblast marker, and it is used as such in our study. For illustrative purposes (e.g. **Fig 1b,c**), high magnification images focusing on the early tumour niche directly underneath tumours, i.e. localised in the upper stroma (lamina propria), were acquired with exposure settings adjusted to the expression levels of PDGFR α in that tissue layer. The latter do not

necessarily reflect the distinctive PDGFR α expression levels, but such adjustments in imaging settings were necessary to properly illustrate changes in the stromal architecture and niche remodelling. All images including both stromal compartments (lamina propria and submucosa) do reveal the heterogeneous expression of this marker. We have now amended the text, figure legends and Supplementary methods to indicate this imaging setting adjustment. The text reads: *“For clarity, exposure settings were adjusted to PDGFR α ^{low} population when images were focussed on lamina propria alone and adjusted accordingly when both lamina propria and submucosa (PDGFR α ^{high}) populations were displayed.”*

5. The authors compare the niche fibroblasts to known CAF signatures and find very little overlap. Did they try to look at full blown tumors in their model? Or in other mouse models of esophageal cancer? It is very much possible that the specific markers here are different and that these are in fact myCAFs as also appears from the ligand-receptor analysis.

Question 5. Reviewer #2:

We fully concur with the reviewer in that tumour-associated fibroblasts resemble the myCAF signature in our scRNA-seq data (**Fig 4c-f; Extended Data Fig 7a-m**). There is a subtle increase in myCAF-associated genes, including *S100a4*, *Vim*, *Mfap5*, *Col1a2*, compared to control conditions (**Extended Data Fig 7k,l**). Additionally, the scRNA-seq and ligand-receptor analysis reveals a signature associated with ECM production and remodelling, again resembling the myCAF function. In contrast, the protein levels of CAF markers (VIM, FAP, FSP1) remained largely unaltered or non-detectable in the niche of surviving pre-neoplastic tumours (**Extended Data Fig 7n,o**). Overall, these observations indicate that, although these fibroblasts might be primed to become myCAFs, such phenotype has not yet been fully established. The stage-dependent transition towards CAFs, was further supported by new experiments showing that advanced cancer stages in the DEN model (invasive carcinomas; 14 months post-DEN) were found to be positive for CAF markers (VIM, FAP, α -SMA; **Extended Data Fig 7q**). This newly presented data confirms that, although CAFs do not appear to be central players at nascent stages, cancer associated phenotypes become apparent as tumours progress.

Interestingly, we found that early tumour fibroblasts show a transcriptional profile that closely recapitulate to that of wound-associated fibroblasts⁴ (see **Rebuttal Figure 6** below). Hence, we propose that at early tumour stages, before the CAF phenotype becomes fully established, fibroblasts exhibit a mixed transcriptional profile dominated by a wound-like signature, most likely as a result of the mutant epithelium, but seemingly transitioning towards an early myCAF state. This is further supported by an active YAP (aYAP) immunostaining in nascent tumour niche fibroblasts, denoting a pre-CAF state in line with previous reports²¹ (**Extended Data Fig 4e**).

Rebuttal Figure 6: Expression level of typical wound-healing and cancer-associated fibroblast markers in Cluster 19 fibroblast population across conditions. Violin plots illustrating log-transformed normalised expression levels of typical myCAF subtype markers and wound-healing associated fibroblast markers in control, DEN (internal control) and tumour conditions within Cluster 19 (C19) fibroblast population.

6. In the last paragraph of the results the authors claim that Sox9 exerts its effect through EGFR signalling. But they do not make the experimental link between Sox9, EGFR/AREG and FN. Is this signalling altered in the Sox9 KO mice? This should be tested in mice, and, importantly, verified in human samples. Specifically, the authors should test whether SOX9 expression is found in early tumor developmental stages or has any survival advantage.

Question 6. Reviewer #2:

We thank the reviewer for their feedback. We have generated new data that provides additional functional depth and validate our original claim. We demonstrate that EGF-SOX9-Fibronectin is central to epithelial-stromal communication and tumour survival at the most incipient stages of tumorigenesis. Please see below a detailed account of our new findings, including human validation:

a. Epithelial EGFR Signalling Recruits Fibroblasts and Remodels the Stroma. Our Cellchat analysis predicts that epithelial-fibroblast communication occurs via EGFR in tumours, and we have since validated this extensively.

We originally proposed that AREG (an EGFR ligand) acts as a fibroblast chemoattractant that promotes ECM deposition and niche formation. New data from regenerative 3D epithelial organ cultures³⁰ now support this, showing that signals from actively expanding SOX9+ keratinocytes favour their compartmentalisation of fibroblasts and ECM remodelling in an EGFR-dependent manner (**Fig 5f-h, Extended Data Fig 10d,e**).

Specifically, we observe that when fibroblasts are co-cultured with 3D epithelia, they self-organise into distinct populations: **PDGFR α ^{low}** (Vimentin-high, Fibronectin-producing) fibroblasts near epithelial cells, and **PDGFR α ^{high}** (Fibronectin-low) fibroblasts in areas distant from the tissue (**Fig 5f,g; Extended Data Fig 10e**). This *in vitro* assay remarkably recapitulated the *in vivo* stromal compartment architecture simply by exposing fibroblasts to epithelial cells. The interaction between actively proliferating epithelial cells and fibroblasts also promoted Fibronectin deposition in the PDGFR α ^{low} compartment (**Fig 5g**). Notably, both fibroblast compartmentalisation and remodelling were prevented by the EGFR inhibitor Gefitinib (**Fig 5f-h; Extended Data Fig 10e**).

Overall, this *ex vivo* experiments confirmed that epithelial cells stimulate fibroblast and ECM remodelling via EGFR signalling.

b. Inhibition of EGFR Reduces Niche+ Tumour Burden. The revised manuscript includes new *in vivo* data demonstrating that treatment with Gefitinib (an EGFR inhibitor) leads to a reduction in early tumour burden 10 days post-DEN treatment, with a significantly lower number of Niche+ tumours being detected in the tissue (**Fig 5i,j**).

c. EGFR Modulates SOX9 Expression. Gefitinib treatment was also found to reduce SOX9 expression in epithelial cells (**Extended data Fig 10d**). This suggests that EGFR signalling may be upstream of SOX9 or perpetuates its expression via a feedback loop.

d. Sox9 deletion Impacts Niche+ Tumours. We previously showed that Sox9 deletion in *Krt14^{CreER}; Sox9^{flox/flox}* DEN-treated mice caused a significant reduction in tumour burden (**Fig 4r; Extended Data Fig 9o,p**). Our new analysis now shows that Sox9 deletion also results in a reduction of nascent Niche+ tumour size one month post-DEN treatment, specifically reaching a similar size to that of Niche- tumours. Sox9 deletion did not cause significant changes in Niche- tumours. This data further reinforces our claim on the role of SOX9 in niche remodelling.

e. **Human validation.** We are pleased that, despite the limited availability of early-stage squamous oesophageal samples, given that this cancer is typically detected at more advanced stages, we were able to partner with collaborators at **Addenbrookes Hospital** (Cambridge, UK), **Guy's and St Thomas'** (London, UK), and **TU Dresden** (Germany) to validate our observations in human material. Below we provide a summary of our main findings:

i. **Human marker expression recapitulates DEN tumour profile:** We have obtained new data from chemo-naïve, early-stage **T1a-b squamous oesophageal cancer** samples. Our immunolabelling showed that these early-stage tumours, unlike adjacent tissue, recapitulate the marker expression found in our DEN mouse model. T1a-b tumours presented a homogeneous expression of KRT6A, KRT17 (a pan-tumour marker in mice; **Fig 6b and c**, respectively) and heterogeneous expression of SOX9 (which marks a subpopulation of tumour cells associated with stromal remodelling in mice). Accordingly, SOX9 colocalises with upregulated AREG expression (**Fig 6b**), found to stimulate fibroblast remodelling (**Fig 5d**).

ii. **SOX9 expression is linked to stromal remodelling:** New data from chemo-naïve, early-stage **T1a-b squamous oesophageal cancer** samples show that the heterogeneous SOX9 expression profile correlated with fibroblast density; SOX9^{high} areas were associated with a higher density of fibroblasts (PDGFR α ; **Fig 6c**).

iii. **SOX9 expression is linked to ECM remodelling:** Immunolabelling of chemo-naïve **high-grade dysplasia (HGD; carcinoma in situ – Tis)** wholemounts reinforced the link between SOX9 expression and stromal remodelling. We observed marked fibronectin deposition in the vicinity of SOX9^{high} tumour cells (**Fig 6d**). This observation was specific to the tumour site, as distal tissue showed no SOX9 or fibronectin immunolabelling.

iv. **Residual dysplastic tissue post-chemotherapy:** Similar features were also found in residual dysplastic tissue post-chemotherapy (Fig. 6e,f), which showed marked ECM remodelling (fibronectin deposition) in the proximity of AREG+/SOX9+ tumour cells.

These observations support the presence of a heterogeneous tumour populations marker by **AREG** and **SOX9** upregulation in early-stages of squamous tumours of the human oesophagus. The data further reveals the association between this population and changes in mesenchymal and ECM remodelling, demonstrating the potential clinical relevance of our study.

7. Figure 4- the authors use Dvorak's theory to describe the pre cancer niche as the wound that never heals. However, they do not relate to the immune compartment that plays an integral role in wound healing processes. Supp figure 2 shows the presence of CD45 cells in nascent tumor niche 10 days post-DEN. Are there differences in immune cell composition between niche+ and niche- tumors?

Question 7. Reviewer #2:

We apologise for the lack of background on the immunity aspect. The role of the immune compartment in early tumour elimination in the DEN model was already explored in our previous collaborative work (Colom et al., Nature 2021)¹. In that previous study we investigated the mechanisms involved in the disappearance of nascent tumours in the DEN model. There, our

results demonstrated that incipient tumours were rapidly lost with no indication of an anti-tumour immune response, including no differences in early tumour persistence in immune-deficient mice (Extended data Figure 6 in Colom et al., Nature 2021)¹. Instead, early tumour clearance was shown to be largely driven by the competitive advantage of neighbouring mutant clones.

However, we do acknowledge the importance of this suggestion, particularly in light of the identification of two distinct early tumour subtypes (Niche+ and Niche-). Hence, we have revisited this point and now provide new data detailing the immune compartment in nascent Niche+ and Niche- tumours within our DEN model. Our key findings are as follows:

- a. **Immune cells in Early Tumours:** As shown in **Fig 3b, c** and **Extended Data Fig 4g**, we found no significant differences in the immune cell infiltration between nascent Niche+ and Niche- tumours at day 10 post-DEN treatment. This finding reinforces our earlier observations (Colom et al., Nature 2021)¹.
- d. **Studies in Immune-deficient Mice:** Additional experiments in immune-deficient mice show no difference in the number of Niche+ and Niche- tumours that develop compared to wild-type mice (**Extended Data Fig 4h-l**). This indicates that the immune response may not be a central player determining early tumour heterogeneity and nascent niche formation. We, however, noted a marginal decrease in tumour size at 10d post-DEN; an effect that lost relevance at later time points (2m post-DEN; **Extended Data Fig. 4l**).
- b. **Immune Cell Signature in Surviving Tumours:** **Fig 4b** and **Extended Data Fig 8** now include a characterisation of immune cell subtypes eight months after DEN treatment as per our scRNA-seq data. We have identified specific immune cell populations that are enriched in surviving tumours compared to the surrounding tissue, consistent with previous reports². Shortly, T cell status in surviving tumours showed an increase in T effector and resident memory cells compared to surrounding control tissue. The increase in T regulatory cells hints an immunosuppressive environment. Monocyte activation resulting in dendritic cell infiltration as well as neutrophil recruitment indicates a transition to Type III immunity (**Extended Data Fig 8e,f,g** and **Supplementary Table 10**).

Minor comments:

1. Page 5: “However, at these early stages, the presence of this structure was largely heterogeneous, with the majority of epithelial lesions (~70%, 199 lesions out of 296 per esophagus, on average) showing no apparent stromal reorganization (Fig. 1c)” – where is this data shown? The authors should present this quantification.

Minor point 1. Reviewer #2

We apologise for not pointing to the relevant figure. The 70% statement comes from **Extended Data Fig 2a** (10 days post-DEN; original manuscript). The text has been revised to include this, as well as a simplified figure highlighting this point (**Fig 1d**). Also, please refer to the relevant numerical data in **Rebuttal Table 1** (copied below), as well as the associated manuscript source data file.

Response to reviewers' Nature ms 2024-07-13618A-Z

	10d post DEN			2m post DEN			8m post DEN			1y post DEN		
	Total	Niche -	Niche+	Total	Niche -	Niche+	Total	Niche -	Niche+	Total	Niche -	Niche+
Animal1	310	202	108	116	43	72	63	17	47	103	57	47
Animal2	245	168	77	133	40	94	103	15	88	53	5	48
Animal3	334	226	107	75	17	58	72	11	61	81	45	36
Average	296	199	97	108	33	75	79	14	65	79	36	44

Rebuttal Table 1. Number of oesophageal tumours detected at different time-points after DEN administration. Total tumours are the sum of Niche- and Niche+ tumours.

2. Supp figure 2c-d – how did you calculate the proportion of endothelial cells and immune cells separately? They are signalling with the same fluorophore.

Minor point 2. Reviewer #2

We thank the reviewer for pointing this out. Immune and endothelial cells were shown in a single channel to emphasise the contribution of fibroblasts to the tumour niche (given that they are the main component of the niche). New figures now show CD45 (immune) and CD31 (endothelial) in different channels for further clarity (**Fig 3c; Extended Data Fig 8a**).

3. Supp figure 2e- what is the sign of invasion? What does ITAG6 mark?

Minor point 3. Reviewer #2

We thank the reviewer for highlighting the lack of background. Integrin $\alpha 6$ (ITGA6) is expressed on the basal surface of epithelial cells, where it mediates adhesion to the basement membrane. In our staining, ITGA6 maintained a continuous basal localisation in Niche+ tumours, consistent with preservation of the epithelial–basement membrane interface. As we did not observe any disruption of this pattern, we concluded that there were no signs of epithelia cell invasion through basement membrane. The text has been amended to clarify this point accordingly.

4. Figure 3h – the authors don't show the expression of SOX9 in healthy control tissue but state in the text is it not expressed as opposed to nascent tumors.

Minor point 4. Reviewer #2

Apologies for not referring to the relevant control images in the text. A representative control area labelled for SOX9 can be found in **Extended Data Fig 9j** (showing both morphologically normal tissue in DEN-treated animals, as well as untreated control tissue). This figure includes controls for other markers of the relevant **Tumour 12** population as per our scRNA-seq reanalysis.

5. "For this we focused on cluster 23 (C23) fibroblasts, a Pdgfralow cluster enriched in tumour cells (Fig. 4e)." I guess the authors mean stromal cells from tumors. The wording is confusing.

Minor point 5. Reviewer #2

We are grateful for the reviewer's attention to detail. This was indeed an error and has been corrected in the new version of the manuscript.

*6. Ext Fig 1b – the significant p values for the different time points – what are they pointing to?
There is only one group – what are they comparing to?*

Minor point 6. Reviewer #2

Apologies for this, the p-values shown in **Extended Data Fig 1b** represent statistical significance determined by a one-way Welch's ANOVA with a post-hoc Dunnett's T3 test for multiple comparisons between consecutive time-points.

Referee #3 (Remarks to the Author):

Study presented by Skrupskelyte et al. aims to identify environmental changes underlying the transition of healthy epithelium towards cancer. The authors used a chemical carcinogen (DEN) to introduce genetic mutations in the squamous esophagus of mice and their observed two types of precancerous lesions. Some were surrounded by nest-like (niche+) structures composed of stromal fibroblasts. Niche+ predominantly developed into cancerous lesions. Next, using heterotypic cultures, they show that cell growth of healthy epithelia is affected by the underlying niche+ stroma. These cells can be more frequently transplanted into nude mice when exposed to niche+ stroma. Next, they performed scRNA-seq on the esophagus and squamous forestomach and investigated transcriptional patterns of the epithelium and stromal compartments. They observed a shift towards proliferating cells in the epithelium and the emergence of Sox9 cells as the disease progressed. In the stromal compartment, they identified Pdgfra low cells as potential drivers of niche+ phenotype. Subsequent cell-to-cell communication analysis identified fibronectin (FN) and EGF signalling pathways as drivers of stoma-to-epithelium interaction in the niche+ environment. They finally demonstrate that Sox9 high keratinocytes can communicate with fibroblasts using EGF induced signalling molecules and that fibroblasts signal to keratinocytes using fibronectin pathways. In general, this study aims to describe a novel mechanism of oesophageal cancer development with a strong focus on the role of microenvironment modulation during the initiation of this cancer. The elucidation of the nongenetic factors driving cancer initiation is of out-most importance and this study offers an important view on this process. The authors' use of chemical carcinogen that recapitulate normal oncogenesis is commendable. The authors further provide ample supplementary information. However, the study has technical and conceptual limitations:

Reviewer #3 Authors' response: We are grateful for the reviewer's insightful feedback and suggestions, which will certainly enhance the presentation of our manuscript. We are also pleased that the reviewer considers that our study "*offers an important view*" on cancer initiation, and that the use of the DEN model in our study is "*commendable*".

1. First of all, in their single cells analysis of fibroblast, the authors avoid detailed analysis of individual clusters. It is very apparent from their analysis that the loss of cluster 8 (fig. 4a, S4e) is the most dramatic change between DEN and Ctrl samples. These cells are not well characterised by the authors and additional information on the identify of these cells (e.g. table of cluster markers) is needed.

Question 1. Reviewer #3

We apologise for missing to elaborate on individual fibroblast clusters. For reference, the revised version of the manuscript, which includes a complete re-analysis of scRNA-seq data, incorporates a detailed marker gene list for all clusters (**Supplementary Table 5**).

The reasoning behind the cluster selection for our downstream analysis resides in empiric observations while characterising the early and persistent tumour niche. As shown by lineage tracing, histology, and the scRNA-seq annotation, our data noted the differential contribution of the *Pdgfra*^{low} and *Pdgfra*^{high} fibroblast populations to the early tumour niche, with *Pdgfra*^{low} being the main fibroblast population constituting the tumour niche (**Fig 3a,d-f; Extended Data Fig 5**). In line with our histological observations, we found that the new *Pdgfra*^{low} cluster 19 is particularly

Response to reviewers' Nature ms 2024-07-13618A-Z

enriched in early tumour samples compared to controls (**Extended Data Fig 7c**). Hence, we considered it justified to focus on this fibroblast cluster for our downstream communication analyses.

For further transparency, we now provide a characterisation of the *Pdgfra*^{low} and *Pdgfra*^{high} fibroblast populations in control and tumour conditions (**Fig 4c-e; Extended Data Fig 7a,b**). Please refer to Supplementary material **Supplementary Tables 5,6**.

2. Secondly, cluster 15 is also largely ignored in the course of authors analysis. These fibroblast in comparison to other fibroblast clusters, see to expand after DEN treatment and in the tumour samples. The presence of acta2 and other fibroblast markers (Fig. S4g) indicates that these fibroblasts might be contractile in nature.

Question 2. Reviewer #3

We appreciate the reviewer's comment. Please note that, following the reviewers' suggestions, we have now revised the analysis of our scRNA-seq data. We have revisited the figures to improve the annotation and cluster selection (See **Fig 4** and **Extended Data Fig 6-9**). We can confirm that our main observations and cell populations remain equivalent to those originally proposed (**Rebuttal Figure 2**). This provides assurance that our results are robust. In addition, we also provide further functional experiments to validate our main conclusions (**Fig 3d,f and Fig 5d-h; Extended Data Fig 10c-f**). We believe that the proposed improvements to the data presentation will enhance the clarity of our message.

We thank the reviewer for their specific question regarding the *Pdgfra*^{high} fibroblast cluster. The selection of fibroblast populations for downstream analysis was based on empirical data, focusing on cells enriched specifically within tumours rather than in surrounding tumour-free tissue. Our histological and fibroblast lineage tracing analyses established that fibroblasts expressing low levels of PDGFR α are the main contributors to the DEN tumour niche (**Fig 3a, d-f; Extended Data Fig 5, 7c**). We have now included new *Pdgfra*-Cre driven lineage tracing data, which further reinforces the choice of *Pdgfra*^{low} fibroblasts as the relevant population in early tumour-mesenchyme crosstalk (**Extended Data Fig 5j-o**). While the *Pdgfra*^{high} population (Original Cluster 15, now Cluster 3) was included in our analysis, it was not found to be a major component of the tumour niche and, therefore, was not the focus of our study. A differential gene expression analysis between *Pdgfra*^{low} and *Pdgfra*^{high} cells is provided for reference (**Fig 4d**). Please also refer to **Supplementary Table 6**, comparing tumour *versus* control or *versus* DEN in each cluster.

3. Although the authors provide a limited set of markers per cell cluster, it appears that fibroblast studies here resemble populations S1, S2 and S3 previously annotated by Davidson et al (<https://doi.org/10.1016/j.celrep.2020.107628>). The studies are different enough (here authors investigate cancer initiation rather than injectable model) however the authors should investigate in detail the shift of clusters 8 and 15, as they resemble studies of Davidson et al and other CAF studies (<https://doi.org/10.1038/s41568-019-0238-1>).

Question 3. Reviewer #3

We completely agree with the reviewer's point that the tumour-associated fibroblasts in our scRNA-seq data marginally resemble the myCAF signature (**Fig 4f; Extended Data Fig 7c-l**). We

observed a subtle increase in myCAF-associated genes like *S100a4*, *Vim*, *Mfap*, and *Col1a2* compared to our control conditions (**Extended Data Fig 7l**). However, comprehensive CAF marker characterisation at the gene and protein level across conditions revealed that the CAF phenotype does not appear to be fully established at the timepoints we originally examined (**Extended Data Fig 7k-o**). The genes indicated by the reviewer are indeed expressed at different levels in *Pdgfra*^{low} and *Pdgfra*^{high} fibroblast populations, but do not show significant changes across conditions at pre-neoplastic tumour stages (see **Rebuttal Fig 7**). Therefore, the presented data argues against a CAF signature during early tumour stages. The revised manuscript now includes references and notes the relevant literature suggested by the reviewer.

To demonstrate that the lack of the relevant CAF markers did not represent an artifact but faithfully represented the biological properties of the tumour at the stage investigated, we have now immunolabelled later-stage tumours (invasive carcinomas; 14 months post-DEN). These tumours were found to be positive for CAF markers (VIM, FAP, α-SMA) (**Extended Data Fig 7q**). Hence, taken together, the presented findings prove that while tumour niche fibroblasts may be primed to become myCAFs, as one may anticipate, such phenotype is not yet fully established at pre-neoplastic stage.

Interestingly, we discovered that early tumour fibroblasts have a transcriptional profile that closely resembles that of wound healing-associated fibroblasts⁴ (see **Rebuttal Fig 8**). Therefore, we propose that at these early stages, before the full CAF phenotype is established, fibroblasts display a mixed transcriptional profile that is dominated by a wound healing-like signature (likely a response to the mutant epithelium), while transitioning toward an early myCAF state. This is further supported by active YAP (aYAP) immunostaining, consistent with previous reports²¹ (**Extended Data Fig 4e**).

Response to reviewers' Nature ms 2024-07-13618A-Z

Rebuttal Figure 7: Expression level of typical cancer-associated fibroblast markers from Davidson et al. **a**, fibroblast cluster distribution in a cropped UMAP space. Clusters 2,4,9 were identified as $Pdgfra^{low}$ and Cluster 3 was identified as $Pdgfra^{high}$. **b**, Heatmap showing expression of representative marker genes from S1, S2 and S3 (Davidson et al) in fibroblast clusters from (**a**), separated by condition. **c**, Heatmap showing expression of canonical CAF markers in fibroblast clusters from (**a**), separated by condition. **a** and **b**, Log-transformed normalised expression levels were averaged by cluster for each gene and scaled across all cells belonging to each group. Scale bar denotes expression range (scale: 0 to 4).

Response to reviewers' Nature ms 2024-07-13618A-Z

Rebuttal Figure 8: Expression level of typical wound healing-associated fibroblast and cancer-associated fibroblast markers in Cluster 19 fibroblast population across conditions. Violin plots depict log-transformed normalised expression levels of typical myCAF subtype markers and genes associated with wound healing response⁴.

4. The choice of *Col1A2* is not obvious as a driver gene for the *Pdgfra*-low fibroblast is poorly justified. This gene seems to be expressed in all fibroblast clusters (S4g) and a potential flow/transcommitment of cells between compartments cannot be excluded. The authors, in addition to *pdgfra*-low, should also lineage-trace other fibroblast populations (especially clusters 8 and 15) to understand if these cells do not contribute to the niche+ phenotype. If only cells from cluster 23 were responsible for the phenotypes observed, lineage tracing using other markers should not result in the clonal clustering described by the authors.

Question 4. Reviewer #3

We fully appreciate the reviewer's point. In response to this request, we have now reinforced our observations on fibroblast niche contribution by performing a new set of lineage tracing experiments. We have lineage traced fibroblasts using the *Pdgfra-Cre^{ERT}* as a driver for reporter recombination, which more directly targets the *Pdgfra^{low}* and *Pdgfra^{high}* populations of interest (new data in **Extended Data Fig 5j-o**). Titration experiments showed a dose-dependent recombination efficiency, with only ~2 cells being labelled per 0.1 mm² at the dose of 0.5mg/20g body weight tamoxifen administration (**Extended Data Fig 5j-m**). We anticipated that the different level of *Pdgfra* expression would allow us to control the level of recombination across the two stromal compartments: PDGFR α ^{low} (lamina propria, i.e. upper stroma) and PDGFR α ^{high} (submucosa, i.e. lower stroma). Indeed, although not as initially anticipated, at the lower tamoxifen dose tested (0.5 mg/20g body weight), we observed a markedly higher recombination efficiency in the PDGFR α ^{low} population (upper stroma), with negligible recombination detected in the PDGFR α ^{high} compartment (lower stroma; **Extended Data Fig 5j-l**). This, together with the fact that PDGFR α ^{low} and PDGFR α ^{high} populations are compartmentalised in the tissue (lamina propria and submucosal stromal compartments, respectively) and delimited by the muscularis mucosae (a layer of smooth muscle cells between the lamina propria and the submucosa; **Extended Data Fig 4a-d**), renders *Pdgfra-Cre^{ERT}* lineage tracing a good model to track fibroblasts in the two compartments.

In the interest of completeness, we performed lineage tracing of fibroblasts in the early tumour niche using both tested doses (**Extended Data Fig 5n,o**). Six weeks post-DEN treatment, we observed that the early tumour niche was formed by *Pdgfra^{low}* derived fibroblast clones that expanded in the upper stroma where these cells reside under normal conditions. No clonal events were found to i) expand in the lower stroma (submucosa, PDGFR α ^{high} compartment) or ii) to span across stromal compartments. These observations support that PDGFR α ^{low} cells (residing in the upper stroma) are the main contributors to the formation of the early tumour niche.

We fully agree with the reviewer that performing lineage tracing on both fibroblast subpopulations, namely *Pdgfra^{high}* (new cluster 3) and *Pdgfra^{low}* (new clusters 2, 4 and 19), would have been ideal. However, our scRNA-seq analysis did not reveal any distinctive markers for which traceable mouse lines are readily available. Examples include *Has1/2* for *Pdgfra^{low}* and *Lama1* for *Pdgfra^{high}*.

We hope that the new lineage tracing experiments outlined above, together with our previous unbiased approach based on *Col1a2*, where both *Pdgfra^{low}* and *Pdgfra^{high}* populations were simultaneously traced showing expansion only in the upper stroma (PDGFR α ^{low} compartment), will convince the reviewer that the *Pdgfra^{low}* fibroblast population is actively contributing to the formation of the early tumour niche.

5. Similarly, the trajectory analysis of the epithelial compartment lacks detail. The authors often switch between projections (monocle components and umap) and use different markers on different embedding types. This makes the analysis difficult to follow. It would be helpful if authors provided projects of all cell types and tissue types onto monocle projects and also the projection of states (1-3) on the Umap projects. Also, vector information in the analysis is missing. RNA velocity could provide information about the direction of cell trajectory. It is not clear to me if the cells flow from states 3 to 2 or vice versa.

Question 5. Reviewer #3

We appreciate the expert advice of the reviewer and apologise for the confusing presentation. Following the reviewer's suggestion, we have re-analysed our scRNA-seq data, providing a focused and clear presentation of our results. We believe the improvements to the data will enhance the clarity of our message. See a brief description of the changes that relate to this point:

- a. **New trajectory analysis:** Following the reviewers' recommendations, we have now performed *Monocle3* pseudotime analysis which infers a trajectory in the UMAP space (**Fig 4 h,i and Extended Data Fig 9e**). The new pseudotime trajectory has enabled us to identify gene modules specific to epithelial tumour cells, revealing two major epithelial tumour sub-populations with distinctive transcriptional signatures (referred to as **Tumour 1** and **Tumour 12**; new and revised data **Fig 4 h-l** and **Extended Data Fig 9e-k**).
- b. **RNA Velocity:** We have now performed RNA Velocity analysis (**Extended Data Fig 9d**) to infer the directionality of trajectories as requested by the reviewer.

Please refer to the revised main text and new **Methods** section "**Cell transition trajectory analysis**". Please refer to the section in page 39, line 1-34.

6. The validation of Sox9 genotype (figure 3o-p) only focuses on tumour burden in this model. The authors did not use this metric in any other setting in the study and additional analysis of these mice is warranted.

Question 6. Reviewer #3

We thank the reviewer for this insightful comment regarding the conditional SOX9 deletion experiment. To clarify the impact of SOX9 on tumour niche heterogeneity, we have now included new data showing that SOX9 deletion in basal epithelial cells reduced the size of Niche+ tumours to values matching those of Niche- tumours (**Extended data Fig 9p**). These observations reinforce the claim that SOX9 expression in epithelial tumour cells promotes the Niche+ tumour phenotype. Please note that, as previously reported, SOX9 expression is necessary for early tumour formation¹⁹. We found that the latter also holds true in the DEN model.

7. In Figure 1e, niche- cluster of cells appears significantly smaller and there is some proliferation in its nuclei (the KRT6A signal seems to be overexposed). What proportion of nuclei in the niche+/niche- regions took up EdU? Figure S2g only has counts which might be consequences of cellular density rather than higher proliferation rate. As these are representative images, it would be helpful if the authors chose regions of similar size to avoid optical illusions.

Question 7. Reviewer #3

We thank the reviewer for their attention to detail. Apologies for the lack of clarity, the data presented in new **Extended Data Fig 2c** are normalised to total basal cell number (proliferative compartment). Thereby, variations in cell density and tumour size are accounted for. This has now been clarified in the revised version of the manuscript. For further transparency, data on basal cell density can now be found in **Extended Data Fig 2dc**.

Additionally, we would like to apologise for the overexposure of the images. Please note that a recognised feature of squamous tissues is that cornified layers (fully keratinised, non-proliferative layers) on the surface of the tissue typically show misleadingly high immunolabelling signal compared to other layers. Hence, when images are projected a level of saturation is unavoidable to make sure that all labelled cells are visualised. Similarly, we recognise that we did not select the most representative images for this figure. Please find revised representative images in **Fig 1e**. These images have been generated omitting the uppermost keratinised non-proliferative layer to avoid overexposure. This has been noted in the figure legend.

“Images were generated omitting upper most suprabasal layer to avoid overexposure upon projection.”

8. SOX9 staining on Figure 3k and 3m has a lot of background and it is very difficult to distinguish real cells/nuclei from the background

Question 8. Reviewer #3

We appreciate the reviewers' concern. Unfortunately, immunostaining of tissue wholemounts (epithelial and stromal composites of around 200 μm) presents challenges for certain proteins and can result in a grainy background with some antibodies. This is particularly the case for SOX9 antibodies. To improve visualisation, we have now replaced original images with alternative examples that exhibit reduced background (**Fig 5a,c**).

9. Until figure 4, authors imply that all fibroblasts are pdgfra+ however, starting in figure 4, they show fibroblasts in submucosa are actually pdgfra-high and the cells in lamina propria are pdgfra-low. This change seems to be very weak in control samples in Figure 4c but much stronger in the tumour. It would be helpful if the authors provided larger fields of view on the earlier figures to understand that pdgfra expression is not constant in all fibroblasts. Also, the figure 4c, there seems to be a continuum of fibroblasts in the normal samples, but in the tumour samples, there is a clear boundary between compartments. The authors could also provide larger fields of view to ensure that the reader can clearly see these transitions.

Question 9. Reviewer #3

We are thankful to the reviewer for raising this point and apologise for the lack of clarity regarding PDGFR α expression.

We can confirm that all fibroblasts express PDGFR α and, hence, it represents a pan-fibroblast marker in the upper GI tract, in line with other tissues³⁴. However, as shown in tissue wholemounts, PDGFR α expression levels are compartmentalised across two distinct stromal layers. PDGFR α^{low} fibroblasts reside in the lamina propria (upper stroma), while PDGFR α^{high} fibroblasts are found in the submucosa (lower stroma), with both compartments being physically separated by the muscularis mucosae (a distinctive layer of smooth muscle cells; **Extended**

Data Fig 4a-d). Hence, we were not entirely surprised when our scRNA-seq analysis revealed clear differences in the expression level of *Pdgfra* across fibroblast clusters (**Fig 4c**).

In retrospect, we realise that the stromal images presented when characterising the early tumour niche, particularly in the first part of the paper, may have been unintentionally misleading. These included high magnification images of the niche directly under early tumours, i.e. displaying only the upper stroma (lamina propria i.e. PDGFR α ^{low} compartment). Our intention was to emphasise the early tumour niche heterogeneity. Hence, these images were acquired with confocal exposure settings optimised to the expression levels of the PDGFR α ^{low} compartment. This enabled us to properly show and illustrate changes in stromal architecture, as well as niche remodelling. However, we appreciate that this did not reflect the differences in PDGFR α expression across stromal layers, inadvertently creating an apparent contradiction.

To address this point, we have taken the following steps:

- a. We have included a clearer description of the stromal compartments and revisited relevant images across the manuscript to show the presence of these two distinct populations.
- b. We have now provided more representative control images in new **Fig 3d-f**, including larger field of view images that more clearly illustrate the heterogeneous nature of the stromal compartment (**Fig 3e,f and Extended Data Fig 5e**).
- c. To emphasise the niche architecture, we have kept a number of images where the PDGFR α settings were optimised for the upper stroma. This has now been explicitly noted in the figure legends and methods (**Fig 1c; Fig 3c; Extended Data Fig 7j**);

Finally, we agree that the differences in PDGFR α compartmentalisation are typically more apparent in tumours due to the pronounced thickening of the lamina propria compared to controls. This has now been noted in the revised manuscript. We thank the reviewer again for prompting us to clarify this crucial point.

10. Multiple figures lack correct annotation of markers displayed. E.g. figure 1b: if the signal from multiple channels is separated then it should be done for every channel (there should be only KRT6A channel in addition to PDGFRA only channel). The PDGFRA only channel is missing annotation. The "Niche+" and "8m" labels do not indicate that these images are the same projection and all metadata is shared. This low-quality image annotation is pervasive in the figures (figure 4c tumour samples is shown twice, one with pdgfra shown in grayscale and once with this signal missing. This is not obvious from the legend or figure annotation. Similar pattern can be found on 1b, 1c, 2g, 3k, 3m, 3h, 5g, 5h.

Question 10. Reviewer #3

We thank the reviewer for pointing this out. Our original intent was to optimise within the space constraints by presenting the most informative channel combinations (merged or selected split-channel images), rather than displaying every channel separately. In our experience, showing all channels together often obscures key features, while showing single channels in isolation may fail to convey the biological context. This was particularly relevant for visualising epithelial–stromal cross-talk in heterogeneous tumours or clonal behaviour, where the presence or absence of the niche needed to be interpreted alongside a specific marker/reporter.

Response to reviewers' Nature ms 2024-07-13618A-Z

Regarding metadata, to avoid overcrowding, when panels were part of a composite, we presented common annotations (time point, niche status, scale bar, and channel information) only once for the composite. Any metadata that differed between panels was annotated individually.

We regret that this presentation strategy led to confusion. To address the reviewer's concern, we have revised the figures throughout the manuscript to ensure that all channels and annotations are consistently and clearly represented, while maintaining readability.

References

- 1 Colom, B. *et al.* Mutant clones in normal epithelium outcompete and eliminate emerging tumours. *Nature* **598**, 510-514 (2021). <https://doi.org/10.1038/s41586-021-03965-7>
- 2 Yao, J. *et al.* Single-cell transcriptomic analysis in a mouse model deciphers cell transition states in the multistep development of esophageal cancer. *Nat Commun* **11**, 3715 (2020). <https://doi.org/10.1038/s41467-020-17492-y>
- 3 Bhowmick, N. A., Neilson, E. G. & Moses, H. L. Stromal fibroblasts in cancer initiation and progression. *Nature* **432**, 332-337 (2004). <https://doi.org/10.1038/nature03096>
- 4 Guerrero-Juarez, C. F. *et al.* Single-cell analysis reveals fibroblast heterogeneity and myeloid-derived adipocyte progenitors in murine skin wounds. *Nat Commun* **10**, 650 (2019). <https://doi.org/10.1038/s41467-018-08247-x>
- 5 Rheinwald, J. G. & Green, H. Serial cultivation of strains of human epidermal keratinocytes: the formation of keratinizing colonies from single cells. *Cell* **6**, 331-343 (1975). [https://doi.org/10.1016/s0092-8674\(75\)80001-8](https://doi.org/10.1016/s0092-8674(75)80001-8)
- 6 Erez, N., Truitt, M., Olson, P., Arron, S. T. & Hanahan, D. Cancer-Associated Fibroblasts Are Activated in Incipient Neoplasia to Orchestrate Tumor-Promoting Inflammation in an NF-kappaB-Dependent Manner. *Cancer Cell* **17**, 135-147 (2010). <https://doi.org/10.1016/j.ccr.2009.12.041>
- 7 Saadi, A. *et al.* Stromal genes discriminate preinvasive from invasive disease, predict outcome, and highlight inflammatory pathways in digestive cancers. *Proc Natl Acad Sci U S A* **107**, 2177-2182 (2010). <https://doi.org/10.1073/pnas.0909797107>
- 8 Dvorak, H. F. Tumors: wounds that do not heal. Similarities between tumor stroma generation and wound healing. *N Engl J Med* **315**, 1650-1659 (1986). <https://doi.org/10.1056/NEJM198612253152606>
- 9 Leoni, G., Neumann, P. A., Sumagin, R., Denning, T. L. & Nusrat, A. Wound repair: role of immune-epithelial interactions. *Mucosal Immunol* **8**, 959-968 (2015). <https://doi.org/10.1038/mi.2015.63>
- 10 Marikovsky, M. *et al.* Appearance of heparin-binding EGF-like growth factor in wound fluid as a response to injury. *Proc Natl Acad Sci U S A* **90**, 3889-3893 (1993). <https://doi.org/10.1073/pnas.90.9.3889>
- 11 Repertinger, S. K. *et al.* EGFR enhances early healing after cutaneous incisional wounding. *J Invest Dermatol* **123**, 982-989 (2004). <https://doi.org/10.1111/j.0022-202X.2004.23478.x>
- 12 Aggarwal, S. *et al.* SOX9 switch links regeneration to fibrosis at the single-cell level in mammalian kidneys. *Science* **383**, eadd6371 (2024). <https://doi.org/10.1126/science.add6371>

Response to reviewers' Nature ms 2024-07-13618A-Z

- 13 Grinnell, F., Billingham, R. E. & Burgess, L. Distribution of fibronectin during wound healing in vivo. *J Invest Dermatol* **76**, 181-189 (1981). <https://doi.org/10.1111/1523-1747.ep12525694>
- 14 Yamamoto, T. *et al.* Animal model of sclerotic skin. I: Local injections of bleomycin induce sclerotic skin mimicking scleroderma. *J Invest Dermatol* **112**, 456-462 (1999). <https://doi.org/10.1046/j.1523-1747.1999.00528.x>
- 15 Franzen, O., Gan, L. M. & Bjorkegren, J. L. M. PanglaoDB: a web server for exploration of mouse and human single-cell RNA sequencing data. *Database (Oxford)* **2019** (2019). <https://doi.org/10.1093/database/baz046>
- 16 Grommisch, D. *et al.* Regionalized cell and gene signatures govern esophageal epithelial homeostasis. *Dev Cell* **60**, 320-336 e329 (2025). <https://doi.org/10.1016/j.devcel.2024.09.025>
- 17 Alcolea, M. P. *et al.* Differentiation imbalance in single oesophageal progenitor cells causes clonal immortalization and field change. *Nat Cell Biol* **16**, 615-622 (2014). <https://doi.org/10.1038/ncb2963>
- 18 Frede, J., Greulich, P., Nagy, T., Simons, B. D. & Jones, P. H. A single dividing cell population with imbalanced fate drives oesophageal tumour growth. *Nat Cell Biol* **18**, 967-978 (2016). <https://doi.org/10.1038/ncb3400>
- 19 Larsimont, J. C. *et al.* Sox9 Controls Self-Renewal of Oncogene Targeted Cells and Links Tumor Initiation and Invasion. *Cell Stem Cell* **17**, 60-73 (2015). <https://doi.org/10.1016/j.stem.2015.05.008>
- 20 Chen, Y. *et al.* Epithelial cells activate fibroblasts to promote esophageal cancer development. *Cancer Cell* **41**, 903-918 e908 (2023). <https://doi.org/10.1016/j.ccell.2023.03.001>
- 21 Calvo, F. *et al.* Mechanotransduction and YAP-dependent matrix remodelling is required for the generation and maintenance of cancer-associated fibroblasts. *Nat Cell Biol* **15**, 637-646 (2013). <https://doi.org/10.1038/ncb2756>
- 22 Myo Min, K. K. *et al.* Overcoming the Fibrotic Fortress in Pancreatic Ductal Adenocarcinoma: Challenges and Opportunities. *Cancers (Basel)* **15** (2023). <https://doi.org/10.3390/cancers15082354>
- 23 Sahai, E. *et al.* A framework for advancing our understanding of cancer-associated fibroblasts. *Nat Rev Cancer* **20**, 174-186 (2020). <https://doi.org/10.1038/s41568-019-0238-1>
- 24 Fotsitzoudis, C. *et al.* Cancer-Associated Fibroblasts: The Origin, Biological Characteristics and Role in Cancer-A Glance on Colorectal Cancer. *Cancers (Basel)* **14** (2022). <https://doi.org/10.3390/cancers14184394>
- 25 Huycke, T. R. *et al.* Patterning and folding of intestinal villi by active mesenchymal dewetting. *Cell* **187**, 3072-3089 e3020 (2024). <https://doi.org/10.1016/j.cell.2024.04.039>
- 26 Brugger, M. D., Valenta, T., Fazilaty, H., Hausmann, G. & Basler, K. Distinct populations of crypt-associated fibroblasts act as signaling hubs to control colon homeostasis. *PLoS Biol* **18**, e3001032 (2020). <https://doi.org/10.1371/journal.pbio.3001032>
- 27 Kumar, N. *et al.* Decoding spatiotemporal transcriptional dynamics and epithelial fibroblast crosstalk during gastroesophageal junction development through single cell analysis. *Nat Commun* **15**, 3064 (2024). <https://doi.org/10.1038/s41467-024-47173-z>
- 28 Manieri, E. *et al.* Role of PDGFRA(+) cells and a CD55(+) PDGFRA(Lo) fraction in the gastric mesenchymal niche. *Nat Commun* **14**, 7978 (2023). <https://doi.org/10.1038/s41467-023-43619-y>
- 29 Gallini, S. *et al.* Injury prevents Ras mutant cell expansion in mosaic skin. *Nature* **619**, 167-175 (2023). <https://doi.org/10.1038/s41586-023-06198-y>

Response to reviewers' Nature ms 2024-07-13618A-Z

- 30 Herms, A. *et al.* Self-sustaining long-term 3D epithelioid cultures reveal drivers of clonal expansion in esophageal epithelium. *Nat Genet* (2024). <https://doi.org/10.1038/s41588-024-01875-8>
- 31 Sinn, E. *et al.* Coexpression of MMTV/*v-Ha-ras* and MMTV/*c-myc* genes in transgenic mice: synergistic action of oncogenes in vivo. *Cell* **49**, 465-475 (1987). [https://doi.org/10.1016/0092-8674\(87\)90449-1](https://doi.org/10.1016/0092-8674(87)90449-1)
- 32 Hu, B. *et al.* Multifocal epithelial tumors and field cancerization from loss of mesenchymal CSL signaling. *Cell* **149**, 1207-1220 (2012). <https://doi.org/10.1016/j.cell.2012.03.048>
- 33 Shultz, L. D. *et al.* Human lymphoid and myeloid cell development in NOD/LtSz-scid IL2R gamma null mice engrafted with mobilized human hemopoietic stem cells. *J Immunol* **174**, 6477-6489 (2005). <https://doi.org/10.4049/jimmunol.174.10.6477>
- 34 Driskell, R. R. *et al.* Distinct fibroblast lineages determine dermal architecture in skin development and repair. *Nature* **504**, 277-281 (2013). <https://doi.org/10.1038/nature12783>

Response to reviewers' Nature ms 2024-07-13618A-Z

Authors' point-by-point response to reviewers (2024-07-13618A-Z)

Skrupskelyte et. al. entitled "Pre-cancerous Niche Remodelling Dictates Nascent Tumour Persistence"

Text from reviewers' comments presented in **blue italics**, our responses to reviewers' questions in **black**, and quoted references are highlighted in **orange italics**.

REVIEWER COMMENTS

Reviewer #1 (Remarks to the Author):

The additional experiments and revised analyses (confirmation of phenotype in immune deficient mice, added immune profiling, Monocle3 trajectories, CellTypist/label transfer annotations, expanded functional work with EGFR inhibition and SOX9 loss, and updated statistical analysis) substantially strengthen the manuscript. My concerns have been addressed to a degree that supports acceptance in principle.

I do want to re-emphasize the importance of fibroblast phenotypic analysis in a contemporary context. The claim that fibroblast phenotypes might be specific to esophageal cancer is not supported by current literature and again raises some concern as to the authors' expertise in fibroblast biology (Gao, Li, Cheng et al., Cancer Cell, 2024 and Liu, Cui, Han et al., Sci Adv, 2025). It appears that the authors are dealing with subtle variations of a Pi16-positive population (originally described in Buechler et al., Nature, 2021 as a stemlike, or steady-state phenotype), although it is difficult to be certain. The authors have shown that the targeted fibroblast phenotype does not represent an activated myCAF or iCAF state, and we appreciate this comparison.

Our final request is that the authors annotate fibroblast phenotypes in the context of one of these cross-tissue studies. This will be important to discern whether the fibroblast phenotype in question is a considerable departure from the well described quiescent (Pi16) phenotype.

Reviewer #1 - Author response: We are grateful to the reviewer for pointing us towards aspects of the data that needed further attention and for recognising that the revision "*substantially strengthen the manuscript*".

We apologise for the lack of clarity regarding the described fibroblast populations. We did not intend to imply that the presented fibroblast phenotypes were specific to our upper gastrointestinal (GI) tract model. In fact, we allude to other studies where heterogeneity in PDGFR α expression is associated with stromal compartmentalisation in other tissues. Please refer to quoted text "*In line with observations in other epithelial tissues^{1,2}, immunofluorescence analysis revealed two fibroblast populations that showed distinctive expression levels of the pan-fibroblast marker PDGFR α , tissue compartmentalisation and morphology (Extended Data Fig. 3a-d).*" page 7, lines 24-27.

To clarify this point, we have followed the reviewer's recommendation to draw parallels between fibroblasts defined in our study and those described in the cross-tissue study by Buechler et al., Nature, 2021³. We have now included the label transfer from this study to our fibroblast dataset and incorporated this in the revised manuscript (**Extended Data Fig. 6r,s; Supplementary Table 3 and Response Figure 1**).

Response to reviewers' Nature ms 2024-07-13618A-Z

As rightly indicated by the reviewer, and hinted by our existing *CellTypist* analysis (**Supplementary Table 3**), *Pdgfra*^{low} fibroblasts (Clusters 2, 4, 19) partially recapitulate the *Pi16*+ fibroblast signature, while *Pdgfra*^{high} fibroblasts (Cluster 3) resemble *Col15a1*+; both types identified as universal fibroblast subtypes across multiple tissues by *Buechler et al. 2021*³ (**Response Figure 1a,b**). Importantly, both *Pi16*+ fibroblasts (by *Buechler et al. 2021*³) and *Pdgfra*^{low} fibroblasts from our study (including Cluster 19, found to be enriched in early tumour fibroblast) share common traits, including low expression levels of *Pdgfra* and high expression levels of *Fn1*, *Pi16*, *Cd34* (**Response Figure 1c**). These data suggest that, as anticipated by the reviewer, tumour niche fibroblasts likely originate from universal *Pi16*+ fibroblasts. This has been noted in the revised version of the manuscript (main text): *“Label transfer analysis from Buechler et al. fibroblast atlas³ indicated that tumour niche fibroblasts likely derive from universal, rather than tissue-specific, fibroblast populations. Pdgfra^{low} fibroblasts, including tumour enriched cluster 19, partially recapitulated the transcriptional signature of the Pi16+ universal population, whereas Pdgfra^{high} fibroblasts aligned with the Col15a1+ universal fibroblast subset (Extended Data Fig. 6r,s).”* page 11, lines 12-17.

Response to reviewers' Nature ms 2024-07-13618A-Z

Response Figure 1. **a**, Upper gastro-intestinal (GI) tract fibroblast cluster distribution in a UMAP space (left) and violin plots (right) showing prediction score for label transfer of $Pi16+$ and $Col15a1+$ population markers across upper GI fibroblast clusters split by condition (right). **b**, UMAP projections denoting label transfer of cross-tissue fibroblast populations (Buechler et al., Nature, 2021) in upper GI fibroblast dataset from Skrupskelyte et al., 2024 (left). Unperturbed/perturbed datasets from Buechler et al., Nature, 2021 middle and right, respectively. **c**, UMAPs showing expression of representative genes (left to right order as per **b**).

Referee #2 (Remarks to the Author):

The authors have thoroughly addressed my comments.

One minor point - in Figure 3b, the authors quantified fibroblasts in $Niche^+$ vs. $Niche^-$ and state that fibroblasts are enriched in $Niche^+$. However, the figure appears to show enrichment in $Niche^-$ instead. Could it be typo?

Reviewer #2 Authors' response: We thank the reviewer for the positive remarks, highlighting the “*thoroughly addressed*” comments. We apologise for this mislabelling and are grateful to the reviewer for identifying this typo in the **Figure 3b**, which has now been corrected.

Referee #3 (Remarks to the Author):

The new manuscript submitted by the authors addresses the previous shortcomings very well. As urged by myself and other reviewers, the authors have introduced significant changes to their scRNA-seq analysis, added new mouse models, validated fibroblast-to-epithelial signaling discoveries, and provided a much-improved narrative that is no longer confounded by difficult-to-follow observations unrelated to the main findings. The revised manuscript elegantly demonstrates that fibroblast populations depositing matrix fibers support the survival of $Niche^+$ DEN-induced precancerous lesions in their model. Furthermore, they present data supporting the role of the EGF–SOX9–FN1 axis as an important driver of $Niche^+$ cell survival and selection.

As requested, the authors also demonstrate that, at least during the early stages of $Niche^+$ tumor development, the immune system does not play a major role—highlighting the importance of the fibroblast population. I am not surprised that the observed fibroblast phenotypes do not resemble mature CAFs, as such cells are typically present only in tumor tissue and not in precancerous lesions.

At this point, I have only a minor comment that should not detract from the importance of this study and its significant contribution to tumorigenesis research:

It is not entirely clear to me how the $Pdgfra$ -CreERT; $R26^{FlConfetti/WT}$ system works. A schematic of the Rosa26 locus would be helpful. I assume that the system includes a STOP codon upstream of Confetti that is removed upon TAM administration; hence, fibroblasts expressing lower levels of $Pdgfra$ would be expected to exhibit lower recombination efficiency and fewer fluorescent markers (as opposed to the observed phenotype). This discrepancy should be mentioned in the text.

Reviewer #3 Authors' response: We are extremely grateful for the reviewer's feedback and suggestions, encouraging comments, as well as considering our study of ‘*importance*’ and a ‘*significant contribution to tumorigenesis research*’. We are pleased that the reviewer found the

Response to reviewers' Nature ms 2024-07-13618A-Z

revised version of the manuscript addressed *'the previous shortcomings very well'* and provides a *'much-improved narrative'* of the manuscript.

As requested by the reviewer we now provide a schematic of the genetic construct in *Pdgfra-Cre^{ER}; R26^{FlConfetti/wt}* animals (**Response Figure 2** and **Extended Data Fig. 4j**). This mouse line was used to lineage trace PDGFR α fibroblasts. In this model tamoxifen stochastically activates Cre-mediated recombination, leading to the expression of 1 out of the 4 reporters of the Confetti construct in sporadic cells expressing *Pdgfra*. Reporter expression is inherited by labelled daughter cells, enabling lineage tracing studies.

Response Figure 2. Genetic construct of *Pdgfra-Cre^{ER}; R26^{FlConfetti/wt}* mouse and Tamoxifen (TAM) induced stochastic Cre-mediated recombination of the confetti cassette in sporadic cells expressing *Pdgfra*.

The aim of this experiment was to trace PDGFR α ^{low} and PDGFR α ^{high} fibroblasts independently by differential tamoxifen induction. However, as rightly pointed by the reviewer, we expected a positive correlation between *Pdgfra* gene expression and recombination efficiency *i.e.* to observe a lower recombination efficiency (less labelled cells) in the PDGFR α ^{low} fibroblast in the lamina propria when compared to PDGFR α ^{high} fibroblast in the submucosae. However, our data suggested that the recombination was higher in PDGFR α ^{low} cells, when compared to PDGFR α ^{high} cells (**Extended Data Fig. 4k,l**). This did not preclude the analysis, as the distinctive recombination efficiency was robust, but this was indeed a paradox. We reasoned that tissue accessibility and vascularisation may have played a role in determining the recombination efficiency. The lamina propria is a loose connective tissue rich in capillaries and highly permeable microvasculature. The submucosa, instead, has a denser fibrillar extracellular matrix and larger vessels, which are less permeable than the capillaries of the lamina propria. This may have determined the concentration of tamoxifen metabolite available for target cells and explain the recombination ambiguities between the two stromal layers. A study investigating *R26 Cre* recombination efficiency across tissues found that high doses of tamoxifen led a reduced recombination efficiency in the submucosae of the small intestine compared to the lamina propria⁴, which supports our hypothesis. This ambiguity is now mentioned in the text: *"Paradoxically, the PDGFR α ^{low} fibroblast population showed a markedly higher recombination efficiency, with negligible recombination detected in the lower PDGFR α ^{high} compartment (Extended Data Fig. 4j-m), likely due to different tamoxifen accessibility between stromal layers."* page 9, lines 10-13.

REFERENCES

- 1 Huycke, T. R. *et al.* Patterning and folding of intestinal villi by active mesenchymal dewetting. *Cell* **187**, 3072-3089 e3020 (2024). <https://doi.org/10.1016/j.cell.2024.04.039>

Response to reviewers' Nature ms 2024-07-13618A-Z

- 2 Manieri, E. *et al.* Role of PDGFRA(+) cells and a CD55(+) PDGFRA(Lo) fraction in the gastric mesenchymal niche. *Nat Commun* **14**, 7978 (2023). <https://doi.org/10.1038/s41467-023-43619-y>
- 3 Buechler, M. B. *et al.* Cross-tissue organization of the fibroblast lineage. *Nature* **593**, 575-579 (2021). <https://doi.org/10.1038/s41586-021-03549-5>
- 4 Kumar, R., Mao, Y., Patial, S. & Saini, Y. Induction of whole-body gene deletion via R26-regulated tamoxifen-inducible Cre recombinase activity. *Front Pharmacol* **13**, 1018798 (2022). <https://doi.org/10.3389/fphar.2022.1018798>

Response to editor's comments Nature ms 2024-07-13618A-Z

Authors' point-by-point response to editor's comments (2024-07-13618A-Z)

Skrupskelyte et. al. entitled "Pre-cancerous Niche Remodelling Dictates Nascent Tumour Persistence"

Text from editor's comments presented in ***blue italics***, our responses to editor's points in ***black***, and quoted references are highlighted in ***orange italics***.

Editor's points:

1. Please provide the manuscript in .docx format.

We have provided the manuscript in .docx format.

2. Please add references to the abstract (if applicable).

No references are required in the abstract.

3. The number of references should generally not exceed 60. Currently you have 99 references

The number of references for the main text has been reduced to 59. References corresponding to methods or supplementary methods have been moved to a separate reference list with the numbering continuing from the main text reference list.

4. Please create a separate reference list for any methods references, making sure that the numbering continues from the main text references.

References corresponding to methods or supplementary methods have been moved to a separate reference list with the numbering continuing from the main text reference list.

5. Please remove the main figures from the article file.

These have been removed and added as separate files.

6. Please note that the legends for the main figures should not exceed 300 words. If it is not possible to reduce the length accordingly, please ensure that the final legends are as close as possible to 300 words

This has been revised as requested.

7. Please reduce subheadings to 40 characters (with spaces) or less.

Response to editor's comments Nature ms 2024-07-13618A-Z

All subheadings have been reduced to 40 or less characters (including spaces), except the section 3 subheading *“Local fibroblasts form the nascent tumour niche”*, which count is 47 characters.

8. There are potential third party rights issues in the figures. Please check the sources of all illustrations and clarify whether permissions are needed to adapt or reproduce them. Please make sure to include the relevant details in third party rights table when you resubmit (more information below). If Biorender or a similar software has been used, please also ensure to provide relevant licenses. In particular please check figure/s: Figure 1a, 1g, 2a-c, 2e, 3d, 4a, 4j, 5a-b, 5d, 5f, 5h-j, 6a, 6e, 6g, Extended data Figure 1h, 2i-l, 3a-b, 3h, 4a, 4j, 4n, 5a, 7a, 8k, 9g-h, 10c-g.

BioRender licenses associated with the listed figure panels have been exported and attached as a zipped folder entitled “Nature ms 2024-07-13618A-Z (Skrupskelyte et al) BioRender Publication License”.

9. Please make sure to provide a third party rights table (more information below) when you resubmit. If Biorender or similar software has been used, please also ensure to provide relevant licenses.

All relevant figure references now contain a link to the publicly accessible part of the figure in BioRender platform. The third-party table is supplied.

10. Please provide a competing interest statement in the main text of the manuscript.

We have included Competing Interests Statement in the main text of the manuscript *“The authors declare no competing interests”*.

13. Please re-supply the Extended data figures individually in EPS, JPEG or TIF format.

Extended data figures have been uploaded individually as TIF.

14. Please ensure that the text size in all figures is at least 5 pt Arial.

This has been revised and applied as requested.

15. Please remove Supplementary information from the article file.

This has been removed from the main text and supplied separately.

Response to editor's comments Nature ms 2024-07-13618A-Z

16. All raw data need to be deposited and made publicly available before acceptance. Please complete deposition to all (raw) data and amend the Data Accessibility Statement to include accession numbers and routes to access all data in the manuscript.

Under the “Data Availability section” we have included *“The sequencing data generated in this study have been deposited in the Gene Expression Omnibus (GEO) repository under accession code GSE271962. The DNA sequencing dataset was deposited at the European nucleotide Archive (ENA) under dataset accession number ERP134942. Source data are provided with this paper.”* Please see relevant links below:

The hyperlink for GEO <https://www.ncbi.nlm.nih.gov/geo/query/acc.cgi?acc=GSE271962>

The hyperlink for ENA <https://www.ebi.ac.uk/ena/browser/view/ERP134942>

17. All custom codes need to be deposited and made publicly available before acceptance. Please amend the Data Accessibility Statement to include routes to access to all codes. In addition to github, custom code should be archived in a doi-minting repository, and both need to be publicly accessible before acceptance.

Under Code Availability section we have included *“No new algorithms were developed for this paper. The analysis code is available on the AlcoleaLab GitHub page.”*

Please see the associated hyperlink <https://doi.org/10.5281/zenodo.17802564>

18. The Methods must detail the maximal tumour measurements/volumes/other endpoints that were permitted by your IACUC, and state that these limits were not exceeded in any of the experiments.

Given that our work focuses on studying internal tumours particularly at early stages, i.e. not-visible, the adverse effects are not associated to tumour size/volume, but rather linked to observable clinical signs in treated mice. The following statement has been included in “Methods Chemical tumourigenesis model” section to address this point: *“Animals exposed to DEN were monitored as per adverse effects stated in our Home Office project licenses (PPL70/8866 and PP7037913) regulated procedures on protected animals. In summary, animals were weighed daily on weekdays for the first week, weekly for the next month and then monthly thereafter. Animals were also checked every day for any clinical signs or abnormal behaviour. Any concerning animals were weighed every other day or daily, if necessary, until the weight was stable again. If the weight loss approached 10%, animals were weighed daily until stable and*

Response to editor's comments Nature ms 2024-07-13618A-Z

received wet mash or palatable diet. Animals showing 15% weight loss measured for 2 consecutive days were sacrificed immediately.”

LENGTH: please reduce the number of main figures to 5 and reduce the length of the text to 4500 words.

We have reduced the main figures to 5 and the main text length to 4491 words.

	EDITORIAL REQUESTS:	AUTHOR RESPONSE:
1.	Data presentation: Please ensure that data presented in a plot, chart or other visual representation format shows data distribution clearly (e.g. dot plots, box-and-whisker plots). When using bar charts, please overlay the corresponding data points (as dot plots) whenever possible and always for $n \leq 10$. (Please see the following editorial for the rationale behind this request and an example https://www.nature.com/articles/s41551-017-0079).	
2.	Statistics: Wherever statistics have been derived (e.g. error bars, box plots, statistical significance) the legend needs to provide and define the n number (i.e. the sample size used to derive statistics) as a precise value (not a range), using the wording “n=X biologically independent samples/animals/cells/independent experiments/n= X cells examined over Y independent experiments” etc. as applicable.	
	Legends requiring revision:  Please note that this information is missing in the legend(s) of extended data figure(s) 4i. Please provide a precise value of ‘n’ in the legend(s) of figure(s) 1f; 2d; extended data figure(s) 3i, j, l; 9b, c, e; 10c. 	Missing information has been added to the relevant figure legends. Please, also refer to the information below for convenience. EDF 4i, “f,h and i data from n=18 areas in control from 3 animals, n=18 areas in DEN from 3 animals and n=16 tumours from 4 animals.” Fig. 1f, “Tumours were quantified in three mice per time point; at 10d n=128 tumours; at 2m n=84 tumours; at 8m n=53 tumours and at 1y n=49 tumours.” EDF 2l, (was Fig. 2d), “Dots represent areas assessed across different biological replicates n=9 control areas, from 3 mice in vivo, 23 control areas from 3 mice in vitro and n=10 tumours from 6 mice in vivo and 7 tumour like structures from 5 mice in vitro.” EDF 3i,j, “N=3 wt and 4 NSG mice at 10d and n=3 wt and 3 NSG at 2m post-DEN time point.” EDF 3l, “Diameter (μm) of Niche+ and Niche- tumours, n=126 (tumours in wt from 3 mice) and n=189 (tumours in NSG from 4 mice) at 10d post DEN and n= 57 (tumours in wt from 3 mice) and n=36 tumours in NSG from 3 mice) at 2m post-DEN.” EDF 9b,c, “3 mice per condition with n=20 DEN areas, 84 Niche- and 35 Niche+ tumours.” EDF 9e,f, “The number of control (n=27) or SOX9 expressing regions (n=42) assessed in tissues from 4 mice.”

		EDF 10c , “Each dot represents a technical replicate; data 4 biological replicates.”
3.	Statistics such as error bars, significance and p values cannot be derived from $n < 3$ and must be removed from all such cases. We strongly discourage deriving statistics from technical replicates, unless there is a clear scientific justification for why providing this information is important. Conflating technical and biological variability, e.g., by pooling technically replicates samples across independent experiments is strongly discouraged. (For examples of expected description of statistics in figure legends, please see the following https://www.nature.com/articles/s41467-019-11636-5 or https://www.nature.com/articles/s41467-019-11510-4). All error bars need to be defined in the legends (e.g. SD, SEM) together with a measure of centre (e.g. mean, median). For example, the legends should state something along the lines of “Data are presented as mean values +/- SEM” as appropriate. All box plots need to be defined in the legends in terms of minima, maxima, centre, bounds of box and whiskers and percentile. Legends requiring revision: Please note that the error bars/error bands need to be defined in the legend(s) of extended data figure(s) 2d; 4i; 5j.	Missing information has been added to the relevant figure legends. Please, also refer to the information below for convenience. EDF 2d, “Data expressed as mean +s.e.m.” EDF 4i, “Data expressed as mean and error bars denote 95% confidence intervals obtained from boot strapping.” EDF 6j (not 5j), “Data expressed as average \pms.d.”
4.	The figure legends must indicate the statistical test used. Where appropriate, please indicate in the figure legends whether the statistical tests were one-sided or two-sided and whether adjustments were made for multiple comparisons. For null hypothesis testing, please indicate the test statistic (e.g. F, t, r) with confidence intervals, effect sizes, degrees of freedom and P values noted. Please provide the test results (e.g. P values) as exact values whenever possible and with confidence intervals noted. Legends requiring revision:  Please indicate the statistical test used for data analysis and where appropriate, please specify whether it was one-sided or two-sided and whether adjustments were made for multiple comparisons, in the legend(s) of extended data figure(s) 4f. Please note that the information on whether the statistical test used was one-sided or two-sided, where appropriate, is missing in the legend(s) of extended data figure(s) 2c; 9b, c. 	Missing information has been added to the relevant figure legends. Please, also refer to the information below for convenience. EDF 4f, “Pairwise Wilcoxon signed-rank tests were performed, and P-values were adjusted using the Holm-Bonferroni correction.” EDF 2c, “One-way Welch’s ANOVA with multiple comparison was used to assess significance.” EDF 9b,c, “Statistical significance assessed by one-way Welch’s ANOVA with multiple comparisons.” Note, the exact p values were displayed in figures, rather than legends.

	3. Please note that the exact p value should be provided, when possible, in the legend(s) of figure(s) 1f; 2d; extended data figure(s) 3g, l; 6i; 8j; 9b, c, e, h.	Fig. 1f, $p=0.0697$; $p=6.72 \times 10^{-7}$; $p=2.41 \times 10^{-7}$; $p=0.0002$. EDF 2l, (was Fig. 2d), $p=0.8133$; $p=0.0013$; $p=0.0031$; $p=0.9986$. EDF 3g, $p=1 \times 10^0$; $p=0.0334$; $p=0.0058$. EDF 3l, $p=2.62 \times 10^{-9}$; $p=0.0197$; $p=0.7638$. EDF 6i, $p=0.0007$; $p=1.86 \times 10^{-5}$; $p=0.0005$. EDF 8j, “Non-significant genes (p value higher than $p=0.05$), green.” EDF 9b, $p=1.25 \times 10^{-4}$; $p=5.26 \times 10^{-8}$; $p=1.43 \times 10^{-3}$. EDF 9c, $p=1 \times 10^{-15}$; $p=3.91 \times 10^{-7}$; $p=1.22 \times 10^{-2}$. EDF 9e, $p=1.83 \times 10^{-11}$. EDF 9h, $p=2.16 \times 10^{-8}$.
5.	Reproducibility: Please state in the legends how many times each experiment was repeated independently with similar results. This is needed for all experiments, but is particularly important wherever results from representative experiments (such as micrographs) are shown. If space in the legends is limiting, this information can be included in a section titled “Statistics and Reproducibility” in the methods section. Legends requiring revision: Please note that this information is missing in the legend(s) of figure(s) 3e; 4e, g; extended data figure(s) 1c-g; 2g-l; 4k, m, o; 6d, k, q; 8m, o; 10g.	Statistics and reproducibility section was revised to include clear description of experiment reproducibility: “All experiments were performed independently at least three times with similar results, unless otherwise stated. The reproducibility of all key findings was confirmed in independent experiments conducted on different days and using independent biological samples.” Fig. 2e, (was Fig. 3e), “Control $n=3$; DEN $n=4$ animals” Fig. 3e, (was Fig. 4e), “Representative images from 6 animals” Fig. 3g, (was Fig. 4g), “Representative images from 6 mice” EDF 1c, “from 5 mice per timepoint” EDF 1d,e, “from 18 mice” EDF 1f, “representative image from 4 mice” EDF 1g, “$n=4$ (1m), 4 (9m), 10 (12m), 2 (14m).” EDF 2g, “Data from $n=3$ animals per timepoint” EDF 2h, “Data from $n=4$ animals” EDF 2j, “Data from 10 biological replicates” EDF 4k, “data from 5 animals”

		EDF 4m, "5 and 3 animals for low and high TAM dose, respectively." EDF 4o, "DEN treated animals represent n=4 (high TAM) and 3 (low TAM)" EDF 6d, "n=3 mice" EDF 6k, "Representative confocal image of a nascent tumour from 6 animals" EDF 6q, "representative data from 5 animals >12m post DEN" EDF 8m, "Representative confocal images from 6 mice" EDF 8o, "Representative confocal image from 10 mice" EDF 10g, "n=5 (control) 4 (BLEO) mice"
6.	Data availability: This journal strongly supports public availability of data and custom code associated with the paper in a persistent repository where they can be freely and enduringly accessed or as a supplementary data file when no appropriate repository is available. If data and code can only be shared on request, please explain why in your data Availability Statement, and also in the correspondence with your editor. For more information, please refer to https://www.nature.com/nature-research/editorial-policies/reporting-standards#availability-of-data Please ensure that datasets deposited in public repositories are now publicly accessible, and that accession codes or DOI are provided in the "Data Availability" section. As long as these datasets are not public, we cannot proceed with the acceptance of your paper. For data that have been obtained from publicly available sources, please provide a URL and the specific data product name in the data availability statement. Data with a DOI should be further cited in the methods reference section.	All datasets are made public: - GSE271962 - https://www.ncbi.nlm.nih.gov/geo/query/acc.cgi?acc=GSE271962 - ERP134942 - https://www.ebi.ac.uk/ena/browser/view/ERP134942 - The AlcoleaLab GitHub page - https://zenodo.org/records/17802564
7.	Micrographs: Please ensure that all micrographs include a scale bar and this scale bar is defined on the panels or in the figure legends.	
8.	Additional Note: In the Source data file, extended data figure 6p is mislabelled as 6q. Kindly rectify this.	Thanks for pointing this out. This has now been corrected.